# Extraembryonic gut endoderm cells undergo programmed cell death during development

Julia Batki ®[1,7], Sara Hetzel ®[1,7], Dennis Schifferl[2], Adriano Bolondi ®[1], Maria Walther[1], Lars Wittler[2], Stefanie Grosswendt[3,4], Bernhard G. Herrmann[2,5] & Alexander Meissner ®[1,6] ✉

Despite a distinct developmental origin, extraembryonic cells in mice contribute to gut endoderm and converge to transcriptionally resemble their embryonic counterparts. Notably, all extraembryonic progenitors share a non-canonical epigenome, raising several pertinent questions, including whether this landscape is reset to match the embryonic regulation and if extraembryonic cells persist into later development. Here we developed a two-colour lineage-tracing strategy to track and isolate extraembryonic cells over time. We find that extraembryonic gut cells display substantial memory of their developmental origin including retention of the original DNA methylation landscape and resulting transcriptional signatures. Furthermore, we show that extraembryonic gut cells undergo programmed cell death and neighbouring embryonic cells clear their remnants via non-professional phagocytosis. By midgestation, we no longer detect extraembryonic cells in the wild-type gut, whereas they persist and differentiate further in p53-mutant embryos. Our study provides key insights into the molecular and developmental fate of extraembryonic cells inside the embryo.

During mouse gastrulation the three germ layers arise and then further differentiate to form the major tissues of the body[1]. Although ectodermal and mesodermal cells in the developing embryo originate exclusively from the pluripotent epiblast, both embryonic and extraembryonic cells contribute to the emerging gut endoderm[2]. Specifically, a subset of extraembryonic endoderm cells, the majority of which will give rise to the yolk sac, intercalate with epiblast-derived definitive endoderm cells[3]. Previous studies have shown that embryonic and extraembryonic cells of the gut approach transcriptional identities that correspond to organ progenitors along the anterior–posterior axis at embryonic day (E) 8.75 (refs. 4–6). The unexpected transcriptional similarity between gut cells of distinct origins is even more noteworthy given the substantial epigenetic differences in their progenitors[7,8].

At E6.5 approximately 80% of the extraembryonic genome is differentially methylated compared with the epiblast, including hypermethylation at hundreds of CpG islands that remain free of DNA methylation in somatic cells. It remains unknown whether the gut cells of extraembryonic origin are epigenetically reprogrammed to the canonical somatic landscape and if they can persist to differentiate into more specialized endodermal tissues.

Reliably tracking and characterizing extraembryonic cells in the gut (exGut) beyond early organogenesis is not trivial and previously used fluorescent reporters rely on genes that are also expressed in embryonic gut cells (emGut); as such, they are only suitable for tracking the extraembryonic cells until late-stage gastrulation[3,9]. More recently, single-cell RNA sequencing (scRNA-seq)-based approaches—which can

[1]Department of Genome Regulation, Max Planck Institute for Molecular Genetics, Berlin, Germany. [2]Department of Developmental Genetics, Max Planck Institute for Molecular Genetics, Berlin, Germany. [3]Berlin Institute of Health (BIH), Charité – Universitätsmedizin Berlin, Berlin, Germany. [4]Berlin Institute for Medical Systems Biology, Max Delbrück Center for Molecular Medicine in the Helmholtz Association, Berlin, Germany. [5]Institute for Medical Genetics, Charité – Universitätsmedizin Berlin, Berlin, Germany. [6]Department of Biology, Chemistry and Pharmacy, Freie Universität Berlin, Berlin, Germany. [7]These authors contributed equally: Julia Batki, Sara Hetzel. ✉e-mail: meissner@molgen.mpg.de

infer the lineage origins—were used to explore the plasticity of endoderm cells[4,5,10]. Integration of genetic lineage tracing has definitively confirmed the dual origin of the gut endoderm but only retrospectively and with limited transcriptional resolution[6]. Another approach to distinguish between embryonic and extraembryonic lineages is the tetraploid embryo complementation assay, which generates completely embryonic stem cell (ESC)-derived animals[11]. Using this assay with two different fluorescent labels, cells of extraembryonic origin were detected in the presumptive gut tube; however, these experiments are limited by the non-physiological tetraploid status of intercalating extraembryonic cells[12].

To overcome these limitations and investigate the epigenome as well as the subsequent fate of exGut cells, we developed a diploid two-colour fluorescent lineage-tracing strategy that can reliably distinguish cells with embryonic or extraembryonic origin through organogenesis and beyond. We confirm the presence of extraembryonic cells in the gut endoderm and identify hundreds of differentially expressed genes. Moreover, we show that the DNA methylation landscape of these cells remains extraembryonic despite their overall transition to an embryonic transcriptome. This genome-wide epigenetic memory explains many of the latent transcriptional differences. We also find that exGut cells are selectively eliminated by E13.5 and their remnants are taken up by neighbouring embryonic cells through non-professional phagocytosis. The programmed cell death is prevented in *p53*-mutant embryos and the persisting exGut cells can further differentiate despite the continued maintenance of the non-canonical epigenome.

## Results

### Two-colour labelling of embryonic and extraembryonic cells

To investigate the molecular state and long-term developmental fate of exGut endoderm cells, we designed a lineage-tracing strategy where two constitutively expressed fluorescent proteins distinctly and permanently label the embryonic and extraembryonic lineages before gut endoderm formation. Specifically, we aggregated diploid mCherry-labelled (mCherry[+]) pre-compaction morula and diploid GFP-labelled (GFP[+]) mouse ESCs cultured with serum and leukaemia inhibitory factor (LIF; Fig. 1a). We established that this combination can create a developmental bias that yields ESC-derived diploid embryos—along with morula-derived diploid extraembryonic tissues—rather than chimaeric offspring[12,13] (Fig. 1b and Extended Data Fig. 1a,b). The selective post-aggregation lineage segregation can already be seen in the distinct localization of labelled cells at the blastocyst stage (Extended Data Fig. 1c). The outcome and overall efficiency of this approach are comparable to tetraploid complementation by morula aggregation but without the disadvantage of generating tetraploid extraembryonic cells (Extended Data Fig. 2a–c)[11,14]. In contrast, diploid complementation via blastocyst injection resulted in the expected chimaeric mCherry–GFP embryos (Extended Data Fig. 2d–f)[15,16]. Collectively, our assessment of aggregation and injection methods showed that the distinct developmental timing of the aggregated cells allows us to avoid the contribution of morula cells to the embryonic lineage, which makes it ideal to selectively investigate the fate of both embryonic and extraembryonic cells.

We further validated the developmental contribution to the embryo and yolk sac in four independent experiments that were collected at E9.5. Using fluorescence microscopy, we confirmed that embryos were either GFP[+] (ESC-derived, *n* = 54) or mCherry[+] (morula-derived, *n* = 2). The two mCherry[+] embryos seemed to reflect a failed aggregation and were excluded from further analyses (Extended Data Fig. 1d,e). The yolk sac tissue contains both mCherry[+] and GFP[+] cells, which would be consistent with yolk sac endoderm (YsEndo) of primitive endoderm origin and embryonic mesodermal cells, including primitive blood (Extended Data Fig. 1b)[1]. The only mCherry signal in the GFP[+] embryos was consistently located in the presumptive gut tube (Fig. 1b). We confirmed the gut localization of these diploid

mCherry[+] extraembryonic cells inside GFP[+] embryos by immunofluorescence staining for FOXA2 (a transcription factor expressed in the gut endoderm) and E-CADHERIN (E-CAD; an epithelial marker; Fig. 1c and Extended Data Fig. 1f).

Together, using our selective diploid aggregation approach, we confirmed previous work showing extraembryonic cell contribution to the gut and could now track them beyond gastrulation.

### Dual[+] cells are embryonic in origin

To investigate exGut cells in more detail, E9.5 lineage-traced embryos were subjected to fluorescence-activated cell sorting (FACS) where we detected 0.01–0.33% mCherry[+] cells of extraembryonic origin (*n* = 9 embryos), which coexisted among the substantially more abundant GFP[+] cells of embryonic origin (>98%; Fig. 1d and Extended Data Fig. 3a,b). Surprisingly, we also observed a population of dual-labelled cells (dual[+]; 0.2–1%) that were clearly GFP[+] but had varying levels of mCherry signal. We independently confirmed the presence of the single-labelled and dual[+] cells in the embryo using light-sheet microscopy, and closer inspection of the dual[+] cells showed GFP[+] cells with mCherry[+] foci (Fig. 1e, white arrowheads, and Extended Data Fig. 3c).

To determine the identity and origin of the dual[+] cells, we isolated both dual[+] and mCherry[+] cells from 15 pooled embryos (Extended Data Fig. 3d) and performed multiplexed scRNA-seq[17]. We captured a total of 3,353 dual[+] and 471 mCherry[+] cells after pre-processing (average of 4,761 genes and 24,984 captured molecules per cell; Supplementary Table 1). Using de novo clustering of the dual[+] cells, we detected seven cell states and annotated these based on published markers from single-cell atlases[4,5,18,19]—five clusters, containing the majority of dual[+] cells, correspond to gut endoderm organ progenitors and the remaining two clusters appear to be mesodermal cell types (Fig. 1f and Extended Data Fig. 3e,f). Next, we took advantage of the single-cell map of dual[+] cells and showed that mCherry[+] cells distributed across the different gut endoderm cell states when projected onto the dual[+] cell reference (Fig. 1f and Extended Data Fig. 3e–g). This is in line with our above-described localization of mCherry[+] cells throughout the gut tube and their overlap with FOXA2 and E-CAD (Fig. 1c and Extended Data Fig. 1f). Expression of previously identified extraembryonic marker genes—such as *Rhox5* and *Trap1a*—is limited to mCherry[+] cells and further confirms their extraembryonic origin[4–6]. In contrast, these genes are not expressed in dual[+] cells, which provides support for their embryonic origin (Fig. 1g).

Together, our analysis showed that mCherry[+] cells in the embryo correspond to exGut endoderm and assigned dual[+] cells as embryonic endoderm and mesoderm.

### Extraembryonic cells die and are cleared by phagocytosis

To explore how the dual[+] cells arise, we performed live imaging of ex utero-cultured embryos using confocal microscopy starting at E7.5 (Supplementary Videos 1 and 2). We found that the mCherry[+] cells in the embryo proper have a spatially dispersed distribution, as previously shown for AFP-labelled exGut cells[3]. Interestingly, in addition to intact mCherry[+] cells, mCherry[+] foci are present at this early developmental stage (Fig. 2a, yellow asterisks). By tracking mCherry[+] cells, we detected events where they die and fragment over a 2 h time window, followed by the emergence of mCherry foci in nearby GFP[+] cells (Fig. 2b and Extended Data Fig. 4a). These mCherry foci became positive for LysoTracker, a dye that labels acidic compartments such as phagolysosomes (Extended Data Fig. 4b,c and Supplementary Video 3). In addition, we found that mCherry[+] cells stain positive for cleaved CASPASE3 (Fig. 2c)[20]. Although both the imaging and expression analyses argue against cell fusion as the source of the dual[+] cells, this possibility cannot be fully excluded.

We observed that mCherry[+] foci are more prominent at the anterior part of the gut endoderm at E7.5 (Fig. 2d). We thus compared the

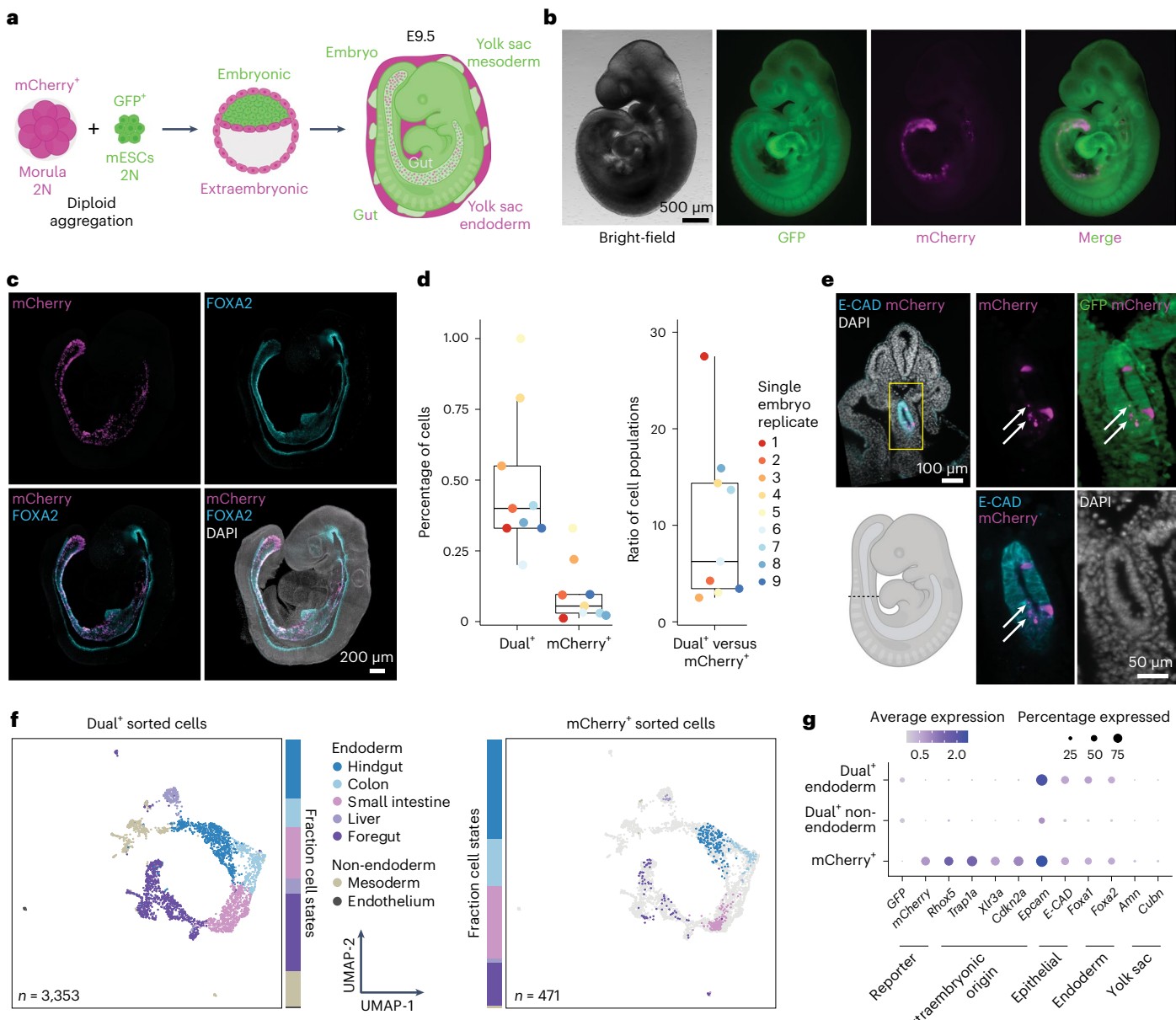

**Fig. 1 | Two-colour lineage tracing identifies dual-labelled embryonic cells.**
**a**, Schematic of the two-colour lineage labelling strategy for lineage tracing (2N indicates that cells are diploid). Embryonic versus extraembryonic lineage segregation can be seen at the blastocyst stage (Extended Data Fig. 1c). At E9.5, embryos are GFP+ and only the gut contains a small fraction of mCherry+ extraembryonic cells (see **b**). **b**, Bright-field (left) and fluorescence (right) microscopy images of an E9.5 embryo generated via the two-colour lineage tracing (n = 54; one representative embryo is shown). **c**, Maximum-intensity projection of optical sections acquired by confocal laser scanning microscopy showing an E9.5 embryo and confirming the presence of mCherry+ extraembryonic cells specifically in the gut, which is positive for FOXA2 (additionally expressed in the notochord and floor plate). Nuclei were stained with 4,6-diamidino-2-phenylindole (DAPI) and immunofluorescence was used for mCherry and FOXA2 (n = 3; one representative embryo is shown). **d**, Percentage of dual+ and mCherry+ cells (left) as well as the ratio of these two populations (right) in E9.5 embryos analysed by flow cytometry. Individual embryos are indicated by colour-coded dots (n = 9). Boxplots: the lines denote the median, the edges denote the interquartile range (IQR), whiskers denote 1.5× the IQR and minima/maxima are defined by dots. **e**, Transversal optical section of an E9.5 embryo acquired by light-sheet imaging (the dashed line in the schematic depicts the axial position, bottom left). E-CAD marks the surface ectoderm and gut endoderm. Magnified views of the gut (yellow box) are shown; mCherry foci are highlighted (white arrows). Nuclei were stained with DAPI and immunofluorescence was used for mCherry and E-CAD (n = 3; a section from one representative embryo is shown). **f**, Uniform manifold approximation and projection (UMAP) coloured by the assigned cell states showing dual+ (left) and mCherry+ (right) cells subjected to scRNA-seq (the dual+ population is indicated in grey on the right). The fractions of cells belonging to the individual cell states are indicated with the bars. **g**, Average log-normalized scRNA-seq expression of reporter transgenes and known marker genes of the indicated cell types. Expression is shown separately for embryonic dual+ endoderm and non-endoderm as well as mCherry+ extraembryonic cells.

E9.5 scRNA-seq endoderm cell cluster compositions of intact exGut cells (mCherry+ only) with emGut cells containing extraembryonic remnants (dual+), which we had sorted into three populations—that is, low, medium and high mCherry intensity (Extended Data Figs. 3d and 4d). Remaining intact mCherry+ cells have a clear bias towards posterior endoderm progenitors, which is in agreement with exGut cells being enriched in the hindgut/midgut (Fig. 2e,f)[3,4]. In contrast, dual+ cells have a tendency towards anterior endoderm types, with a

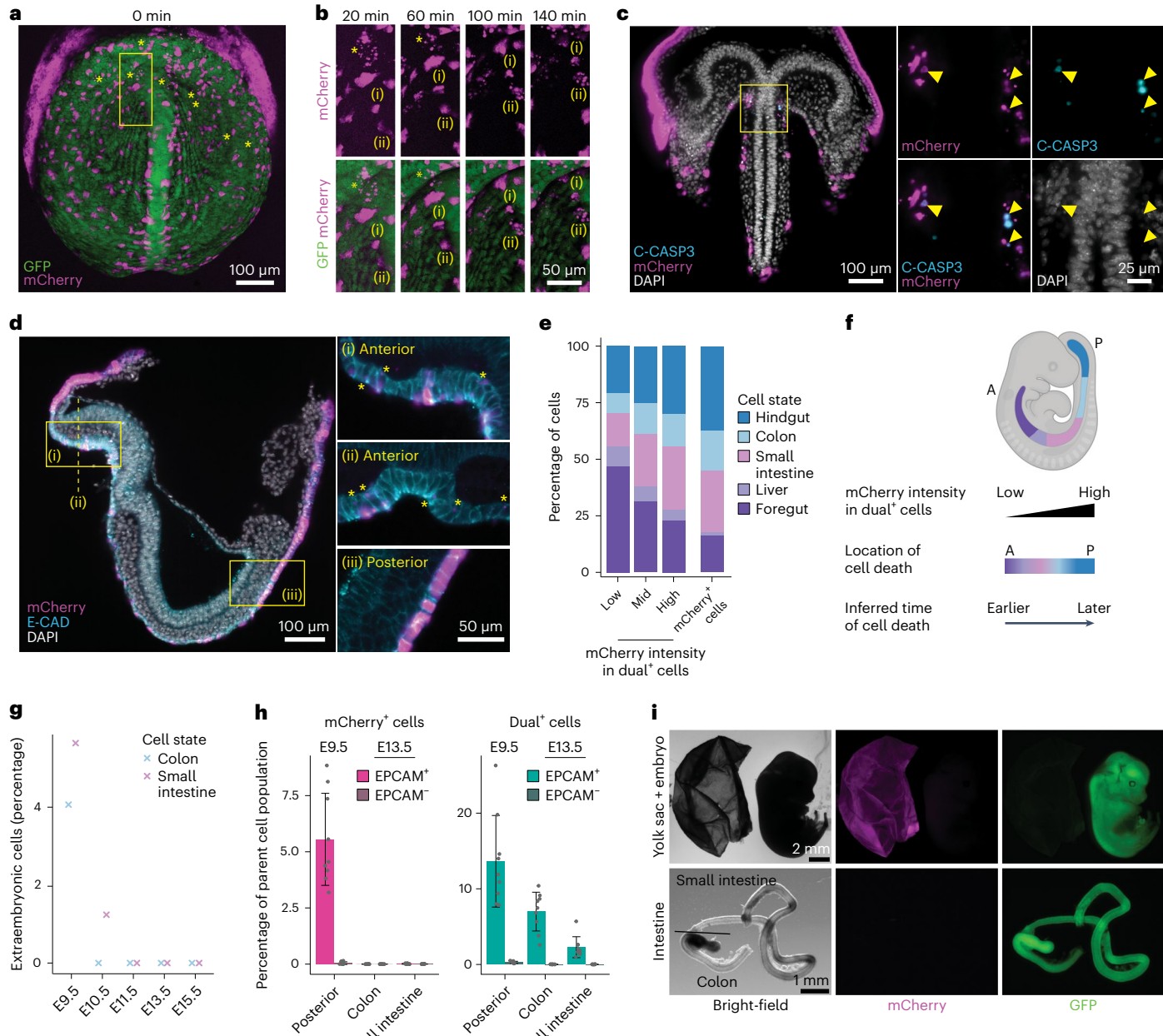

**Fig. 2 | Elimination of exGut cells by midgestation. a,** Maximum-intensity projection of optical sections (confocal laser scanning microscopy) showing an E7.5 embryo at the start of the ex utero culture and live imaging. In addition to intact mCherry⁺ extraembryonic cells in the embryo, mCherry⁺ foci were detected (yellow asterisks; *n* = 4, one representative embryo is shown). **b,** Magnified view of the region in the yellow box in **a** at different time points of the live imaging. (i),(ii), Two mCherry⁺ cells, which become fragmented, have been highlighted. **c,** Ventral view (light-sheet microscopy) of an E7.5 embryo. The magnified views (right) of the region in the yellow box in the main image (left) show mCherry⁺ cells that are positive for cleaved CASPASE3 (C-CASP3; yellow arrowheads). Nuclei were stained with DAPI and immunofluorescence was used for C-CASP3 and mCherry (*n* = 3, one representative embryo is shown). **d,** Lateral view of an E7.5 embryo (left). Magnified views of the yellow boxes ((i) and (iii)) and the yellow dashed line ((ii), transversal section) are provided (right). Yellow asterisks highlight mCherry⁺ foci. Nuclei were stained with DAPI and immunofluorescence was used for

mCherry and E-CAD (*n* = 3, one representative embryo is shown). **e,** Percentage of cells assigned to endodermal cell states in the different sort populations (low, intermediate and high dual⁺, and mCherry⁺) from E9.5 lineage-traced embryos using scRNA-seq. **f,** Schematic of the gut endoderm organ distribution along the anterior–posterior axis (top) and a summary of the proposed spatiotemporal regulation of exGut cell elimination (bottom). **g,** Percentage of cells with extraembryonic origin (defined as Rhox5⁺Trap1a⁺ cells) in the colon and small intestine from E9.5 to E15.5 (scRNA-seq data from Zhao et al.[19]). **h,** Proportion of mCherry⁺ and dual⁺ cell content of the indicated parent cell populations (EPCAM⁺ and EPCAM⁻) in the posterior part of E9.5 embryos and E13.5 organs (*n* = 9). The bars denote the mean, the error bars denote the s.d. and individual replicates are shown as dots. **i,** Bright-field and fluorescence microscopy images of a lineage-traced E13.5 embryo and its corresponding yolk sac. The intestine was manually separated into the colon and small intestine, indicated by the black line (*n* = 4, one representative embryo is shown). A, anterior; P, posterior.

clear correlation between mCherry intensity and anterior–posterior positioning: dual⁺ cells with lower mCherry fluorescence are more frequently found with anterior cell states, whereas those with high

mCherry intensities are enriched for posterior cell states. This implies that mCherry intensity reflects the time of the elimination, where lower levels of mCherry in GFP⁺ cells would correspond to earlier events.

Combined, our data point to programmed cell death of exGut cells and their remnants being cleared via non-professional efferocytosis (a form of phagocytosis) by neighbouring emGut cells[21], which results in dual[+] cells with a distinct spatiotemporal pattern.

## Extraembryonic cells are eliminated by midgestation

Next, we analysed an available scRNA-seq atlas of the mouse gastrointestinal tract between E9.5 and E15.5 (ref. 19), and detected exGut cells via *Rhox5* and *Trap1a* expression at E9.5 but not at later stages (Fig. 2g and Extended Data Fig. 4e). This points to their complete elimination, although it is also possible that the extraembryonic cell population is too rare to be captured by scRNA-seq or the selected genes are no longer expressed.

In line with the single-cell data, at E9.5, the posterior half of the gut, which will develop into colon and small intestine, still contains a substantial fraction of mCherry[+] cells (Fig. 2h). To determine the ultimate fate of these exGut cells, we isolated lineage-traced embryos at E13.5, dissected the colon and small intestine, and used flow cytometry to quantify their endoderm (EPCAM[+]) and non-endoderm (EPCAM[−]) cell populations (Fig. 2h,i and Extended Data Fig. 4f–h). No mCherry[+] cells were detected in the colon of nine embryos, including both endoderm and non-endoderm. Analysis of the small intestine confirmed this observation: four of the nine embryos lacked any mCherry[+] cells, whereas the remaining five showed only minor traces of mCherry[+] cells (less than 0.04%). We also isolated cells from E12.5 embryos and observed elimination in endoderm populations already at this stage, with only a small number of mCherry[+] cells remaining in the non-endoderm fraction of the small intestine (Extended Data Fig. 4i). Note that the small intestine protrudes from the body cavity during these developmental stages[22] and thus the few detected mCherry[+] cells may also reflect contamination from mCherry[+] extraembryonic tissues. As controls, we confirmed that mCherry[+] cells have the potential to contribute to both organs by generating chimaeric GFP–mCherry embryos as well as that the mCherry reporter remains expressed by generating complete mCherry[+] embryos (Extended Data Fig. 5).

Together, our data demonstrate the elimination of extraembryonic cells from endodermal organs by midgestation.

## Origin-specific transcriptional signatures in the gut

For a more comprehensive transcriptome analysis of lineage-traced embryos, we adapted the Smart-Seq2-based protocol to low-input bulk samples[23]. We isolated E6.5 epiblast as well as distal and proximal extraembryonic endoderm (exEndo 1 and 2, respectively), the progenitor populations that differentiate into embryonic and exGut cells as well as extraembryonic yolk sac, respectively (Fig. 3a). At E9.5, we sorted YsEndo and gut cells from the posterior half of embryos (midgut and hindgut), where the majority of the remaining mCherry[+] exGut cells are found at this developmental stage (Extended Data Fig. 6a,b and Supplementary Table 2). This gut-cell isolation approach should minimize transcriptional differences due to spatial localization along the anterior–posterior axis. For differential gene expression analysis, we selected dual[+] emGut cells (EPCAM[+]) as a closely matched (stage and position) embryonic control (Extended Data Fig. 6b).

Although the exEndo 1 and 2 cell populations are transcriptionally highly similar to each other and different from the epiblast at E6.5, they give rise to exGut and YsEndo cells with strikingly diverged transcriptomes at E9.5. In contrast, the E9.5 exGut cells are transcriptionally similar to emGut cells (Fig. 3b and Extended Data Fig. 6c). Next, we compared E9.5 exGut and emGut cells and identified 302 differentially expressed genes (156 up- and 146 downregulated genes, which we termed 'exGut high' and 'exGut low', respectively; Fig. 3c, Extended Data Fig. 6d and Supplementary Tables 3,4). As expected, both mCherry and GFP were detected as differentially expressed in addition to the reported extraembryonic marker genes[4–6], which are specifically expressed in the mCherry[+] cells (Fig. 3c and Extended

Data Fig. 6c). Intriguingly, gene ontology analysis showed that exGut low genes are associated with axon guidance and components of the synaptic membrane (Fig. 3d and Extended Data Fig. 6e). These genes are mostly not expressed in the E6.5 extraembryonic cells and fail to reach the embryonic expression levels once activated in the E9.5 exGut cells (Fig. 3e). Genes related to these terms are generally involved in cell–cell communication and have well-described function in both neural and non-neural contexts[24]. These differences raise the possibility that exGut and emGut cells may use distinct cell–cell interaction modes. In contrast, the exGut high genes were enriched for known germline genes that also show high expression levels in the early extraembryonic lineage and apparently cannot be repressed to match the expression status in the emGut cells (Fig. 3d,e and Extended Data Fig. 6e). The germline signature is supported by a strong enrichment of exGut high genes on the X chromosome, a known hotspot for germline and placental genes[25] (Extended Data Fig. 6f,g). Independent of our lineage tracing, we found indications of the above-described transcriptional signatures by analysing published scRNA-seq data of E8.75 and E9.5 gut (Extended Data Fig. 6h).

In summary, our in-depth gene expression analysis identified gene sets that clearly distinguish gut cells based on their lineage origin.

## Global epigenetic memory in exGut cells

The known role of DNA methylation in silencing germline genes[26,27] suggested that their continued expression may be linked to the non-canonical distribution of DNA methylation in the extraembryonic cells[7,8]. Previous studies showed upregulation of a set of genes, including germline genes, in embryos lacking the de novo DNA methyltransferases DNMT3B and DNMT3A[28,29] (Extended Data Fig. 7a and Supplementary Table 5). Interestingly, we found that this gene set also displays overall higher expression in exGut cells (Fig. 4a).

To explore their DNA methylation landscape at the global level, we generated whole-genome bisulfite sequencing (WGBS) datasets matching the tissues of our RNA-seq cohort (Supplementary Table 6). Strikingly, and in contrast to the converging transcriptome, the exGut cells preserve their original hypomethylated genome and do not convert to the globally high levels present in the emGut cells (Fig. 4b,c and Extended Data Fig. 7b). The same epigenetic memory holds true for over a thousand CpG islands (CGIs) that are hypermethylated in the exEndo compared with the epiblast and remain unchanged in exGut cells (Fig. 4c and Supplementary Table 7). Next, we focused on gene promoters that have differential expression in E9.5 gut cells and found that select promoters of exGut low genes exhibit higher DNA methylation in exGut cells compared with emGut cells, whereas the opposite was true for a fraction of exGut high genes (Fig. 4d–f and Extended Data Fig. 7c). Interestingly, for many promoters with strong differences in DNA methylation, we could already observe a difference in methylation between the precursor cell types at E6.5 (Fig. 4e and Extended Data Fig. 7d,e). We detected exGut low genes that were partially methylated in the extraembryonic progenitor and could subsequently only be expressed at low levels in the exGut cells as well as exGut high genes—including DNA methylation-sensitive germline genes—that were already expressed in the extraembryonic progenitor at E6.5 and could not be silenced by E9.5 (Fig. 4e,f and Extended Data Fig. 7d–f).

These results demonstrate that exGut cells preserve the characteristic extraembryonic epigenome, which also explains many of the differentially expressed genes and highlights a striking molecular disparity to their neighbouring cells with embryonic origin.

## p53 disruption allows extraembryonic cells to persist

Our differential gene expression analysis at E9.5 identified genes that are expressed at higher levels in exGut cells and were shown to act downstream of p53 (a known effector in programmed cell death, encoded by the *Trp53* gene, hereafter referred to as *p53*), a signature that we could

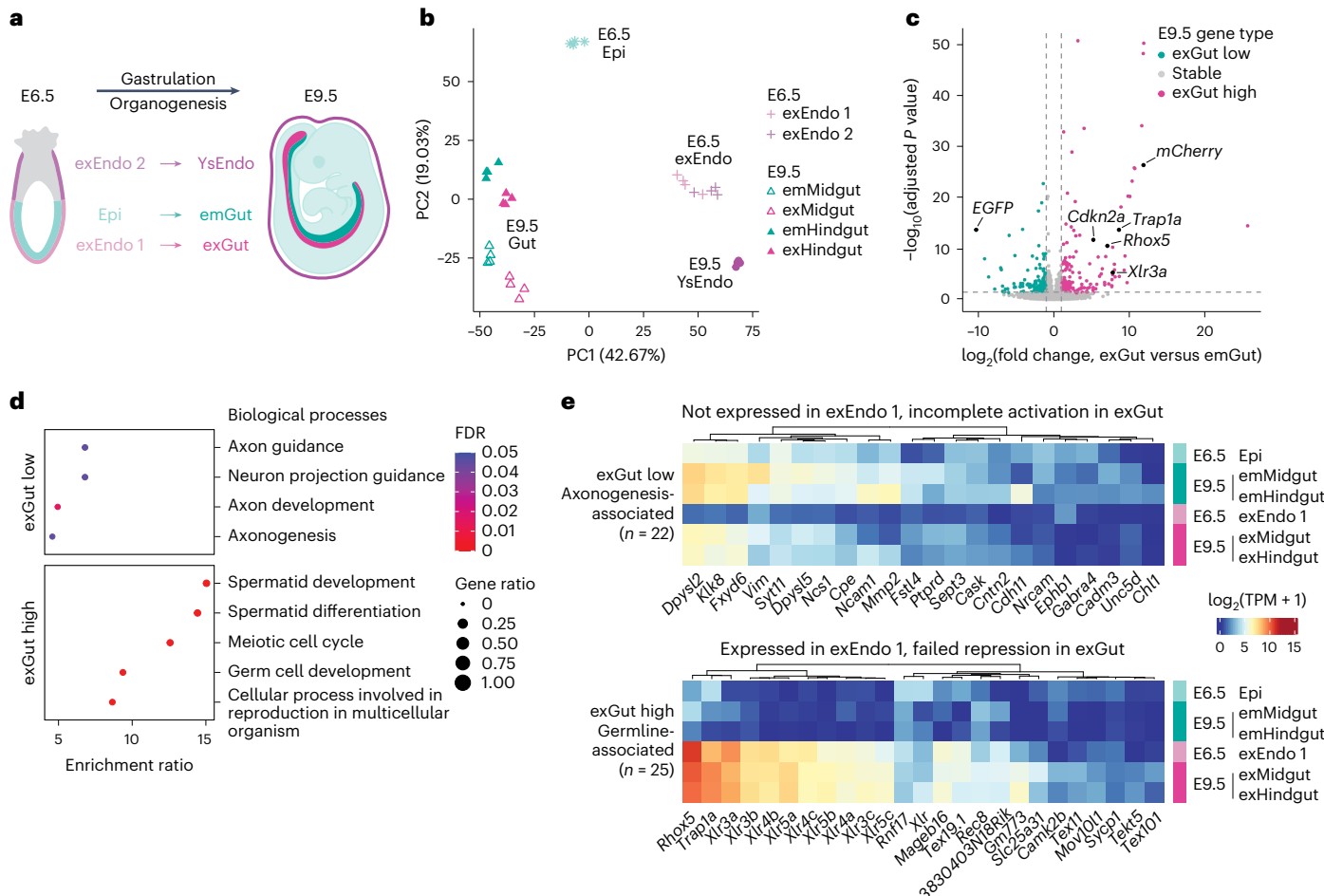

**Fig. 3 | Transcriptional signatures of gut cells reflect their lineage origins.**
**a**, Schematic of the origin and composition of gut endoderm: emGut cells (E9.5) originating from the embryonic epiblast (Epi, E6.5) and exGut cells (E9.5) originating from the distal part of the extraembryonic endoderm at E6.5 (exEndo 1) both contribute to the formation of gut endoderm. The proximal part of the extraembryonic endoderm (exEndo 2) gives rise to the E9.5 YsEndo. E6.5 Epi and E9.5 emGut cells are positive for GFP, whereas E6.5 exEndo 1 and 2, E9.5 exGut and YsEndo cells are positive for mCherry in lineage-traced embryos. E9.5 emGut cells can also be positive for mCherry in the case of dual+ cells that have taken up remnants of dying mCherry+ cells (not illustrated). **b**, Principal component analysis (PCA) of E6.5 and E9.5 RNA-seq samples based on the 5,000 most variably expressed genes. Samples are largely separated by tissue, with exGut cells grouping close but distinguishable from their embryonic counterparts. **c**, Genes that are differentially expressed (two-sided Wald test; P values were adjusted for multiple testing using false discovery rate (FDR)) between E9.5 exGut and emGut (including both midgut and hindgut). Genes that are expressed

at significantly higher levels in exGut cells compared with emGut cells (termed exGut high) include known extraembryonic marker genes as well as the mCherry transgene, whereas GFP is expressed at significantly higher levels in emGut cells (termed exGut low). Vertical and horizontal lines denote log2 fold change and adjusted P-value boundaries used for differential expression calling. **d**, Overrepresentation analysis of exGut low and high genes in biological processes. The exGut low genes are enriched in axonogenesis-related processes, whereas exGut high genes are enriched in germline- and meiosis-associated processes. **e**, Expression (log2-transformed) of a selection of exGut low (top) and high (bottom) genes associated with the terms shown in **d** in E6.5 progenitor and E9.5 gut cells. The selected exGut low genes are not expressed in the E6.5 exEndo 1 and are insufficiently activated in the E9.5 exGut cells. In contrast, exGut high genes are generally not expressed, or are expressed at low levels, in the E6.5 Epi and E9.5 emGut samples, whereas both the extraembryonic progenitor and exGut cells express them. TPM, transcripts per million.

also recapitulate in published scRNA-seq datasets (Fig. 5a and Extended Data Fig. 8a)[4,19,30]. To investigate whether p53 is directly involved in the lineage-specific cell elimination, we generated four *p53*-mutant E13.5 embryos by electroporating zygotes with Cas9 and three guide RNAs (gRNAs) targeting *p53* exons (Fig. 5b and Extended Data Fig. 8b). As controls we used four stage-matched wild-type embryos. We dissected the gastrointestinal tract, sorted EPCAM+ endoderm cells and performed multiplexed scRNA-seq, which recovered 9,278 and 9,710 single-cell profiles from wild-type and *p53*-mutant embryos, respectively (average of 3,849 genes and 14,774 captured molecules per cell; Supplementary Table 8). De novo clustering of the wild-type cells resulted in nine cell states, which we annotated based on known marker genes[19,31] as endoderm-derived epithelial cells of distinct parts of the colon, small intestine, stomach and pancreas (Fig. 5b,c and Extended Data Fig. 8c–e).

Next, we showed that the *p53*-mutant cells distribute similarly across the different organ clusters when projected onto the reference map of wild-type cells (Fig. 5b,c and Extended Data Fig. 8c–g). Using Rhox5 and Trap1a, consistent with our data above, we found virtually no sign of extraembryonic cells in the wild-type E13.5 embryos, whereas in the *p53* knockouts (KOs), we readily detected extraembryonic cells, even up to 6% in the proximal colon (Fig. 5d and Extended Data Fig. 8h).

To investigate whether additional time may lead to epigenetic resetting, we utilized our two-colour lineage-tracing strategy with a modification that allows for the generation of extraembryonic-specific KOs (Extended Data Fig. 9a). We collected embryos at E9.5 and isolated the small intestine and colon at E13.5. Using fluorescence microscopy and flow cytometry analysis, we found that mCherry+ extraembryonic cells persisted in the organs when *p53* was mutated

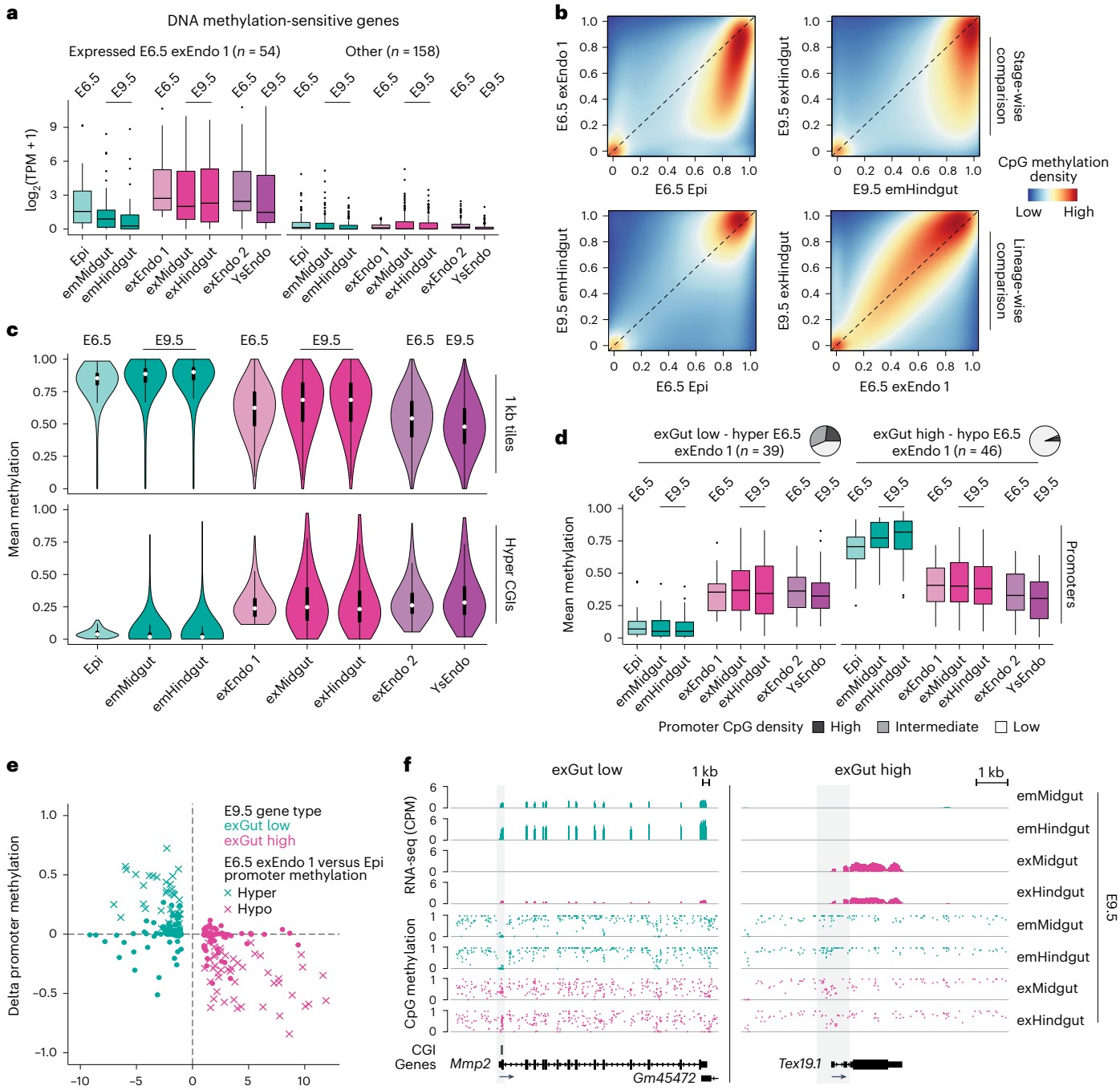

**Fig. 4 | Preserved epigenetic memory of lineage origins. a**, Expression levels (log$_2$-transformed) of DNA methylation-sensitive genes (defined by ref. 28,29; $n$ = 4 biological replicates) separated based on whether or not a gene is expressed in E6.5 exEndo 1. **b**, CpG-wise comparison of DNA methylation between E6.5 progenitors, E9.5 emGut and exGut cells of the hindgut (emHindgut and exHindgut, respectively; WGBS). Global DNA methylation correlated better with lineage (bottom) than with developmental stage (top). **c**, Average genome-wide methylation (top; one kilobase (kb) tiles, $n$ = 1,791,329) and the methylation of CGIs hypermethylated in the E6.5 exEndo 1 compared with the epiblast (bottom; hyper CGIs, $n$ = 1,121), WGBS. Boxplots: the white dots denote the median, the edges denote the IQR and whiskers denote 1.5× the IQR (minima/maxima are indicated by the violin plot range). **d**, Levels of promoter methylation of exGut low genes that are hypermethylated (left) and exGut high genes that are hypomethylated (right) in E6.5 exEndo 1 compared with the epiblast. The pie charts (top) indicate the promoter CpG density of the respective gene sets.

**c,d**, $n$ = 1 or 2 biological replicates. **a,d**, Boxplots: the lines denote the median, the edges denote the IQR, the whiskers denote 1.5× the IQR and outliers are represented by dots. **e**, Comparison of the log$_2$-transformed fold change of E9.5 differentially expressed genes (exGut low and high) with the respective delta promoter methylation between exGut and emGut. A lower DNA methylation level corresponds to higher gene expression, which is most pronounced for the promoters for which a difference in DNA methylation can already be observed in the E6.5 gut progenitors. **f**, Genome browser tracks of the *Mmp2* (exGut low) and *Tex19.1* (exGut high) locus showing RNA-seq coverage and WGBS. *Mmp2* is expressed in emGut samples where it is associated with an unmethylated promoter. The intermediate methylation of the promoter CGI in exGut samples only allows low expression levels in the exHindgut. *Tex19.1* is fully methylated in emGut samples, which corresponds to a complete silencing of the gene. In contrast, lower DNA methylation levels of the low CpG density promoter in exGut samples allow expression.

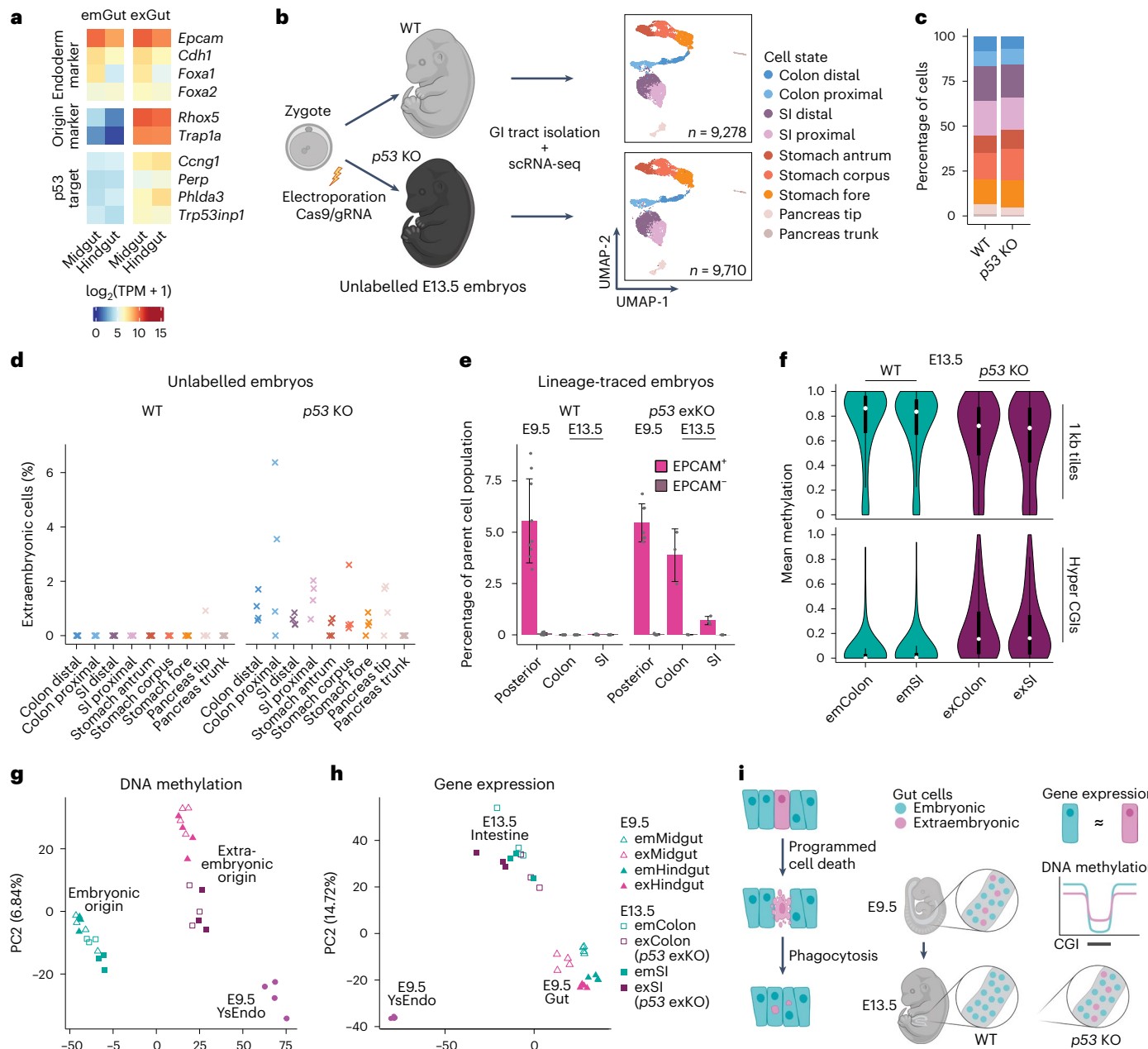

**Fig. 5 | p53 disruption allows the survival of exGut cells with origin-specific signatures. a**, Expression levels (log₂-transformed) of endoderm and extraembryonic origin marker genes as well as four exGut high genes that are known p53 target genes in E9.5 gut samples. **b**, Schematic of the generation of wild-type (WT) and *p53*-KO embryos using Cas9-mediated genetic perturbation (left). UMAP of WT (top) and *p53*-KO (bottom) epithelial cells sorted from the gastrointestinal (GI) tract of E13.5 embryos (right; *n* = 4 independent biological replicates per condition). Colours indicate the assigned cell state. **c**, Percentage of cells that belong to the individual cell states in WT and *p53*-KO GI tracts. **d**, Percentage of extraembryonic cells (Rhox5⁺Trap1a⁺ cells) in the WT and *p53*-KO organs of unlabelled embryos of the experiment shown in **b** and **c** (all four individual embryo replicates are shown). In the WT, only one cell with Rhox5 and Trap1a expression was found at E13.5, whereas a substantial fraction with extraembryonic origin survived in the *p53* KO. **e**, Comparison of the proportion of mCherry⁺ cell content in WT lineage-traced embryos (left; data from Fig. 2h)

and lineage-traced embryos with extraembryonic-specific *p53* KO (exKO; right) showing in the posterior part of E9.5 embryos (*n* = 5) and the colon and small intestine from E13.5 embryos (*n* = 3). The bars denote the mean, the error bars denote the s.d. and individual replicates are shown as dots. **f**, Average genome-wide methylation (top; 1 kb tiles, *n* = 583,771) and the methylation of CGIs hypermethylated in the E6.5 exEndo 1 compared with the epiblast (bottom; hyper CGIs, *n* = 1,292) in E13.5 tissues (reduced representation bisulfite sequencing, RRBS). The violin plot characteristics are the same as in Fig. 4c (*n* = 3 biological replicates). **g**, DNA methylation PCA of E9.5 and E13.5 RRBS samples based on the 5,000 most variably methylated 1 kb tiles where samples largely separate according to lineage origin. **h**, Gene expression PCA of E9.5 and E13.5 RNA-seq samples based on the 5,000 most variably expressed genes, where samples largely separate according to tissue type. **i**, Simplified schematic showing the developmental fate and molecular characteristics of extraembryonic cells in the gut. SI, small intestine.

(Fig. 5e and Extended Data Fig. 9b–f). Moreover, we recapitulated the *p53*-KO-dependent survival and proliferation of exGut cells in vitro (Extended Data Fig. 10a–c). Next, we profiled DNA methylation

(Supplementary Table 9) and found that surviving *p53*-mutant extraembryonic cells at E13.5 still preserve the characteristic extraembryonic epigenome (Fig. 5f). A comparison with E9.5 samples showed that the

surviving *p53*-mutant extraembryonic cells were most similar to cells with the same lineage origin based on the DNA methylome, regardless of the developmental stage and tissue (Fig. 5g). In contrast, when we compared their transcriptional profiles (Supplementary Tables 2 and 3), extraembryonic cells at E13.5 were more similar to embryonic cells of the same developmental stage and tissue than to the YsEndo and E9.5 gut cells with extraembryonic origin, which suggests a differentiation trajectory similar to their embryonic counterparts (Fig. 5h). Notably, latent origin-specific gene expression signatures are still largely retained in line with the epigenetic memory (Extended Data Fig. 10d,e).

In summary, *p53*-mutant extraembryonic cells can survive despite maintaining their extraembryonic epigenome, which remarkably also allows for continued development and contribution to organs at later stages.

## Discussion

Embryonic development holds many unresolved mysteries, including the fate and role of extraembryonic cells that contribute to the embryonic gut. Here we showed that these exGut cells retain a global epigenetic memory, which also explains some of the remaining transcriptional differences. By midgestation, programmed cell death and clearance via non-professional phagocytosis results in their elimination, which can be overcome by loss of p53 and leads to the survival of molecularly distinct extraembryonic cells in the gut (Fig. 5i).

To support normal development and maintain homeostasis in adult tissues, dying cells and their remnants have to be cleared[32], which is particularly relevant in epithelial tissues to keep their overall integrity[33]. Within these contexts, dying cells are often extruded, whereas we observed that extraembryonic gut remnants are taken up, raising the possibility that they may serve an additional role as shown for zebrafish embryos[34]. Clearance of these extraembryonic remnants precedes the emergence of macrophages (immune cells with professional phagocytic activity)[35]. We found evidence that they are mainly taken up by neighbouring epithelial gut cells, thereby providing an argument for non-professional phagocytosis as the clearance mechanism[32]. This result also supports the immune-like function of epithelia, which was reported for pre-implantation vertebrate embryos[36]. Epithelial cells were also shown to play a prominent role in adult immune surveillance, where oncogenic neighbours are eliminated by cell competition in a process termed epithelial defence against cancer[37]. The nature of exGut cell elimination resembles instances of cell competition, including those described for the epiblast and surface ectoderm of mouse embryos[38]. Further work is needed to test whether exGut cells are similar to the so-called 'loser' cells. Molecular recognition of distinct, potentially harmful cells depends on cell–cell communication and cell surface proteins have recently been shown to mediate aberrant cell removal in an epithelial tissue[39]. Along these lines, we found differentially expressed genes encoding membrane proteins, suggesting that a cell surface code may ensure the selective elimination of extraembryonic cells in the embryonic context.

Our detailed transcriptional analysis demonstrated, in agreement with previous studies, that extraembryonic cells are capable of acquiring distinct transcriptional states that correspond to multiple endodermal organ progenitors. However, our work also expands the limited set of previous marker genes that distinguish gut cells with the two lineage origins. Many of these genes, including germline genes, are expressed in the extraembryonic progenitors at the onset of gastrulation, pointing to transcriptional memory. Among them is Trap1a, which shows little to no expression in normal somatic cells and was the first tumour rejection antigen identified in mice[40]. More generally, cancer/testis antigens are typically restricted to the germline and extraembryonic tissues, and are aberrantly expressed in various cancers[41]. A large fraction of these genes is located in clusters on the X chromosome and sensitive to loss of DNA methylation, which are characteristics shared with many of the exGut marker genes. Furthermore, our DNA methylome analysis demonstrates the relationship between some transcriptional differences and a more globally distinct epigenome, where exGut cells retain their original epigenetic status, including intermediate genome-wide levels and hypermethylation at select CGIs. It is unclear how this distinct non-embryonic form of genome regulation can support the range of transcriptional states that emerge along the gut axis. The disruption of *p53* in exGut cells demonstrated that the window and ability of these cells to differentiate can be extended—although how long remains to be determined in future studies.

In summary, our data settle key questions about the molecular state and fate of extraembryonic cells in mouse post-implantation development while adding to the ongoing investigations of reprogramming cellular identities as well as epithelial immune-like function.

## Online content

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

## Methods

### Laboratory animals

All animal work performed in this study was approved by the local authorities (LAGeSo Berlin, license numbers G0243/18 and G0098/23). Mice were kept in individually ventilated cages under specific pathogen-free conditions in animal rooms with a light cycle of 12 h/12 h, temperatures of 20–24 °C and humidity of 45–65%. The mice received autoclaved water and a standard rodent diet ad libidum. Hsd:ICR (CD-1) and C57BL/6J mice were obtained from Envigo/Inotiv.

### Culture of mESCs

The mouse ESCs (mESCs) used in this study originate from the F1G4 background[42]; the parental male cell line was obtained from the laboratory of A. Nagy (Toronto, Canada). The mESCs were cultured on gelatinized plates coated with mitotically inactive primary CD-1 mouse embryo fibroblasts (MEFs) in the presence of serum and LIF in mESC medium (KnockOut DMEM (Thermo Fisher Scientific, 10829018), 15% fetal bovine serum (FBS; Thermo Fisher Scientific, 16140071), 1×GlutaMAX supplement (Thermo Fisher Scientific, 35050038), 1×non-essential amino acids (Thermo Fisher Scientific, 11140035), 1:1,000 2-mercaptoethanol (Thermo Fisher Scientific, 21985023), 1×penicillin–streptomycin (Thermo Fisher Scientific, 15140122) and laboratory-purified recombinant LIF). The cells were passaged every 2–3 d by first rinsing the cells twice with Dulbecco's phosphate-buffered saline (DPBS; Thermo Fisher Scientific, 4190144), detaching them using 1×TrypLE express enzyme (Thermo Fisher Scientific, 12604021) and then plating on MEFs; the medium was changed every day.

### Generation of GFP⁺ reporter mESCs

Wild-type F1G4 cells were used to generate the GFP⁺ reporter mESCs. The targeting vector contained the CAG promoter, EGFP coding sequence and SV40 polyadenylation signal, flanked by sequences homologous to the Rosa26 locus (the genomic coordinates are listed in Supplementary Table 10). This was co-transfected with a plasmid encoding Cas9 (PX459; Addgene, 62988) and containing a gRNA targeting the Rosa26 locus (the gRNA sequence is listed in Supplementary Table 10) using FuGENE HD transfection reagent (Promega, E2311) as per the manufacturer's instructions. Briefly, 400,000 mESCs were plated the day before transfection. For the transfection, 8 µg of each plasmid DNA was diluted in 125 µl Opti-MEM (Thermo Fisher Scientific, 31985062); 25 µl of FuGENE reagent (room temperature) was diluted with 100 µl Opti-MEM. The diluted FuGENE was added to the diluted DNA, incubated at room temperature for 15 min and added dropwise to the cells. The medium was changed the next day. GFP-expressing mESCs were isolated by FACS 48 h after transfection and single-cell-derived clones stably expressing GFP were picked and expanded (GFP⁺ mESCs).

### Generation of mCherry⁺ reporter mice

To generate mCherry⁺ reporter mESCs, the PiggyBac EF1a–mCherry transposon plasmid from the previously published construct in Chan et al.[6] was used with the following modifications. The triple gRNA cassette was removed and the plasmid was re-ligated to ensure stable expression of mCherry driven by a full-length intron-containing EF1a promoter. Ubiquitous chromatic opening element and insulators were maintained to preserve a high level of mCherry expression in every cell. Wild-type F1G4 cells were transfected with this EF1a–mCherry plasmid and the improved super piggyBac transposase using Xfect mESC transfection reagent (Takara, 631320) according to the manufacturer's instructions. After 48 h, mCherry-expressing cells were isolated by FACS and single-cell-derived clones stably expressing mCherry were picked and expanded. Next, $G_0$ mice were generated via tetraploid complementation assay[43] and $G_0$ male mice were crossed with C57BL/6J females (age, ≥8 weeks) to generate $F_1$ animals. For the establishment and maintenance of the mCherry⁺ colony, C57BL/6J females (age, ≥8 weeks) were regularly mated with mCherry⁺ males (age, ≥8 weeks);

mCherry expression was verified for the different generations using DFP-1 Dual Fluorescent Protein Flashlight with Royal Blue and Green excitation (NIGHTSEA).

### Embryo complementation assays

For all complementation assay types described below, mCherry⁺ males (age, ≥8 weeks) were mated with Hsd:ICR (CD-1) females (age, 6–20 weeks) for embryo isolation at the specified stages. The embryos are cultured in potassium simplex optimized medium (KSOM; Cosmo Bio, R-B074) under mineral oil at 37 °C and 5% $CO_2$. Embryos developing to expanded blastocysts were re-transferred into the uterine horns (maximum 15 embryos on each side) of a pseudopregnant CD-1 female (age, 6–20 weeks) at 2.5 d post coitus (generated by mating with vasectomized CD-1 males; age, ≥12 weeks). Post-implantation embryos were isolated according to the indicated embryonic day, with a 24 h developmental delay accommodated due to the re-transfer procedure.

### Diploid complementation assay by morula aggregation—two-colour lineage tracing

Pre-compaction morula-stage (4–8 cell stage) embryos were isolated at 2.5 d post coitus, treated with Acidic Tyrode's solution (Sigma-Aldrich, T1788) to remove the zona pellucida and placed in handmade depressions of the aggregation plate (Falcon, 353001). GFP⁺ mESCs were thawed and cultured on MEFs, in serum and LIF conditions, for 2 d and colonies were detached from the plate by a brief (approximately 30 s) trypsinization step (Thermo Fisher Scientific, 25300054). A small colony of 8–15 cells was then picked and added to each depression containing a single mCherry⁺ pre-compaction morula. The aggregates were incubated for 48 h, and those that successfully formed expanded blastocysts were re-transferred. We refer to this strategy as the two-colour lineage tracing, and GFP⁺ embryos with gut-specific mCherry signal were used for downstream experiments. GFP⁺ cells are male as they were derived from the parental F1G4 mESC line, whereas mCherry⁺ cells can be either female or male, as pre-implantation embryos were generated via natural mating. Note that low mESC quality, incorrect morula-staging or aggregate preparation can result in technical failures that lead to chimaeric embryos. Previously, chimaeric offspring were obtained via diploid complementation assay by morula aggregation but using a different mESC line (R1)[12,43].

### Tetraploid complementation assay by morula aggregation

Embryos at the two-cell stage were isolated at 1.5 d post coitus and electrofused in a CF.150/B cell fusion device (BLS) using the following settings: 2 V HF sinus to align the embryos along the electrodes, 30 V pulse, 35 µs pulse length and one repeat pulse. The embryos were then monitored for fusion. The following day, mCherry⁺ pre-compaction tetraploid morulae (4–8 cell stage) were treated with Acidic Tyrode's solution to remove the zona pellucida and placed in handmade depressions of the aggregation plate. GFP⁺ mESCs were thawed and cultured on MEFs, in serum and LIF conditions, for 2 d and colonies were detached from the plate by a brief (approximately 30 s) trypsinization step. A small colony of 8–15 cells was then picked and added to each depression containing a single mCherry⁺ tetraploid morula, followed by the addition of another zona pellucida-free mCherry⁺ tetraploid morula. The aggregates were incubated for 48 h and those that successfully formed expanded blastocysts were re-transferred.

### Diploid complementation assay by blastocyst injection

Pre-compaction morula-stage embryos were isolated at 2.5 d post coitus. These were cultured overnight in KSOM medium at 37 °C and 5% $CO_2$ until they reached the expanded blastocyst stage. GFP⁺ mESCs were thawed and cultured on MEFs, in serum and LIF conditions, for 2 d and then 8–12 single cells were injected into the blastocyst using an Eppendorf CellTram 4r oil microinjector. Embryos were cultured for at least 2 h before re-transfer.

## Generation of *p53*-KO embryos

Zygote electroporation was used to generate KO embryos as described previously[18]. Briefly, pronuclear stage 3 zygotes were isolated 0.5 d post coitus from CD-1 females (age, 6–20 weeks) mated with CD-1 males (age, ≥8 weeks). Electroporation reactions were set up according to the manufacturer's guidelines (Integrated DNA Technologies, Alt-R CRISPR–Cas9 ribonucleoprotein (RNP) complex protocol) just before the electroporation. Briefly, 4.5 μl of 200 μM Alt-R CRISPR–Cas9 tracrRNA (Integrated DNA Technologies, 1072533) and 3 μl of each 100 μM Alt-R CRISPR–Cas9 crRNA XT targeting *p53* (the gRNA sequences are listed Supplementary Table 10) were mixed, heated to 95 °C for 5 min and allowed to anneal at room temperature for 5 min. This crRNA–tracrRNA mix and 3 μl of 61 μM Alt-R S.p. HiFi Cas9 nuclease V3 (Integrated DNA Technologies, 1081059) was diluted in 133.5 μl Opti-MEM and incubated at room temperature for 20 min. We used a NEPA21 electroporator with a 5 mm gap electrode chamber and the following settings: for the poring pulse, 225 V, 2 ms pulse length, 50 ms pulse interval, four pulses, 10% decay rate and + polarity; for the transfer pulse, 20 V, 50 s pulse length, 50 ms pulse interval, five pulses, 40% decay rate and alternating +/− polarity. Electroporated zygotes were washed and cultured in KSOM medium at 37 °C and 5% $CO_2$ until they reached the expanded blastocyst stage. They were then re-transferred into the uterine horns (maximum 15 embryos on each side) of a pseudopregnant CD-1 female at 2.5 d post coitus (generated by mating with vasectomized CD-1 males). Post-implantation embryos were isolated according to the indicated embryonic day, with a 24 h developmental delay accommodated due to the re-transfer procedure. As a control, stage-matched embryos were used from natural mating of wild-type CD-1 females (age, 6–20 weeks) and males (age, ≥8 weeks), with midday of vaginal plug considered to be E0.5. As the embryos were generated via natural mating, they can be either female or male (wild-type, three females and one male; *p53*-KO, one female and three males).

## Generation of extraembryonic lineage-specific *p53*-KO embryos combined with the two-colour lineage-tracing strategy

We generated *p53*-KO zygotes as described in the section 'Generation of *p53*-KO embryos', except that zygotes were isolated from CD-1 females mated with mCherry[+] males. When the *p53*-KO mCherry[+] embryos reached the pre-compaction morula stage at 2.5 d post coitus, they were treated with Acidic Tyrode's solution to remove the zona pellucida and placed in handmade depressions of the aggregation plate. GFP[+] mESCs were thawed and cultured on MEFs, in serum and LIF conditions, for 2 d and colonies were detached from the plate by a brief (approximately 30 s) trypsinization step. A small colony of 8–15 cells was then picked and added to each depression containing a single *p53*-KO mCherry[+] pre-compaction morula. The aggregates were incubated for 65–70 h and those that successfully formed expanded blastocysts were re-transferred. Post-implantation embryos were isolated according to the indicated embryonic day, with a 24 h developmental delay accommodated due to the re-transfer procedure.

## E6.5 post-implantation embryo collection and preparation for downstream experiments

Deciduae were collected into ice-cold HBSS (Gibco, 14175095), E6.5 embryos were collected into ice-cold M2 medium (Merck, MR-015-D) and tissues were isolated as described previously[8]. Briefly, the embryos were bisected at the embryonic–extraembryonic border, washed in three drops of HBSS and incubated for 15 min at 4 °C in 0,5% trypsin (Thermo Fischer Scientific, 15400054) with 2,5% pancreatin (Sigma-Aldrich, P3292-25G) dissolved in DPBS. The distal exEndo was manually separated from the epiblast by drawing the distal half through a narrow glass capillary. Similarly, the proximal exEndo was manually separated from the extraembryonic ectoderm. The epiblast and the exEndo tissues from individual embryos (generated via two-colour lineage tracing) were collected into RLT Plus Buffer (Qiagen, 1053393) for RNA-seq (see below); pooled proximal or distal exEndo tissues were collected from wild-type CD-1 embryos for WGBS (see below) in Lysis buffer (10 mM Tris–HCl pH 8.0 (Thermo Fisher Scientific, 15568025), 10 mM NaCl (Sigma-Aldrich, S5150-1L), 10 mM EDTA (Thermo Fisher Scientific, 15575020), 0.5% SDS (Thermo Fisher Scientific, AM9822) and 300 μg ml⁻¹ proteinase K (New England Biolabs, P8107S)).

## E9.5 post-implantation embryo collection and preparation for downstream experiments

Deciduae were collected into ice-cold HBSS, E9.5 embryos (somite number, 18–28) were dissected in ice-cold M2 medium, the extraembryonic tissues were completely removed and the yolk sac was kept. For the scRNA-seq analysis (see below), whole lineage-traced embryos were used to determine the cell-type identities of mCherry[+] and dual[+] cells. For assessing extraembryonic cell content in lineage-traced embryos (comparing wild-type and *p53* extraembryonic-specific KO), the embryos were cut in half with a micro knife along the anterior–posterior axis and the posterior half was used for further experiments. For RNA-seq, RRBS and WGBS experiments (see below), wild-type lineage-traced E9.5 embryos were cut in half with a micro knife along the anterior–posterior axis. The midgut was manually isolated from the posterior half using tungsten needles (Fine Science Tools, 10130-10) and the most posterior part, containing the hindgut, was also kept. For each midgut and hindgut replicate, corresponding tissues from four embryos were pooled.

The embryos, isolated tissues and yolk sac were washed in ice-cold HBSS and dissociated with 0.25% trypsin–EDTA (Gibco, 25200056) for 10 min at 37 °C to obtain single cells. This was quenched with KnockOut DMEM (Thermo Fisher Scientific, 10829018) containing 10% FBS (PAN-Biotech, P30-2602) and 0,05 mg ml⁻¹ DNase I (Merck, 11284932001) to dissociate the cells via pipetting, and the cells were also washed once with this buffer. After blocking with normal mouse serum (Invitrogen, 31881) for 5 min on ice, the cells were stained for EPCAM (Alexa Fluor 647 anti-EPCAM; BioLegend, 118212) in FACS buffer (HBSS with 2% FBS and 0,5 mM EDTA (Thermo Fischer Scientific, 15575020)) for 10 min on ice. Specifically for the pooled midgut and hindgut samples, enrichment of EPCAM[+] cells was performed by magnetic separation (MACS) using anti-Cy5/anti-Alexa Fluor 647 MicroBeads (Miltenyi Biotec, 130-091-395), as per the manufacturer's instructions, with the MS columns (Miltenyi Biotec, 130-042-201) and an OctoMACS Separator (Miltenyi Biotec, 130-042-109). Finally, the cells were stained with 0.02% DAPI (Roche Diagnostics, 102362760019) in FACS buffer for 8 min on ice, washed once, resuspended in FACS buffer and kept on ice until flow cytometry analyses or sorting.

## E13.5 post-implantation embryo collection and preparation for downstream experiments

E13.5 embryos were dissected in DMEM/F-12 medium (Thermo Fischer Scientific, 21041025) containing 10% FBS. For scRNA-seq analysis of the wild-type and *p53*-KO embryos (see below), the gastrointestinal tract was isolated, after which dissociation, staining for EPCAM, enrichment by MACS and preparation for FACS was performed as described earlier for E9.5 midgut and hindgut samples. For RNA-seq and RRBS (see below), E13.5 lineage-traced embryos were collected, from which the intestine was isolated and split into the small intestine and colon parts with a micro knife. Dissociation, staining for EPCAM and sample preparation for FACS were then performed as described earlier for E9.5 embryos.

## FACS

A FACS Aria Fusion (BD Biosciences) flow cytometer with an 85 μm nozzle and a BD FACS Diva software were used for all sorting experiments. Dead cells were excluded based on DAPI staining and cells were analysed or sorted based on the signal intensity of EPCAM

(Alexa Fluor 647 anti-EPCAM), GFP and mCherry. For the scRNA-seq analysis (see below) to determine the cell-type identities of mCherry+ and dual+ cells, wild-type lineage-traced whole E9.5 embryos were used and the dual+ cells were sorted into three populations based on the mCherry signal intensities. For the low-input bulk RNA-seq, RRBS and WGBS analysis (see below), gut cells from manually isolated tissues were used from E9.5 wild-type lineage-traced embryos, EPCAM+ dual+ cells were sorted with low mCherry+ signal intensity as embryonic gut control for the mCherry+ extraembryonic gut. (Note that at E9.5, the EPCAM+ GFP-only cell population contains surface ectoderm and primordial germ cells in addition to embryonic gut endoderm cells, thus the GFP-only population is not suitable as an embryonic gut endoderm control[44,45]. As established by our scRNA-seq, dual+ cells are of embryonic origin and EPCAM+ dual+ cells correspond to gut endoderm.) For RNA-seq, RRBS and WGBS analysis of E9.5 YsEndo, cells positive for both EPCAM and mCherry were isolated. For the low-input bulk RNA-seq and RRBS of organs from E13.5 lineage-traced embryos with extraembryonic p53-KO, EPCAM+GFP+ cells were sorted as the embryonic control for mCherry+ extraembryonic cells because at this stage there is no contamination by non-endoderm cell types unlike at E9.5. For the scRNA-seq analysis of wild-type and *p53*-KO gastrointestinal tract, EPCAM+ endodermal cells were sorted. The sorted cells were collected into ice-cold RLT Plus buffer for bulk RNA-seq and RRBS or into Lysis buffer for WGBS, whereas the cells were sorted into ice-cold DPBS for single-cell RNA-seq analysis with MULTI-seq labelling. Flow cytometry data analyses were done using FlowJo (BD, version 10.8.1). Representative gating strategies are provided in Supplementary Information 1.

### Low-input bulk RNA-seq

The protocol for RNA-seq library preparation was adopted from the Smart-RRBS protocol[23] for low-input bulk samples with a few modifications. Briefly, instead of using single cells, 40–50 cells were sorted from the pooled posterior parts (midgut and hindgut) of lineage-traced E9.5 embryos or the corresponding yolk sac tissues into 20 μl RLT Plus buffer. Similarly, 40–50 cells were sorted from individual organs (colon and small intestine) isolated from lineage-traced E13.5 embryos. Individual progenitor tissues (Epi, exEndo 1 and 2) were isolated from E6.5 lineage-traced embryos and each was collected into 20 μl RLT Plus buffer. Poly(A)+ messenger RNA was separated from genomic DNA using beads with Oligo(dT) and reverse transcribed in the presence of a template-switching oligonucleotide; for complementary DNA amplification, the cycle number was reduced to 15–16 cycles. The PCR products were purified using AMPure XP beads (Beckman Coulter, A63881) and normalized to 0.1 ng/μl concentration. Then, the Nextera XT DNA Library Preparation Kit (Illumina #FC-131-1024) was used to obtain the RNA-seq libraries. The quality and concentration of the obtained libraries were measured using High Sensitivity D1000 ScreenTape and reagents (Agilent, 5067-5584 and 5067-5585) on an Agilent 4150 TapeStation. The libraries were then sequenced using 100-base-pair paired-end sequencing on a NovaSeq 6000 platform.

### Low-input bulk RRBS

The protocol for RRBS library preparation was adopted from the Smart-RRBS protocol[23] for low-input bulk samples with a few modifications. Briefly, instead of using single cells, 40–50 cells were sorted from the pooled posterior parts (midgut, hindgut) of lineage-traced E9.5 embryos and the corresponding yolk sac tissues into 20 μl RLT Plus buffer. Similarly, 40–50 cells were sorted from individual organs (colon and small intestine) isolated from lineage-traced E13.5 embryos. The poly(A)+ mRNA-depleted fraction containing genomic DNA was purified using AMPure XP beads. Genomic DNA was digested with both MspI and HaeIII, followed by end repair, A-tailing and ligation of indexed methylated adaptors. After pooling, bisulfate conversion was performed using an EpiTect fast DNA bisulfite kit (Qiagen, 59824).

The cycle number to amplify the bisulfite-converted DNA was reduced to 14–15 and no size fractionation was performed. The quality and concentration of the obtained libraries were measured using High Sensitivity D1000 ScreenTape and reagents on an Agilent 4150 TapeStation. The libraries were then sequenced using 100-base-pair single-end sequencing on a NovaSeq 6000 platform.

### WGBS

Genomic DNA was isolated by phenol–chloroform extraction from wild-type E6.5 exEndo tissues and from cells sorted from the pooled posterior parts (midgut and hindgut) of lineage-traced E9.5 embryos as well as the corresponding yolk sac tissues in Lysis buffer. The genomic DNA was then sheared in micro TUBE AFA Fiber Pre-Slit Snap-Cap tubes (Covaris, 520045), followed by phenol–chloroform extraction. The purified DNA was bisulfite-converted using an EZ DNA methylation-gold kit (Zymo, D5005) and WGBS libraries were processed using an Accel-NGS Methyl-seq DNA library kit (Swift Biosciences, 30096) as per the manufacturer's recommendations. Libraries were prepared with nine (for E6.5 samples) or 12–13 (for E9.5 samples) final PCR cycles and cleaned using AMPure XP beads. The quality and concentration of the obtained libraries were measured using High Sensitivity D5000 ScreenTape and reagents (Agilent, 5067-5592 and 5067-5593) on an Agilent 4150 TapeStation. The libraries were then sequenced using 150 base pair paired-end sequencing on a NovaSeq 6000 platform.

### MULTI-seq labelling

MULTI-seq labelling was performed as previously described[17] with few modifications. Briefly, the FACS-isolated single-cell suspension of each sample in DPBS was incubated with a unique Barcode-Lipid modified oligonucleotide 'anchor' mix (200 nM final concentration each) for 5 min on ice. Next, a 200 nM 'co-anchor' mix was added to each sample and the cells were incubated for an additional 5 min on ice. The reaction was quenched by adding 200 μl of 1% BSA (Sigma-Aldrich, A4503) in PBS and the cell suspensions were then washed twice with 0.4% BSA in PBS. Next, the four samples were pooled in 0.4% BSA in PBS and subjected to the 10x scRNA-seq procedure (described in the following section). The sequences of the MULTI-seq oligonucleotides are listed in Supplementary Table 11.

### ScRNA-seq

The scRNA-seq experiment was performed as previously described[18]. Briefly, the cell suspension after MULTI-seq labelling was filtered using Scienceware Flowmi cell strainers (40 μm), the cell concentration was determined using a haemocytometer and the cells were subjected to scRNA-seq (10x Genomics, Chromium Single Cell 3' v3.1). Single-cell libraries were generated according to the manual, with one modification: fewer PCR cycles were run than recommended during cDNA amplification ($n = 11$) or library generation/sample indexing ($n = 10$) to increase the library complexity. The quality and concentration of the obtained libraries were measured using High Sensitivity D5000 ScreenTape and reagents on an Agilent 4150 TapeStation. The libraries were sequenced with paired-end fragments according to the parameters described in the manual.

### ScRNA-seq and MULTI-seq barcode recovery

The steps described in the previous section were applied to generate the scRNA-seq library of the MULTI-seq sample, with two modifications: (1) during the cDNA amplification step, 1 μl of an oligonucleotide to enrich for the MULTI-seq barcodes was added to the reaction (see Supplementary Table 11 for the oligonucleotide sequence) and (2) after cDNA amplification and incubation with SPRIselect beads, the MULTI-seq barcode-containing supernatant was collected and subjected to further incubation with SPRIselect beads to recover the MULTI-seq barcode as previously described[17]. MULTI-seq barcode

recovery and integrity were measured using High Sensitivity D5000 ScreenTape and reagents on an Agilent 4150 TapeStation. The obtained material was then used as input for MULTI-seq barcode library preparation (described in the following section).

## MULTI-seq barcode library preparation

MULTI-seq barcode libraries were prepared as previously described[17]. Briefly, 15 ng input material obtained from the 10x cDNA purification (described in the previous section) was used to perform library PCR using KAPA HiFi HotStart ReadyMix (Roche, KK2601) in a 50 µl reaction with the following steps: 95 °C for 5 min; ten cycles of 98 °C for 15 s, 60 °C for 30 s and 72 °C for 30 s; 72 °C for 1 min and hold at 4 °C. Next, AMPure XP beads cleanup (1.6×) was performed to purify the MULTI-seq barcode libraries. The quality and concentration of the obtained libraries were measured using High Sensitivity D5000 ScreenTape and reagents on an Agilent 4150 TapeStation. The libraries were then sequenced using asymmetric end sequencing on a NovaSeq 6000 platform.

## Microscopy

Embryos were imaged using a Zeiss Axio Zoom V16 stereo microscope for bright-field and fluorescence microscopy to acquire whole embryo overviews. A weak background signal, presumably autofluorescence, can be seen, which is often noted and due to the limitations of this technique to eliminate out-of-focus light when thick biological specimens are imaged. Embryos were imaged using a Zeiss LSM880 laser scanning microscope with an Airyscan detector or a Zeiss light-sheet LS Z1 microscope to acquire high-resolution images and optical sections. Appropriate filters for GFP, mCherry, DAPI, Alexa Fluor 546, Alexa Fluor 568 and Alexa Fluor 647 were used. To image blastocysts, the fixed embryos were mounted in a DAPI-containing mounting medium (Biozol, VEC-H-1200). For light-sheet microscopy, the specimens were cleared and embedded in 1.5% low-melting agarose (Sigma-Aldrich, A9414) in PBS. Agarose columns containing the samples were inserted into the RIMS-filled acquisition chamber and cleared for an additional 5 h to overnight, depending on the tissue volume. Post-acquisition processing was performed using the ZEN Blue/Black (Zeiss) and ImageJ software packages.

## Whole-mount immunofluorescence and tissue clearing

If not specified otherwise, incubation in buffers was performed at room temperature on a roller. Embryos selected for immunofluorescence were collected in 4 ml glass vials (Wheaton 224882), fixed in 4% paraformaldehyde for 1 h and then washed with PBS (3 × 10 min) and PBST (PBS containing 0.5% Triton X-100; Merck, 9002-93-1) at room temperature (3 × 10 min). For blocking, the embryos were incubated in PBSTB (PBST containing 10% FBS) at 4 °C for a minimum of 24 h. Primary antibody incubation was performed in PBSTB at 4 °C for 48–96 h (the antibodies are listed in Supplementary Table 12). After incubation, the remaining antibody solution was diluted by rinsing the samples 3× with PBSTB, followed by washing with PBSTB (3 × 10 min) and PBST (3 × 10 min). After washing, the specimens were incubated overnight in PBSTB at 4 °C. Secondary antibody incubation was performed in PBSTB at 4 °C for 24–48 h. The embryos were rinsed 3× with PBSTB and washed with PBSTB + 0.02% DAPI (2 × 20 min), followed by PBST + 0.02% DAPI (3 × 20 min) and transferred to eight-well glass-bottomed slides (Ibidi, 80827). After additional washing steps in PBS (3 × 10 min), the embryos were either imaged or processed for tissue clearing. For tissue clearing, stained embryos on eight-well glass slides were incubated in 0.02 M phosphate buffer (0.005 M $NaH_2PO_4$ and 0.015 M $Na_2HPO_4$, pH 7.4; 3 × 5 min). Before clearing, fresh refractive index matching solution (RIMS, 133% w/v Histodenz; Sigma-Aldrich, D2158) in 0.02 M phosphate buffer was prepared and applied to the samples after careful removal of the phosphate buffer. The clearing was performed at 4 °C on a shaking incubator for at least 24 h.

## Ex utero culture of embryos and live-cell imaging

E7.5 embryos were dissected in equilibrated (5% $CO_2$) and pre-warmed (37 °C) M2 medium and then transferred to eight-well glass-bottomed slides with a 10 µl drop of Matrigel (Corning, 356231) to position the embryo. Next, equilibrated and pre-warmed culture medium was added (50% rat serum (Janvier labs, Sprague Dawley rat serum) and 50% DMEM-F-12 medium). The embryos were cultured and imaged at 37 °C with 5% $CO_2$ using a Zeiss LSM880 confocal laser scanning microscope. LysoTracker deep red (Invitrogen, L12492) was used at a concentration of 1 µM in the embryo culture media.

## In vitro culture of sorted gut cells

Wild-type lineage-traced embryos or lineage-traced embryos with extraembryonic *p53* KO were isolated at E9.5. The embryos were cut in half along the anterior–posterior axis with a micro knife and the posterior half, including the midgut and hindgut region, was used further. A single-cell suspension was prepared for FACS as described earlier. EPCAM+ exGut (mCherry+) and EPCAM+ emGut (utilizing the dual+ cells) cells were sorted into gut culture medium: 75% IMDM medium (Invitrogen, 21056-023), 25% F-12 (Invitrogen, 11765-054), 0.5×B27 (Invitrogen, 12587-010), 0.5×N2 (Invitrogen, 17502-048), 0.5 mM ascorbic acid (Sigma-Aldrich, A4403), 0.05% BSA, 2 mM GlutaMAX (Thermo Fischer Scientific, 35050038), 1×penicillin–streptomycin (Thermo Fisher Scientific, 15140122) and 55 µM 2-mercaptoethanol (Thermo Fisher Scientific, 21985023) supplemented with 20 µg µl⁻¹ FGF-2 (PeproTech, 100-18C), 2 µM CHIR99021 (Merck, 361571) and 10 µM Rho kinase inhibitor (Abcam, ab120129). Based on FACS cell counts, 300 cells were transferred into single wells of an ultra-low-attachment 96-well plate (Costar, 7007). The cells formed aggregates on the first day after plating and the medium was changed every day, until day 5, with gut culture medium supplemented with 2% Matrigel.

## Morphometric analysis of gut-cell assemblies

The processing and analysis for gut-cell assembly morphometries were carried out in ImageJ/Fiji. Briefly, with a semi-automated routine, the outlines of the assemblies were manually drawn from maximum-intensity-projection bright-field images and the subsequent area was calculated by the macro. The obtained values were subsequently used for plotting.

## ScRNA-seq processing

Raw reads (FASTQ) were generated using 10x Genomics Cell Ranger mkfastq (version 6.0.1; default parameters)[46]. The FASTQ files were aligned to the mouse reference genome (mm10 including GFP and mCherry transgenes), filtered and the unique molecular identifiers were counted using 10x Genomics Cell Ranger count (version 6.0.1; default parameters). The resulting cell–barcode matrix was loaded into R and converted to a Seurat object (R package Seurat version 4.1.0)[47]. Cells with ≥15% reads mapping to the mitochondrial genome and <2,000 genes detected were removed. Demultiplexing based on MULTI-Seq barcodes was performed using the R package deMUL-TIplex according to the tutorial (version 1.0.2; https://github.com/chris-mcginnis-ucsf/MULTI-seq; two to four rounds of quantile sweeps for sample classification were performed)[17]. All cells with a valid MULTI-Seq barcode and classified as singlets after the sample classification were considered for downstream analyses.

For the re-processing of E8.75 gut endoderm samples from Nowotschin et al.[4], raw FASTQ files were obtained from GSE123046 (E8.75 GFP+ and GFP−), processed with 10x Genomics Cell Ranger mkfastq and counted as described above. For the E9.5–E15.5 gastrointestinal tract cohort from Zhao et al.[19], processed data and metadata were obtained from GSE186525.

## RNA-seq processing

Raw reads were subjected to adaptor and quality trimming with cutadapt (version 4.1; parameters: --quality-cutoff 20 --overlap

5 --minimum-length 25 --interleaved --adaptor AGATCGGAAGAGC -A AGATCGGAAGAGC), followed by poly-A trimming with cutadapt (parameters: --interleaved --overlap 20 --minimum-length --adaptor 'A[100]' --adaptor 'T[100]')[48]. The reads were aligned to the mouse reference genome (mm10 including GFP and mCherry transgenes) using STAR (version 2.7.9a; parameters: --runMode alignReads --chimSegmentMin 20 --outSAMstrandField intronMotif --quantMode GeneCounts)[49] and transcripts were quantified using stringtie (version 2.0.6; parameters: -e)[50] with the GENCODE annotation (release VM23). To generate RNA-seq coverage tracks (CPM) per tissue and time point, single replicate BAM files were merged using the samtools (version 1.18) 'merge' command[51]. Coverage tracks were subsequently generated using the deepTools (version 3.5.2) 'bamCoverage' command[52].

## RRBS processing
Raw reads were subjected to adaptor and quality trimming using cutadapt (version 4.1; parameters: --quality-cutoff 20 --overlap 5 --minimum-length 25; Illumina TruSeq adaptor). The trimmed reads were aligned to the mouse genome (mm10 including GFP and mCherry transgenes) using BSMAP (version 2.90; parameters: -v 0.1 -s 12 -q 20 -w 100 -S 1 -u -R -D C-CGG)[53]. Methylation rates were called using mcall from the MOABS package (version 1.3.2; default parameters)[54]. Due to the overall low coverage of low-input RRBS replicates, here the replicates were combined at the raw count level of methylated and unmethylated CpGs and merged methylation rates were calculated subsequently to increase the coverage. Only CpGs covered by at least five (single RRBS replicates) or ten (merged RRBS replicates) and at maximum 150 reads were considered for downstream analyses.

## WGBS processing
Raw reads were subjected to adaptor and quality trimming using cutadapt (version 4.1; parameters: --quality-cutoff 20 --overlap 5 --minimum-length 25; Illumina TruSeq adaptor clipped from both reads), followed by trimming of ten and five nucleotides from the 5′ and 3′ ends, respectively, of the first read and 15 and five nucleotides from the 5′ and 3′ ends, respectively, of the second read. The trimmed reads were aligned to the mouse genome (mm10 including GFP and mCherry transgenes) using BSMAP (version 2.90; parameters: -v 0.1 -s 16 -q 20 -w 100 -S 1 -u -R). Duplicates were removed using the 'MarkDuplicates' command from GATK (version 4.3.0.0; parameters: --VALIDATION_STRINGENCY = LENIENT --REMOVE_DUPLICATES = true)[55]. Methylation rates were called using mcall from the MOABS package (version 1.3.2; default parameters). Only CpGs covered by at least ten and at most 150 reads were considered for downstream analyses, with the exception of genome browser tracks where a minimum of five reads per CpG was used.

## Bioinformatic analysis
All analyses were carried out using R 4.1.0 unless stated otherwise.

## Cell-state annotation
**E9.5.** Cell states present in E9.5 dual[+] cell populations were defined using the R package Seurat (version 4.1.0). Due to the sex bias present in our complete scRNA-seq dataset (dual[+] cells are always male due to the ESC line used for the aggregation; mCherry[+] cells can be male or female and were later integrated with the dual[+] cells), genes located on the Y chromosome were excluded. Across the dual[+] cells, the 2,000 most variable genes were detected across all cells and sort populations (dual[+] low, intermediate and high). Gene counts were log₂-normalized ('NormalizeData') and scaled using the function 'ScaleData' accounting for cell-cycle and mitochondrial expression effects (parameters: vars. to.regress = c('percent.mt', 'S.Score', 'G2M.Score')). A UMAP was used to represent the cells in two dimensions using the function 'RunUMAP' (parameters: reduction = 'pca', dims = 1:21) based on the PCA ('RunPCA', number of principal components used as input for the

UMAP determined by manually inspecting the corresponding elbow plot). Seven clusters of cells were identified using the functions 'FindNeighbors' (parameters: reduction = 'pca', dims = 1:10) and 'FindClusters' (parameters: resolution = 0.2). Marker genes per cluster were identified with the 'FindAllMarkers' function (parameters: only. pos = TRUE, min.pct = 0.1, logfc.threshold = 0.1). Based on the detected marker genes per cluster and literature-based markers of cell states we assigned clusters to the following cell states: hindgut[5], colon[19], small intestine[19], liver[4], foregut[4], mesoderm[18] and endothelium[18]. The mCherry[+] cells were then integrated with the dual[+] reference dataset and assigned to the nearest cell state ('FindTransferAnchors' and 'MapQuery').

**E13.5.** Cell states present in the E13.5 wild-type gastrointestinal tract were defined similar to our E9.5 annotation with minor deviations. After manual inspection of the elbow plot, 20 principal components were used as input for the 'RunUMAP' function. Nine clusters were identified and annotated based on the detected marker genes as colon distal[19], colon proximal[19], small intestine distal[19], small intestine proximal[19], stomach antrum[19], stomach corpus[19] and stomach fore[19] as well as pancreas tip[31] and trunk[31]. *p53*-KO cells were then integrated with the wild-type reference dataset and assigned to the nearest state as described earlier. As this experiment is not based on the lineage-traced embryos, cells were split by lineage origin based on the expression of *Rhox5* and *Trap1a*, two known marker genes for gut endoderm cells with extraembryonic origin. Cells with detected expression of both genes were considered as exGut, whereas cells with no detected expression of both genes were considered emGut. Cells with expression of only one of the two genes were discarded for downstream analysis.

**Public data.** Cell-state annotations for the E8.75 gut endoderm samples from Nowotschin et al.[4] as well as the E9.5 to E15.5 gastrointestinal tract samples from Zhao et al.[19] were obtained from the respective provided metadata. For the E8.75 dataset, only cells assigned to endodermal cell states (colon, liver, lung, pancreas, small intestine, thymus and thyroid) were considered. For the E9.5–E15.5 dataset, only the cell states associated with the large and small intestinal epithelium were considered for Fig. 2g; all cells except those assigned to 'Unknown of E9.5' were considered for Extended Data Fig. 4e. Cells were split by lineage origin as described earlier for the E13.5 data.

## Differential gene expression
Genes that were differentially expressed in E9.5 exGut (test) compared with emGut (control) cells were determined using DESeq2 (version 1.32.0, parameters: minReplicatesForReplace = 10)[56]. As explained in the 'FACS' section, dual[+] cells were used as the emGut control due to the contamination of E9.5 GFP[+]EPCAM[+] cells with surface ectoderm and primordial germ cells[44,45], which would have a strong effect on gene expression analysis. Dual[+] cells have been shown by the scRNA-seq analysis to be of embryonic, mostly endoderm, origin with extraembryonic remnants. To reduce the contamination of extraembryonic fragments, only dual[+] cells with low mCherry signal were sorted ('FACS' section). Although the dual[+] cells can contain some extraembryonic transcript remnants, this will only weaken the actual differential signal between exGut and emGut cells and not introduce unrelated, cell-type-specific signatures such as present in the single GFP[+]EPCAM[+] population.

Only genes with TPM > 1 in at least four of 16 samples (four biological replicates for exMidgut, exHindgut, emMidgut and emHindgut each) were used as input for the analysis. Genes located on the Y chromosome were removed and the analysis was restricted to protein-coding genes. Genes with log₂(fold change) > 1 and an adjusted $P < 0.05$ were termed E9.5 exGut high, whereas genes with log₂(fold change) < −1 and an adjusted $P < 0.05$ were termed E9.5 exGut low. Overrepresentation analysis of differentially expressed genes in the Gene Ontology term database for biological processes and cellular

components was carried out using the R package (and function) Web-GestaltR (version 0.4.4; parameters: minNum = 10, maxNum = 500, sigMethod = 'top' and topThr = 5)[57]. Genes that were classified as E9.5 exGut low and associated with the axon-related biological processes or the cellular components related to synaptic membranes detected by the overrepresentation analyses were termed 'axonogenesis-associated'. Genes that were classified as E9.5 exGut high and associated with the germ cell-related biological processes or the synaptonemal complex and condensed chromosome-related cellular components were termed 'germline-associated'. This list was complemented by exGut high genes that have been reported to be specifically expressed in the germline (*Rhox5*, ref. [58]; *Trap1a*[59]; *Mageb16*, ref. [60]; *Tekt5*, ref. [61]; *Tex101*, ref. [62]; *Xlr3c*[63] and *Slc25a31*, ref. [64]). Heatmaps of log$_2$-transformed TPMs averaged across replicates per tissue were generated using the R package ComplexHeatmap (version 2.7.11)[65]. Genes known to escape X chromosome inactivation were obtained from Marks et al. and overlapped with the exGut high genes to exclude a potential effect of the sex bias between emGut and exGut cells on our differential expression analysis due to incomplete X chromosome inactivation in females[66]. *Z*-score-transformed differentially expressed genes were clustered using all E9.5 gut and E13.5 intestine samples (averaged per cell type and time point) using *k*-means (parameters: iter.max = 1,000, nstart = 100).

### Definition of DNA methylation-sensitive genes
DNA methylation-sensitive genes were defined based on previous work that studied the effects of the KO of the DNA methyltransferase *Dnmt3b*[28] and both de novo methyltransferases together (*Dnmt3b* and *Dnmt3a*)[29] in E8.5 embryos. These studies reported upregulation of different gene groups following the KOs: (1) genes that were methylated in wild-type embryos and lost methylation after DNA methyltransferase(s) KO accompanied by an increase in expression (germline genes were reported to be associated with this gene group) and (2) genes that were upregulated with no direct link to DNA methylation, which were mostly considered to represent secondary effects[28,29]. DNA methylation-sensitive genes for each study were therefore determined based on gene expression and DNA methylation analyses from the respective studies as follows.

For *Dnmt3b* KO, genes that were upregulated with log$_2$(fold change) > 1, an adjusted $P$ < 0.05 and contained an exon that overlapped with a hypomethylated differentially methylated region as defined by Auclair et al. were selected[28].

For double KO, genes that were upregulated with a log$_2$(fold change) > 1, an adjusted $P$ < 0.05 and a highly methylated CpG-rich promoter in wild-type embryos ('group 3' according to the classification by Dahlet et al.) were selected[29].

The overall set of DNA methylation-sensitive genes was defined by the union of *Dnmt3b*- and double KO-specific DNA methylation-sensitive genes.

### Genomic feature annotation
We generated 1 kb genomic tiles by segmenting the genome using bedtools makewindows (version 2.30.0; parameters: -w 1000 -s 1000)[67]. Promoters were defined as 500 bp upstream and 500 bp downstream of the transcription start site. Genes were defined to have a promoter CGI if 20% of a CGI or 20% of the promoter overlapped (bedtools intersect; version 2.30.0; parameters: -f 0.2, -F 0.2, -e).

Hyper CGIs were defined using the methylation difference of mouse epiblast and exEndo 1 (WGBS data). The CGIs were termed hyper CGIs if the difference of the average methylation of a CGI was more than 0.1 when comparing averaged exEndo 1 replicates to averaged epiblast replicates. In addition, either more than half of the CpGs within a CGI were required to have a minimum difference of 0.1 or the CGI was required to contain a differentially methylated region with higher methylation in exEndo 1. Differentially methylated regions were called based on CpGs located in CGIs using metilene

(version 0.2–8; parameters: -m 10 -d 0.1 -c 2 -f 1 -M 80 -v 0.7) and filtered for $Q$-value < 0.05 (ref. [68]). CGIs methylated in the epiblast (≥0.15) were excluded from the set.

### Genome-wide DNA methylation analysis
E6.5 WGBS replicates were averaged per CpG for each tissue (two replicates of each E6.5 epiblast, exEndo 1 and exEndo 2). For both WGBS and RRBS samples, methylation rates were averaged per genomic tile and CGI. Only features located on autosomes and with at least three covered CpGs were considered for violin plots displaying 1 kb tiles and hyper CGIs. Genome browser tracks displaying CpG methylation rates were generated using IGV (version 2.15.2)[69].

### Promoter DNA methylation analysis
The promoter methylation level of differentially expressed genes was determined as follows. If the promoter of a gene overlapped with a CGI (or multiple) covered by WGBS or RRBS samples, the average methylation value of the CGI(s) was used as the promoter methylation level. If no CGI overlapped the transcription start site, the average methylation in the promoter region was considered as the promoter methylation level ('Genomic feature annotation' section). Promoters were subdivided into low, intermediate and high CpG density promoters (LCP, ICP and HCP, respectively) using the following criteria. Every promoter that did not overlap with a CGI (covered by WGBS or RRBS) and for which the complete promoter region was used to determine the methylation level was considered a low CpG density promoter. If one or more CGIs overlapped the promoter region and were used to determine the promoter methylation level, the promoter was classified as ICP or HCP depending on the observed-to-expected (O/E) CpG ratio as obtained from the UCSC Genome Browser (that is, HCP, O/E ratio > 0.8; ICP, otherwise). In addition, promoters were divided into E6.5 exEndo 1 hyper- and hypomethylated based on the difference to the epiblast (hypermethylated, delta methylation exEndo 1 versus epiblast > 0.1; hypomethylated, delta methylation exEndo 1 versus epiblast < −0.1).

Many genes classified as E9.5 exGut high are located on the X chromosome (29/156 genes). This affects the methylation level of X chromosome-specific regions in females due to the inactivation and full methylation of one of the two X chromosome copies[70]. Due to our aggregation method, the E9.5 emGut cells are always male (dependent on the ESC line used for aggregation), whereas the E9.5 exGut and YsEndo with extraembryonic origin can be male or female. In addition, our E6.5 WGBS samples were obtained from pooled naturally mated embryos, which also can be male or female. Therefore, the promoter methylation levels of E6.5 epiblast, exEndo 1 and exEndo 2 as well as E9.5 exGut and YsEndo samples can be biased towards higher methylation levels by sequencing reads that stem from inactivated X chromosomes in female embryos. However, the promoter DNA methylation levels of DNA methylation-sensitive E9.5 exGut high genes (Fig. 4d) are drastically different between E9.5 emGut and exGut even given this potential bias: emGut (male) cells are almost completely methylated, whereas exGut (male and/or female) cells display low-to-intermediate methylation levels (on average half of the methylation level observed in emGut cells).

### Statistics and reproducibility
No statistical methods were used to pre-determine sample sizes but our sample sizes are similar to those reported in previous publications[4,10,18,28]. The sample sizes are indicated in the figure panels or legends. Before downstream analysis and experiments, resorping embryos were excluded. For the downstream experiments with the two-colour lineage tracing, only embryos with gut-specific mCherry signal were used, mCherry[+]-only embryos were excluded. No other data were excluded. For the RNA-seq and RRBS experiments, three or four replicates were generated. Embryos were pooled for E9.5, whereas individual embryo replicates were generated for E6.5 and E13.5. For the

WGBS experiments, two replicates were generated for each E6.5 tissue (exEndo 1 and 2) and one replicate was generated for each E9.5 tissue. For the E9.5 scRNA-seq analysis, one experiment using cells of different sort groups (dual[+] low, intermediate and high populations, mCherry[+] population) from 15 pooled embryos was performed. For the E13.5 scRNA-seq analysis, four wild-type embryos and four *p53*-KO embryos were included in the experimental set-up labelled by MULTI-seq barcodes, which allowed comparison of cell-state distributions across individual embryo replicates. For imaging experiments and FACS analysis, 3–10 embryos were analysed (the exact number is indicated in the respective figure or legend). All attempts at replication were successful. For assessing the outcome of the complementation assays, embryos were collected without a preconceived selection strategy or prioritization by morphology. Our genomic analyses were independent of human intervention and each sample was analysed equally in an unbiased fashion. The investigators were not blinded to the conditions of the experiments during data collection and analysis. All statistical tests were two-sided and were chosen as appropriate for data distribution.

### Reporting summary

Further information on research design is available in the Nature Portfolio Reporting Summary linked to this article.

## Data availability

Sequencing data that support the findings of this study have been deposited in the Gene Expression Omnibus (GEO) under the accession code GSE250084. Previously published scRNA-seq datasets of E8.75 gut endoderm and E9.5–E15.5 gastrointestinal tract that were re-analysed here were obtained from GSE123046 and GSE186525, respectively. The WGBS datasets of wild-type E6.5 epiblast were obtained from GSE137337. The mouse reference genome mm10 was obtained from UCSC (https://hgdownload.soe.ucsc.edu/goldenPath/mm10/bigZips/). Annotations of CpG islands for mm10 were downloaded from UCSC (https://genome.ucsc.edu/cgi-bin/hgTables). The mm10 gene annotation was downloaded from GENCODE (VM23, https://www.gencodegenes.org/mouse/release_M23.html). Source data are provided at https://doi.org/10.5281/zenodo.10926934 (ref. 71). All other data supporting the findings of this study are available from the corresponding author on reasonable request.

## Code availability

Custom code is available at https://doi.org/10.5281/zenodo.10926934 (ref. 71).

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

## Acknowledgements

We thank all of the members of the Meissner laboratory for their support. We thank Z. D. Smith for critical reading of the manuscript,

and H. Kretzmer, A. Taguchi, A. L. Mattei, P. E. Andersen, M. M. R. Pinto and S. Zsoter for discussions. We thank the MPIMG Animal Facility and Transgenic Unit for their support—in particular K. Macura, A. Landsberger, J. Fiedler and D. Micic—as well as the MPIMG Sequencing Core Facility—in particular S. Klages, N. Mages and S. Paturej. We also thank C. Giesecke-Thiel and the MPIMG Flow Cytometry Facility, in particular M. Piedavent-Salomon, for support with sorting experiments, and T. Mielke and the MPIMG Microscopy Service Group for assistance with imaging. This work received funding from the Max Planck Society (A.M.) and a EMBO long-term fellowship (J.B., ALTF 25-2020). Schematic illustrations were created with BioRender.com.

## Author contributions

J.B., S.H. and A.M. conceived and designed the study. J.B., S.H. and A.M. prepared the manuscript with assistance from the other authors. J.B. performed post-implantation embryo experiments, with assistance from M.W., and also carried out microscopy, flow cytometry analysis, RNA-seq, RRBS, WGBS and in vitro culture experiments. S.H. performed the initial processing of RNA-seq, RRBS, WGBS and scRNA-seq samples as well as downstream analyses of all sequencing datasets. D.S. performed confocal laser scanning microscopy, light-sheet microscopy and image analysis, supervised by B.G.H. D.S. and J.B. performed live imaging. A.B. and J.B. performed scRNA-seq with MULTI-seq labelling, which was also advised and interpreted by S.G. J.B. designed the linage-tracing strategy and generated GFP+ mESCs. A.B. generated the mCherry+ mouse line. L.W. performed and supervised the pre-implantation embryo experiments with advice from J.B. J.B., S.H. and A.M. interpreted the data, and A.M. supervised the work.

## Funding

## Competing interests

A.M. is an inventor on a patent (US20200109456A1) related to hypermethylated CGI targets in cancer. A.M. is a co-founder and scientific advisor of Harbinger Health. The remaining authors declare no competing interests.

## Additional information

**Extended data** is available for this paper at https://doi.org/10.1038/s41556-024-01431-w.

**Correspondence and requests for materials** should be addressed to Alexander Meissner.

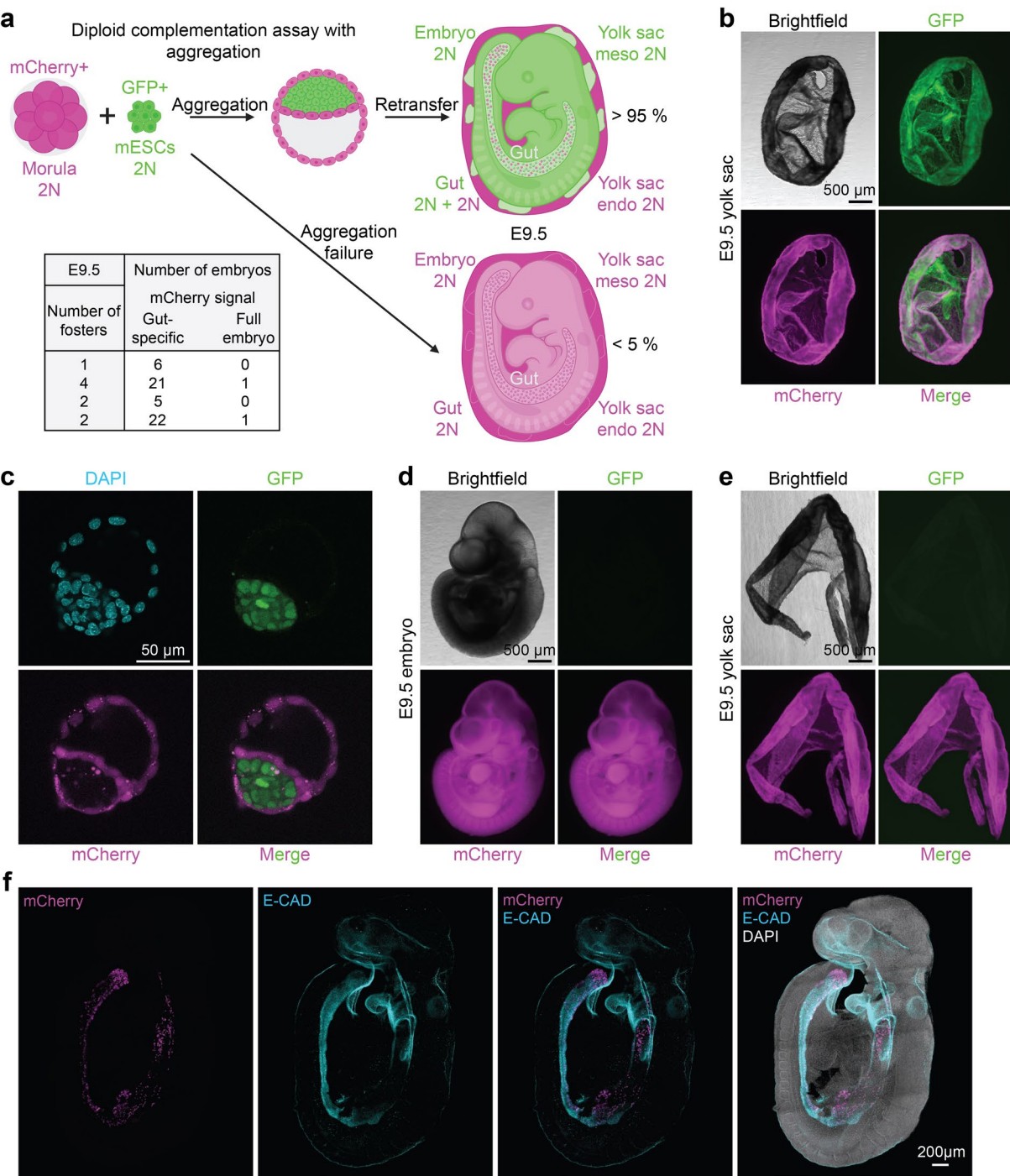

**Extended Data Fig. 1 | Two-colour lineage tracing by stage-specific diploid complementation. a)** Schematic illustrating the two-colour lineage tracing strategy used throughout the study to selectively and stably label embryonic and extraembryonic lineages. An mCherry+ pre-compaction morula is aggregated with a GFP+ ESC colony (2N indicates that both are diploid). Representative experiments at E9.5 are summarized in the table, where in the majority of the cases (more than 95%, 54/56), embryos were GFP+, and only the gut region contained diploid mCherry+ cells, while in rare cases, embryos were fully mCherry+ without detectable GFP+ cells (less than 5%, 2/56). **b)** Bright-field and fluorescence microscopy images of a yolk sac (corresponding to the embryo shown in Fig. 1a) generated via the two-colour lineage tracing (n = 54, one representative yolk sac is shown). **c)** Confocal laser scanning microscopy images showing an expanded blastocyst, where the GFP signal is present in the region indicating the early epiblast, while the mCherry signal is present in the region indicating the extraembryonic lineages. Nuclei were stained with DAPI

(n = 10, one representative embryo is shown). **d)** Bright-field and fluorescence microscopy images of an E9.5 embryo, which contains only diploid mCherry+ cells (n = 2, one representative embryo is shown) as a likely outcome of failed aggregation and mESC incorporation. Such fully mCherry+ embryos were excluded from further experiments. **e)** Bright-field and fluorescence microscopy images of a yolk sac (corresponding to the embryo shown in Extended Data Fig. 1d), which contains only diploid mCherry+ cells as a likely outcome of failed aggregation and mESC incorporation (n = 2, one representative yolk sac is shown). **f)** Maximum-intensity projection of optical sections acquired by confocal laser scanning microscopy showing a lineage-traced E9.5 embryo and confirming the presence of mCherry+ extraembryonic cells specifically in the gut tube. E-CADHERIN, a surface marker of epithelial cells, is present not only in the gut endoderm but also in the surface ectoderm, where no mCherry+ cells are located. Nuclei were stained with DAPI, and immunofluorescence was used for mCherry and E-CAD (n = 3, one representative embryo is shown).

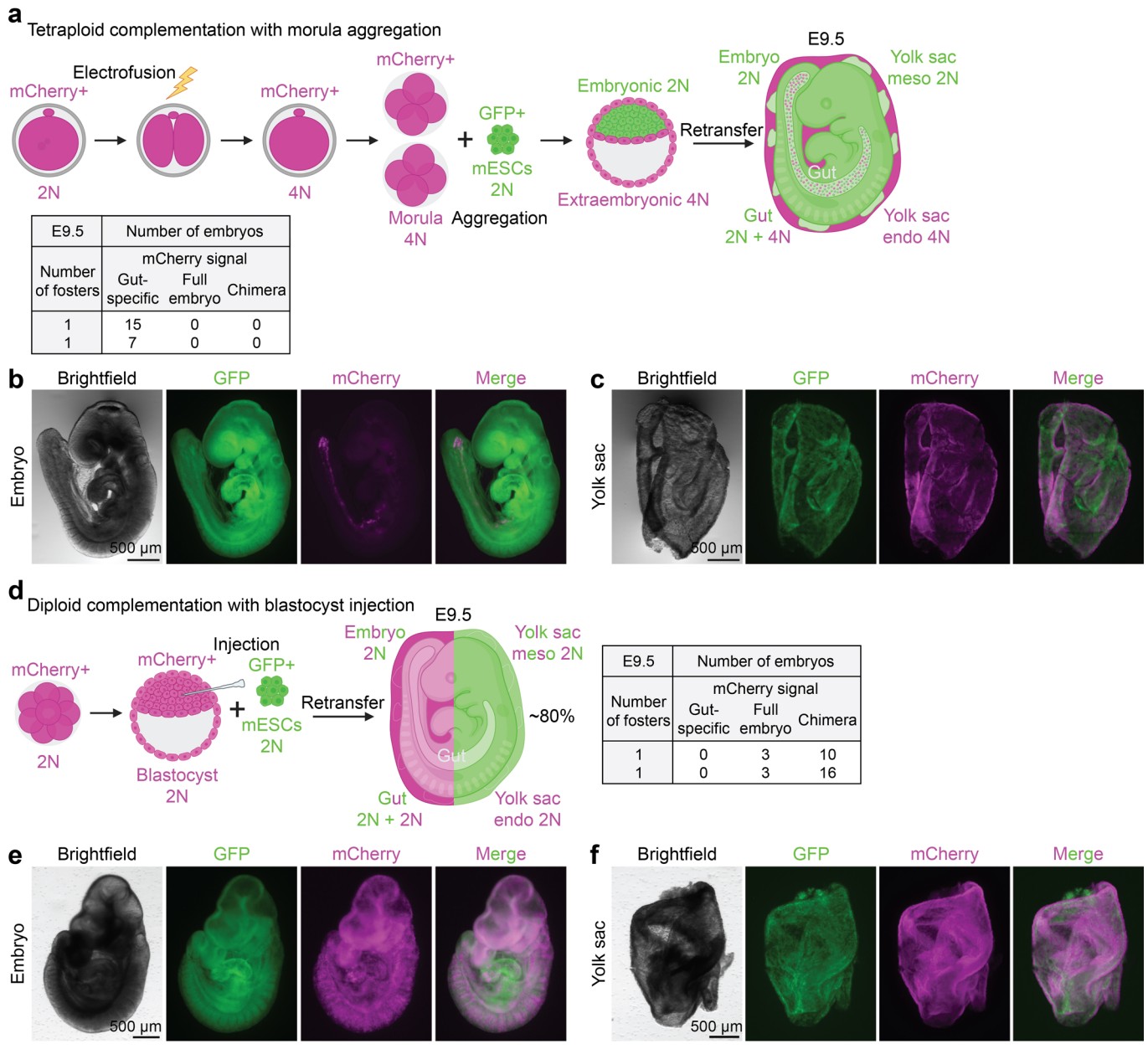

**Extended Data Fig. 2 | Benchmarking different complementation strategies.**
**a**) Schematic illustrating the conventional tetraploid complementation by morula aggregation, with selective and stable lineage labelling (similar to the two-colour lineage tracing strategy used throughout the study) with the caveat that extraembryonic cells are tetraploid. 2-cell-stage mCherry+ embryos are electrofused. Then, two tetraploid mCherry+ pre-compaction morulas are aggregated with a GFP+ mESC colony (2N indicates diploid, 4N indicates tetraploid). At E9.5, the embryo overall consists of diploid GFP+ cells, where only the gut contains tetraploid mCherry+ extraembryonic cells. At E9.5, embryos without visible malformations were collected and counted: 1) mCherry signal specific to the region resembling the gut tube or 2) fully mCherry+ embryos or 3) chimaeras where both mCherry+ and GFP+ cells are broadly distributed (data provided in the table). **b**) Bright-field and fluorescence microscopy images of an E9.5 embryo generated via tetraploid complementation with morula aggregation, which is overall diploid and GFP+, while tetraploid mCherry+ extraembryonic cells are present only in a distinct area resembling the gut tube

($n = 22$, one representative embryo is shown). **c**) Bright-field and fluorescence microscopy images of a yolk sac corresponding to the embryo shown in **b** ($n = 22$, one representative yolk sac is shown). **d**) Schematic illustrating the diploid complementation by blastocyst injection, which leads to fully chimeric embryos because embryonic and extraembryonic lineages are not distinctly labelled (in contrast to the two-colour lineage tracing strategy used throughout the study). GFP+ mESCs are injected into blastocyst-stage mCherry+ embryo (2N indicates diploid). As the blastocyst already has a defined ICM, mCherry+ cells contribute to the embryo proper as well, resulting in chimeras. At E9.5, embryos without visible malformations were collected and counted as described in **a** (data provided in the table). **e**) Bright-field and fluorescence microscopy images of an E9.5 embryo generated via diploid complementation by blastocyst injection ($n = 26/32$, one representative chimeric embryo is shown). **f**) Bright-field and fluorescence microscopy images of a yolk sac corresponding to the embryo shown in **e** ($n = 26/32$, one representative yolk sac is shown).

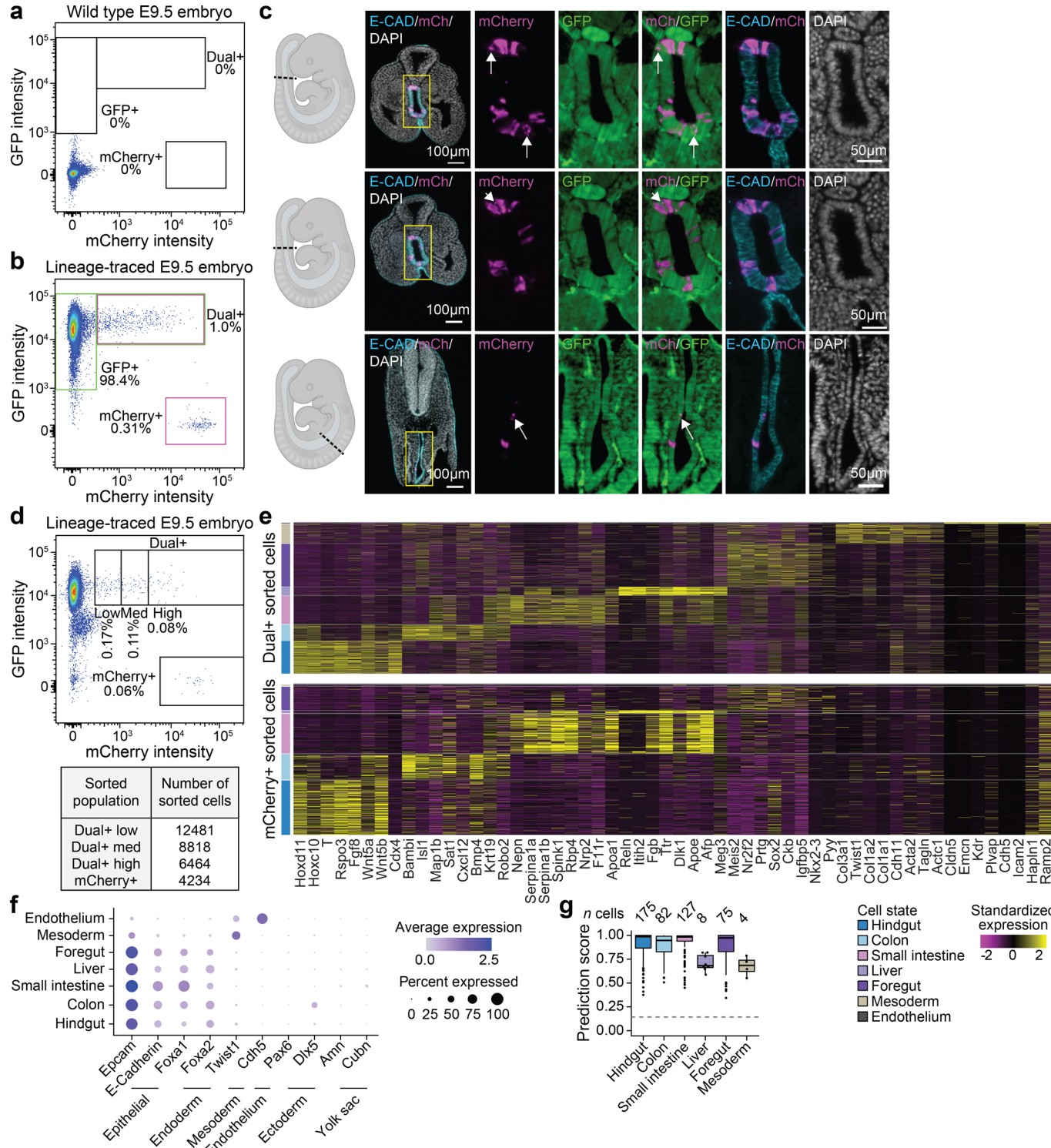

**Extended Data Fig. 3 | Cell type identity of dual+ and mCherry+ cells. a)** Flow cytometry dot plot depicting mCherry and GFP intensities in a single E9.5 wild type embryo used as a negative control in comparison to the E9.5 lineage-traced embryo (Extended Data Fig. 3b). Three populations and their abundance are indicated. **b)** Flow cytometry dot plot depicting mCherry and GFP intensities in a single E9.5 lineage-traced embryo (*n* = 9). Three populations and their abundance are indicated: single GFP+, single mCherry+ and dual+ with both GFP and mCherry signals. **c)** Transversal optical sections as in Fig. 1e for additional axial positions, depicted by the dashed lines in the schematics (*n* = 3, sections from one representative embryo are shown). **d)** Flow cytometry dot plot depicting mCherry and GFP intensities in pooled E9.5 lineage-traced embryos (*n* = 15). Four sorted populations and their abundance are indicated: single mCherry+ cells and dual+ cells with high GFP level plus low, medium, or high mCherry

levels. Absolute cell numbers of sorted cells are summarized in the table; each population was labelled with distinct MULTI-Seq surface barcodes. **e)** Heatmap representation of the standardized, log-normalized expression levels of marker genes of cell states in dual+ and mCherry+ cells. **f)** Average log-normalized expression of lineage marker genes across cells of each cell state (dual+ and mCherry+ combined). **g)** Boxplot of prediction scores for mCherry+ cells as they are assigned to their respective cell state based on the local neighbourhood of dual+ cells. Overall, mCherry+ cells are assigned with a high probability to their respective cell state compared to others, and scores are more confident for endodermal cell state assignments. The dashed line denotes the prediction score that reflects equal association with any of the seven cell states (random assignment). Lines denote the median, edges denote the IQR, whiskers denote 1.5 × IQR, and outliers are represented by dots (*n* = 15 biological replicates).

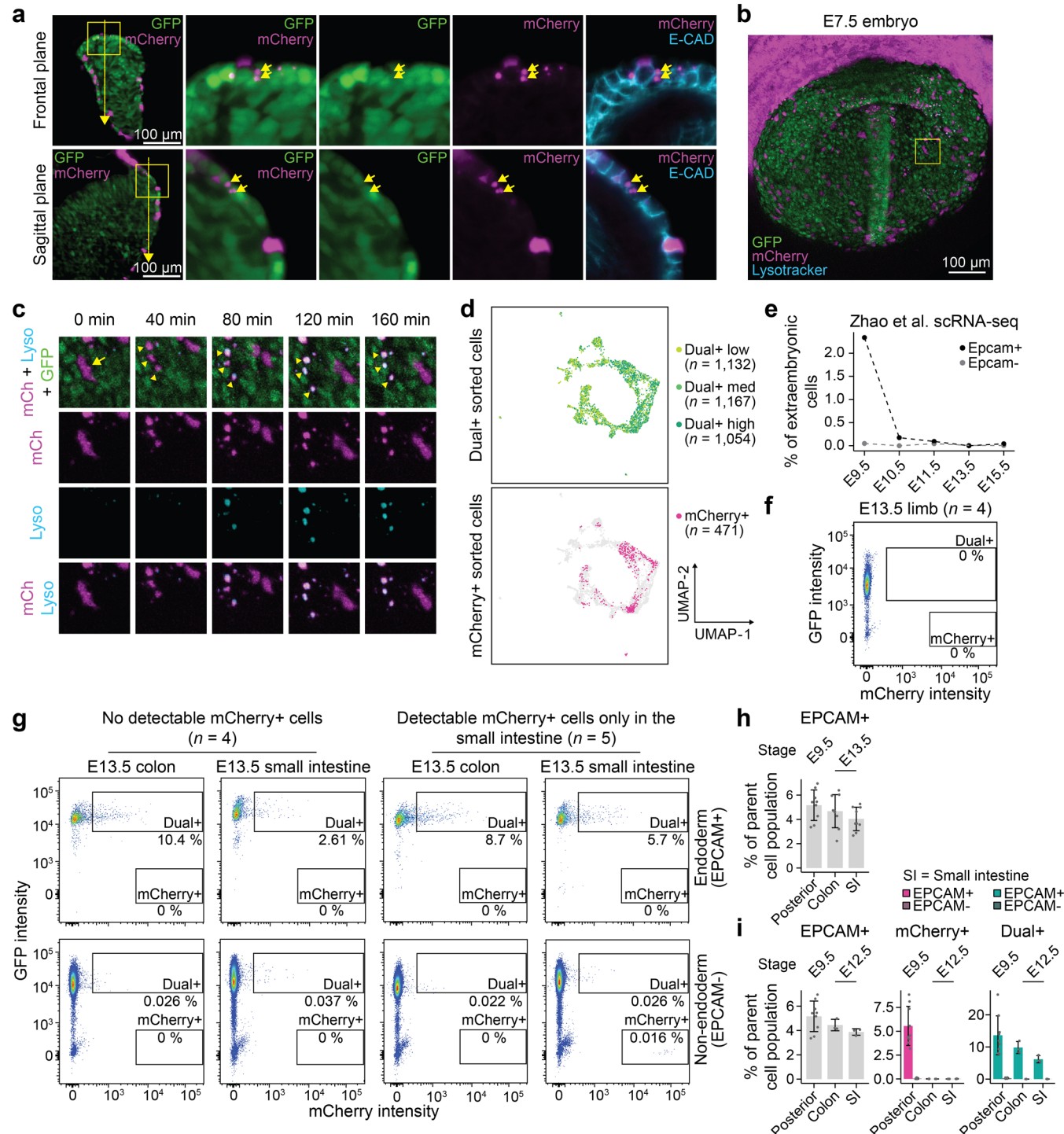

**Extended Data Fig. 4 | Fragmenting mCherry+ cells are phagocytosed and cleared from intestinal organs. a**) Frontal (top) and sagittal (bottom) optical sections of an E7.5 embryo acquired by light sheet microscopy, with immunofluorescence for E-CADHERIN and mCherry (*n* = 3, one representative embryo is shown). The yellow boxes highlight the zoomed-in images, and the yellow lines indicate the position of the sagittal transection in the frontal plane and the frontal transection in the sagittal plane. The yellow arrows point to the same two mCherry+ foci, which are inside GFP+ cells. **b**) Maximum-intensity projection of optical sections as described for Fig. 2a, additionally using the Lysotracker dye (*n* = 4, one representative embryo is shown). The yellow box highlights the zoomed-in window for **c. c**) Zoomed-in view showing time points during live imaging. One mCherry+ cell is highlighted with a yellow arrow at the start of the experiment, which becomes fragmented and the remnants are highlighted with yellow arrowheads. **d**) UMAP of dual+ (top) and mCherry+

(bottom) cells coloured by their original sort group. **e**) Percentage of cells with extraembryonic origin (defined as Rhox5+/Trap1a+ cells) within the gastrointestinal tract from E9.5 to E15.5 split by Epcam expression status (data from Ref. 19). **f**) Flow cytometry dot plot of the limb from an E13.5 lineage-traced embryo (*n* = 4). **g**) Flow cytometry dot plots of the colon and small intestine from E13.5 lineage-traced embryos showing the endoderm (EPCAM+) and non-endoderm (EPCAM−) fractions. Left: mCherry+ cells are not detected (*n* = 4). Right: mCherry+ cells are not detected in the colon, while a trace amount is detected in the small intestine (*n* = 5). **h**) EPCAM+ content of the posterior part of E9.5 embryos, and E13.5 colon and small intestine (*n* = 9) corresponding to Fig. 2h. Bars denote the mean, error bars denote the standard deviation, and single replicates are indicated by dots. **i**) EPCAM+, mCherry+ and dual+ content of the posterior part of E9.5 embryos (*n* = 9) and E12.5 colon and small intestine (*n* = 3). Plot characteristics are the same as in **h**.

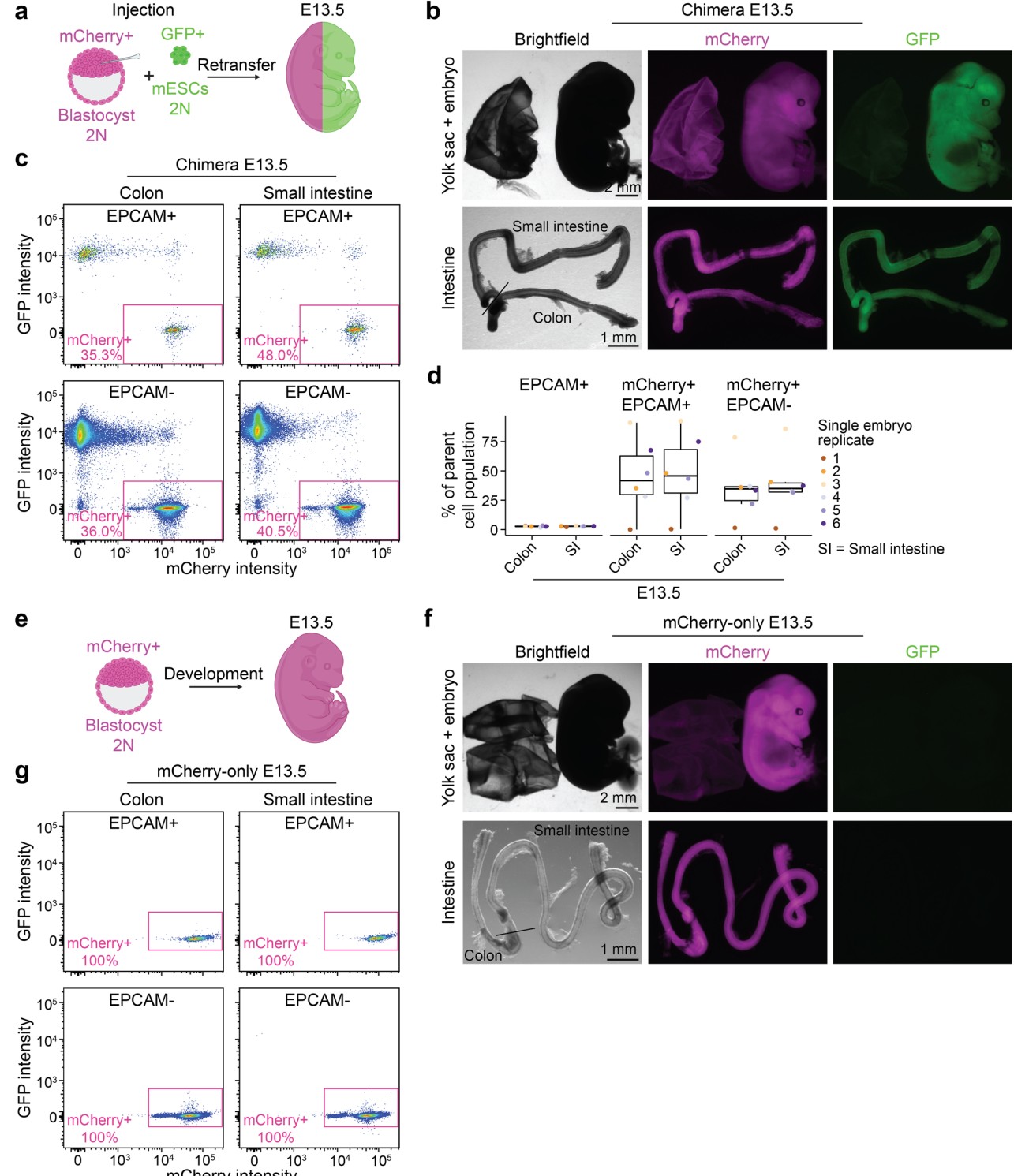

Extended Data Fig. 5 | See next page for caption.

**Extended Data Fig. 5 | Full developmental potential of mCherry+ cells.**
**a**) Schematic illustrating the generation of a chimaeric embryo at E13.5 by diploid complementation with blastocyst injection (as described in Extended Data Fig. 2d,e). This served as a control to show that mCherry+ cells, when also contributing via the embryonic lineage, have full developmental potential.
**b**) Bright-field and fluorescence microscopy images of an E13.5 chimaeric embryo and its corresponding yolk sac, generated by diploid complementation with blastocyst injection. The intestine was manually separated into the colon and small intestine, indicated by the black line (*n* = 6, one representative embryo is shown). **c**) Representative flow cytometry dot plots of the corresponding colon and small intestine from an E13.5 chimeric embryo showing the endoderm (EPCAM+) and the non-endoderm (EPCAM−) fractions (*n* = 6). mCherry+ cells are present in significant proportion confirming that mCherry+ cells are eliminated from the embryo only if they originate from the extraembryonic lineage, such as in our two-colour lineage tracing. **d**) Boxplot showing the abundance of epithelial endoderm cells (EPCAM+), the abundance of mCherry+ cells in the endoderm

(EPCAM+) and non-endoderm (EPCAM−) populations in the organs isolated from E13.5 chimeras. The single embryo replicates are indicated by colour-coded dots. Lines denote the median, edges denote the IQR, whiskers denote 1.5× IQR, and minima/maxima are defined by dots. **e**) Schematic illustrating the generation of a fully mCherry+ embryo at E13.5 via natural mating. This served as a control to exclude that silencing of the mCherry transgene is the reason why mCherry+ extraembryonic cells are not detected in E13.5 embryos generated by the two-colour lineage tracing (Fig. 2). **f**) Bright-field and fluorescence microscopy images of an mCherry-only E13.5 embryo and its corresponding yolk sac generated via natural mating (*n* = 1). The intestine was manually separated into the colon and small intestine, indicated by the black line. **g**) Flow cytometry dot plots of the colon and small intestine from an E13.5 mCherry-only embryo showing the endoderm (EPCAM+) and the non-endoderm (EPCAM−) fractions (*n* = 1). All the cells are mCherry+, and no silencing of the mCherry transgene occurs.

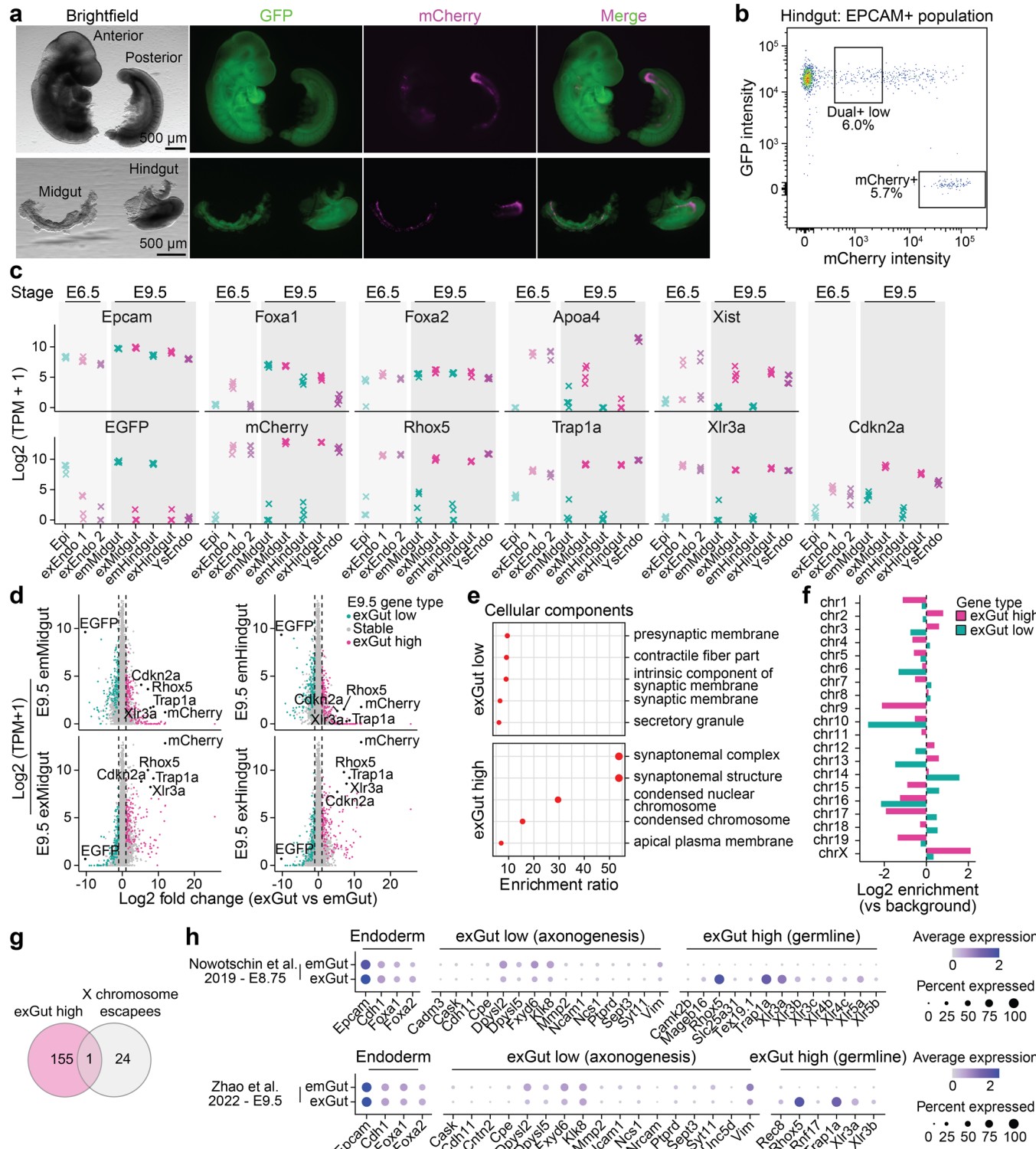

**Extended Data Fig. 6 | See next page for caption.**

**Extended Data Fig. 6 | Gene expression differences between embryonic and extraembryonic gut cells. a)** Bright-field and fluorescence microscopy images of an E9.5 embryo generated via the two-colour lineage tracing. The embryo was manually split into anterior and posterior halves (upper row). The posterior half, containing a large fraction of mCherry+ cells, was used for manually isolating the midgut and the tailbud contains the hindgut (lower row). These were used for sorting, then RNA-seq and WGBS ($n = 16$, one representative embryo is shown, corresponding tissues from four embryos were pooled). **b)** Flow cytometry dot plot of the epithelial fraction (EPCAM+) from the pooled hindgut tissues ($n = 4$). mCherry and GFP intensities were used to sort mCherry+ extraembryonic gut cells and dual+ cells with low mCherry intensity as embryonic hindgut. Our single-cell RNA-seq experiment (Fig. 1) confirmed that epithelial dual+ cells are gut endoderm of embryonic origin and, therefore, ideal to utilize as a stage-matched embryonic comparison. **c)** Log2-transformed expression of origin and lineage marker genes for E6.5 epiblast and extraembryonic endoderm as well as E9.5 gut and yolk sac endoderm in single replicates. **d)** Scatterplot comparing the log2 fold change between exGut and emGut samples with the average log2-transformed expression in emMidgut, emHindgut, exMidgut and exHindgut. **e)** Overrepresentation analysis of exGut low and high genes in cellular components. **f)** Log2-transformed enrichment of the chromosomal location of exGut low and high genes compared to the genomic background distribution of all genes (=0 equals no difference, > 0 implies enrichment, < 0 implies depletion). exGut high genes are enriched on the X chromosome. **g)** Overlap of exGut high genes with genes known to escape X chromosome inactivation[46]. The small overlap suggests that the expression of exGut high genes is not caused by sex differences between emGut and exGut or the effect of double dosage from X chromosome inactivation escapees. **h)** Average log-normalized expression of endoderm marker genes, axonogenesis-associated exGut low genes and germline-associated exGut high genes across embryonic (emGut) and extraembryonic (exGut) gut cells from E8.75 (ref. 4) and E9.5 (ref. 19) embryos using published datasets.

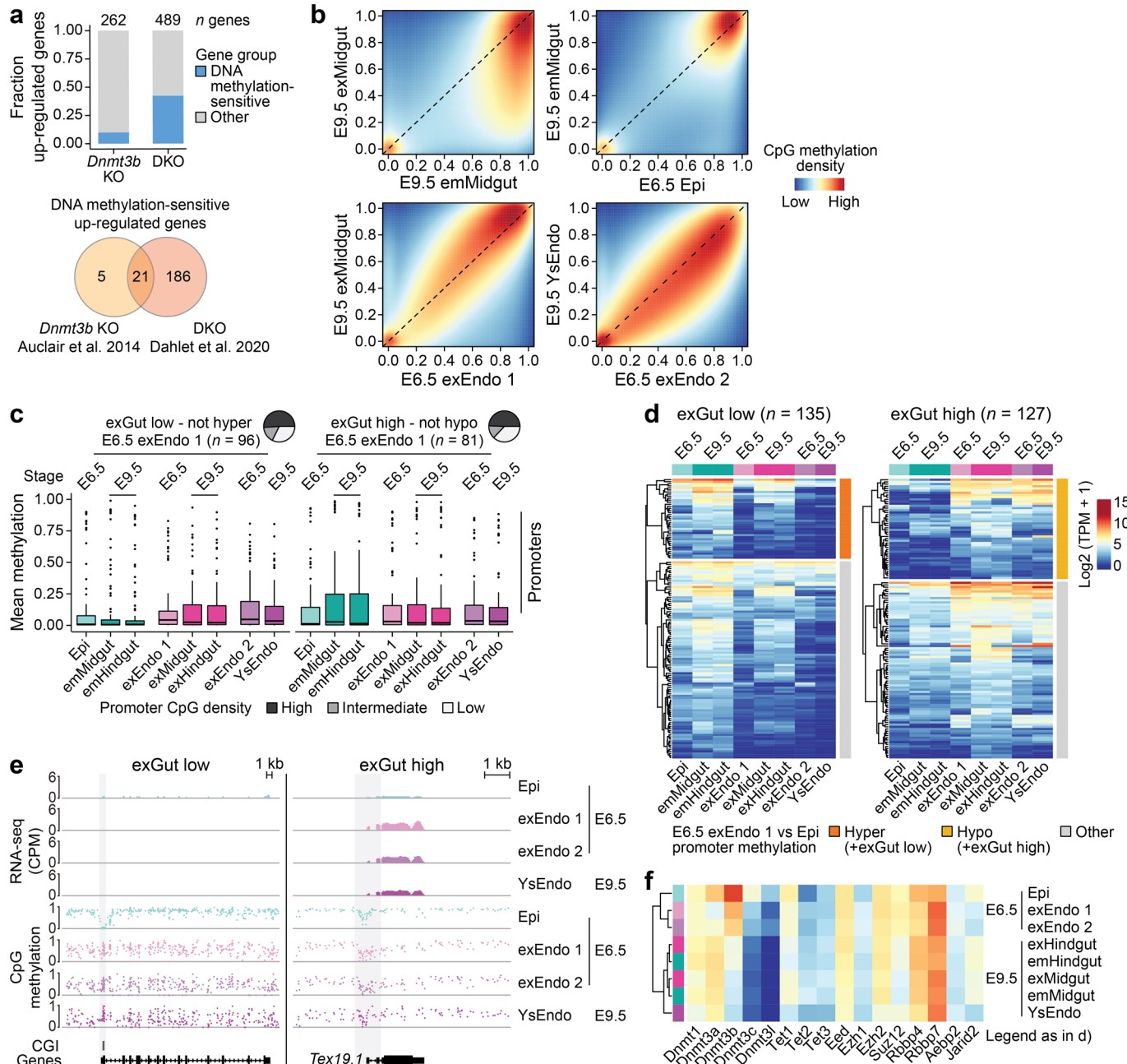

**Extended Data Fig. 7 | Extraembryonic gut cells maintain the extraembryonic DNA methylation landscape. a)** Top: Fraction of differentially upregulated genes in E8.5 Dnmt3b or Dnmt3a/Dnmt3b knockout compared to wild type embryos termed DNA methylation-sensitive (or other) as previously defined by Auclair et al.[28] and Dahlet et al.[29]. Bottom: Overlap of DNA methylation-sensitive upregulated genes in E8.5 *Dnmt3b-* or *Dnmt3a/Dnmt3b*-knockout embryos. **b)** Density plot showing the CpG-wise comparison between E6.5 progenitors, E9.5 emMidgut, exMidgut and YsEndo (WGBS). **c)** Boxplot showing the promoter methylation of exGut low genes that are not hypermethylated in E6.5 exEndo 1 compared to epiblast (left) and exGut high genes that are not hypomethylated in E6.5 exEndo 1 compared to epiblast (right). Low promoter methylation can be observed across all stages and tissues. Pie charts indicate the promoter CpG density of the respective gene sets. Boxplot characteristics and sample sizes as in Fig. 4d. **d)** Log2-transformed expression of all exGut low and high genes with

sufficient promoter coverage by WGBS (see Methods) split by methylation status in the E6.5 exEndo 1 compared to Epi. All profiled E6.5 and E9.5 tissues are shown. **e)** Genome browser track of the *Mmp2* (exGut low) and *Tex19.1* (exGut high) loci showing RNA-seq coverage and WGBS for the E6.5 progenitor cells and the E9.5 YsEndo. Mmp2 is lowly expressed in the epiblast (unmethylated promoter) and not expressed in the extraembryonic tissues (hypermethylated promoter). Tex19.1 is expressed across all tissues but higher in the exEndo and YsEndo, which correlates with stronger promoter hypomethylation. **f)** Heatmap showing the expression levels of epigenetic regulators in E6.5 progenitors and E9.5 gut cells. The different DNA methylation landscapes observed between cells of embryonic and extraembryonic origin are not clearly linked to differences in expression levels of epigenetic regulators. Instead, samples are more similar to each other by developmental time point.

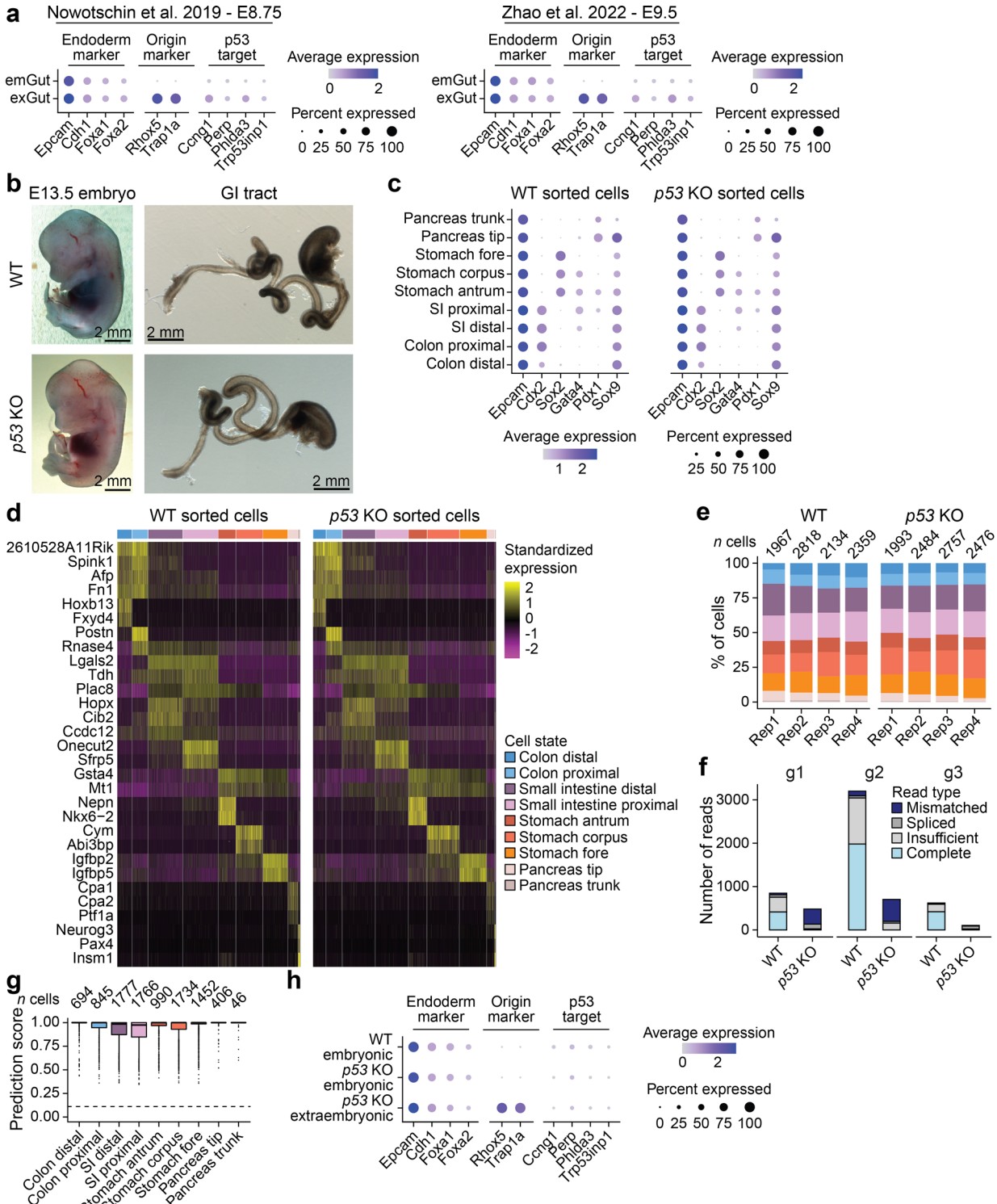

**Extended Data Fig. 8 | Single-cell profiling of *p53* mutant gastrointestinal tract. a)** Average log-transformed expression of the genes shown in Fig. 5a using published single-cell data[4,19]. Cells were subdivided into emGut and exGut based on Rhox5 and Trap1a expression (see Methods). **b)** Bright-field microscopy images of a WT and a *p53* KO E13.5 embryo, and the isolated gastrointestinal (GI) tracts (*n* = 4, one representative embryo is shown for each condition). No developmental phenotype is observed for the *p53* KO embryo and GI tract compared to the WT. **c)** Average log-transformed expression of gastrointestinal epithelial marker genes in single cells corresponding to cell states annotated for WT and *p53* KO embryos. **d)** Heatmap representation of the standardized, log-normalized expression levels of marker genes of cell states in E13.5 WT and *p53* KO cells. **e)** Percentage of single cells assigned to the different cell states for each

E13.5 WT and *p53* KO single embryo replicate. **f)** Quantification of different read types spanning the Cas9 target sequences (g1 to g3) in WT and *p53* KO cells. For the *p53* KO cells, virtually no complete, error-free reads can be found implying the successful knockout of the target gene. **g)** Boxplot of prediction scores for *p53* KO cells as they are assigned to their respective cell state based on the local neighbourhood of WT cells. The dashed line denotes the prediction score that reflects equal association with any of the nine cell states (random assignment). Lines denote the median, edges denote the IQR, whiskers denote 1.5 × IQR, and outliers are represented by dots (*n* = 4 biological replicates). **h)** Average log-transformed expression of the genes shown in Fig. 5a in the E13.5 WT and *p53* KO cells. Cells were subdivided into emGut and exGut based on Rhox5 and Trap1a expression (see Methods).

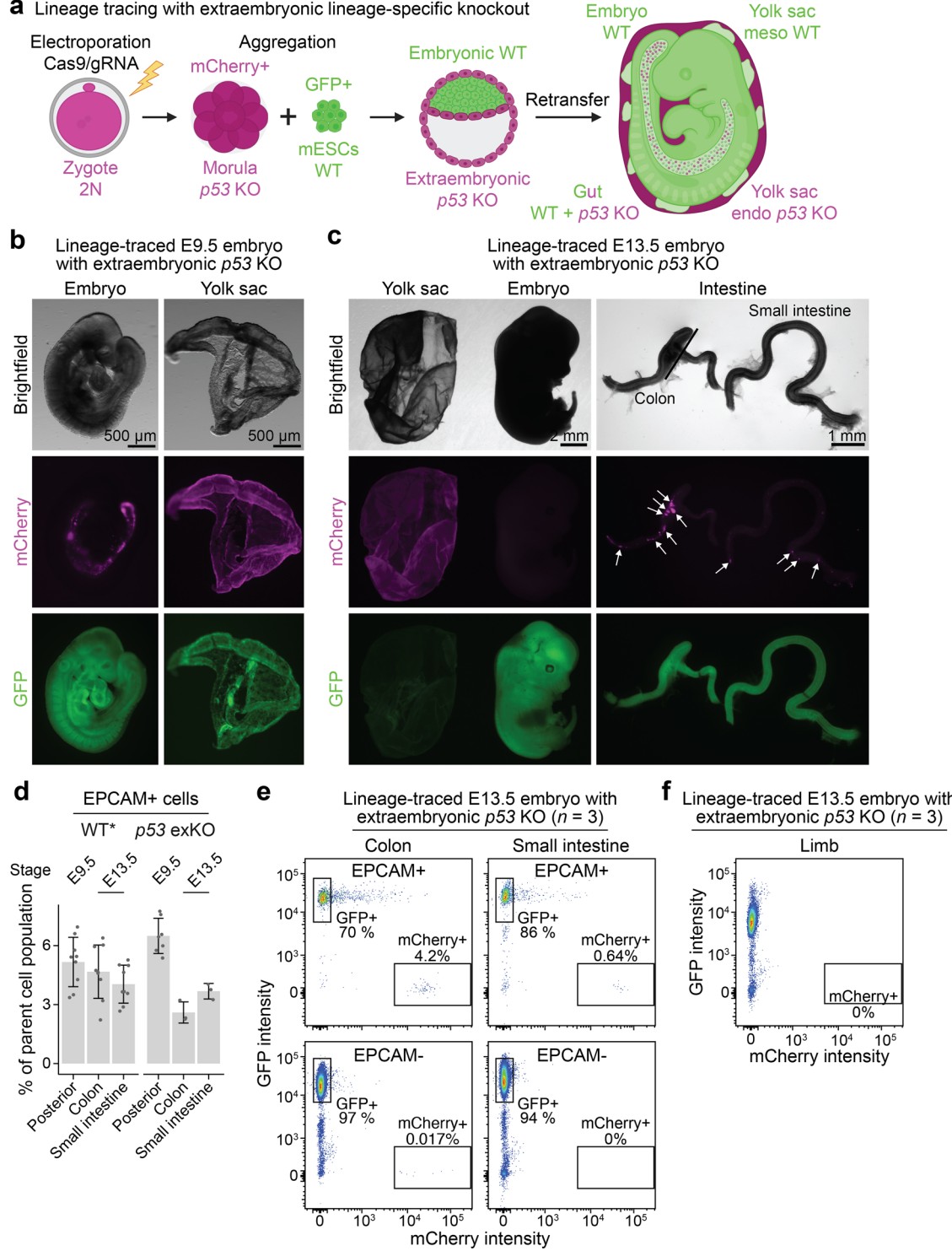

**a** Lineage tracing with extraembryonic lineage-specific knockout

**b** Lineage-traced E9.5 embryo with extraembryonic *p53* KO

**c** Lineage-traced E13.5 embryo with extraembryonic *p53* KO

**d** EPCAM+ cells

**e** Lineage-traced E13.5 embryo with extraembryonic *p53* KO (*n* = 3)

**f** Lineage-traced E13.5 embryo with extraembryonic *p53* KO (*n* = 3)

**Extended Data Fig. 9 | See next page for caption.**

**Extended Data Fig. 9 | Two-colour lineage tracing with extraembryonic *p53* knockout. a**) Schematic illustrating the two-colour lineage tracing strategy combined with extraembryonic lineage-specific *p53* knockout (KO). The mCherry+ zygote is electroporated with Cas9/gRNA complex, and once reaching the pre-compaction morula stage, the *p53* KO embryo is aggregated with a GFP+ mESC colony. As a result, the extraembryonic lineages are *p53* KO, including the mCherry+ gut cells of extraembryonic origin, while the GFP+ embryonic lineage is wild type. **b**) Bright-field and fluorescence microscopy images of an E9.5 embryo and its corresponding yolk sac, generated via the two-colour lineage tracing combined with extraembryonic lineage-specific *p53* KO (*n* = 5, one representative embryo is shown). **c**) Bright-field and fluorescence microscopy images of an E13.5 embryo and its corresponding yolk sac, generated via the two-colour lineage tracing combined with extraembryonic lineage-specific *p53* KO. The dissected intestine was manually separated into the colon and small intestine, indicated by the black line. The overall GFP+ intestine contains

mCherry+ cells, indicated by the white arrows (*n* = 3, one representative embryo is shown). **d**) EPCAM+ content of WT lineage-traced embryos, showing the posterior part of E9.5 embryos, and the colon and small intestine from E13.5 embryos (*WT data from Extended Data Fig. 4h used here as comparison, *n* = 9). Additionally, EPCAM+ content of lineage-traced embryos with extraembryonic-specific *p53* KO is presented showing the posterior part of E9.5 embryos (*n* = 5) and the colon and small intestine from E13.5 embryos (*n* = 3). Bars denote the mean, error bars denote the standard deviation, and single replicates are indicated by dots. **e**) Flow cytometry dot plots of the corresponding colon (left) and small intestine (right) from an E13.5 lineage-traced embryo with extraembryonic lineage-specific *p53* KO showing the endoderm fraction (EPCAM+, top) and non-endoderm fraction (EPCAM–, bottom) of the isolated organs (*n* = 3). mCherry+ cells with extraembryonic origin are detected. **f**) Flow cytometry dot plot of the limb from an E13.5 lineage-traced embryo (*n* = 3).

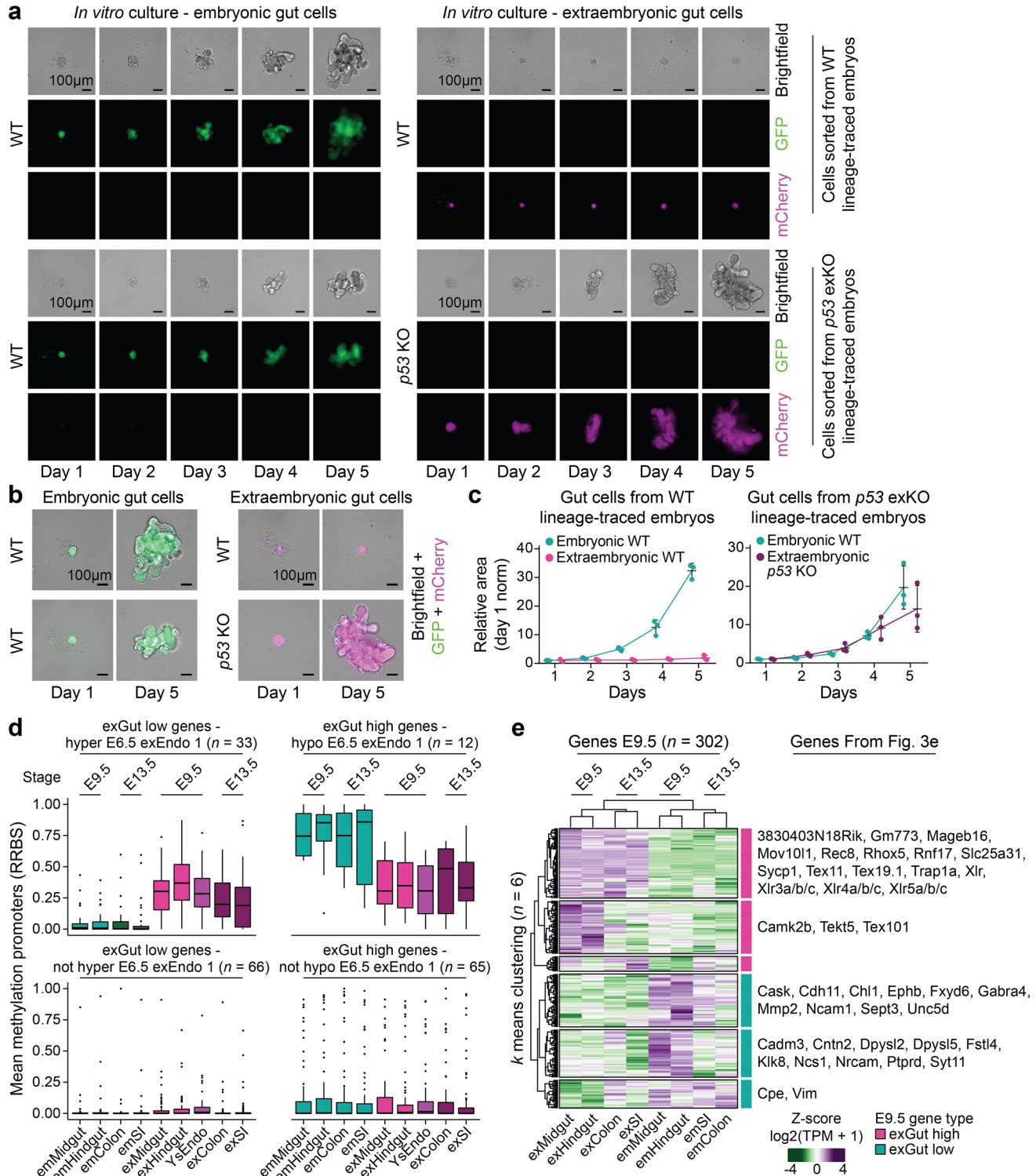

**Extended Data Fig. 10 | Persisting p53-mutant extraembryonic cells in vitro and in vivo. a**) Single channel bright-field and fluorescence microscopy images, showing time points of representative in vitro cultured gut cell assemblies over 5 days, sorted from the posterior part of wild type lineage-traced embryos (top) or the posterior part of embryos with extraembryonic *p53* KO (bottom) at E9.5 (*n* = 3, one representative gut cell assembly is shown for each condition). Embryonic gut cells show substantial growth and WT extraembryonic gut cells do not show signs of proliferative capacity, while *p53* KO extraembryonic gut cells show substantial and comparable growth to the embryonic gut cells. **b**) Merged bright-field and fluorescence microscopy images from the in vitro culture experiment at day 1 and day 5 from **a**. **c**) Growth quantification of in vitro cultured gut cell assemblies (represented in **a**,**b**) as determined by the relative area calculated by normalizing

the assembly area to the average area on day 1 (*n* = 3). Central line denotes the mean, whiskers denote standard deviation. **d**) Boxplots showing the exGut low and high gene groups separated by promoter methylation in the E6.5 exEndo 1 compared to the epiblast, for E9.5 and E13.5 tissues (RRBS). The distinct promoter methylation patterns of cells with embryonic and extraembryonic origin are still present at E13.5. Lines denote the median, edges denote the IQR, whiskers denote 1.5× IQR, and outliers are represented by dots (*n* = 3–4 biological replicates). **e**) *Z*-score-transformed expression across E9.5 gut and E13.5 intestine samples of all E9.5 exGut low and high genes split using *k*-means clustering. Germline- and axonogenesis-associated genes shown in Fig. 3e are indicated next to their assigned cluster.

# Reporting Summary

## Statistics

For all statistical analyses, confirm that the following items are present in the figure legend, table legend, main text, or Methods section.

| n/a | Confirmed | |
|---|---|---|
| ☐ | ☒ | The exact sample size (*n*) for each experimental group/condition, given as a discrete number and unit of measurement |
| ☐ | ☒ | A statement on whether measurements were taken from distinct samples or whether the same sample was measured repeatedly |
| ☐ | ☒ | The statistical test(s) used AND whether they are one- or two-sided *Only common tests should be described solely by name; describe more complex techniques in the Methods section.* |
| ☐ | ☒ | A description of all covariates tested |
| ☐ | ☒ | A description of any assumptions or corrections, such as tests of normality and adjustment for multiple comparisons |
| ☒ | ☐ | A full description of the statistical parameters including central tendency (e.g. means) or other basic estimates (e.g. regression coefficient) AND variation (e.g. standard deviation) or associated estimates of uncertainty (e.g. confidence intervals) |
| ☐ | ☒ | For null hypothesis testing, the test statistic (e.g. *F*, *t*, *r*) with confidence intervals, effect sizes, degrees of freedom and *P* value noted *Give P values as exact values whenever suitable.* |
| ☒ | ☐ | For Bayesian analysis, information on the choice of priors and Markov chain Monte Carlo settings |
| ☒ | ☐ | For hierarchical and complex designs, identification of the appropriate level for tests and full reporting of outcomes |
| ☒ | ☐ | Estimates of effect sizes (e.g. Cohen's *d*, Pearson's *r*), indicating how they were calculated |

*Our web collection on statistics for biologists contains articles on many of the points above.*

## Software and code

Policy information about availability of computer code

| | |
|---|---|
| Data collection | 10x Genomics Cell Ranger (version 6.0.1), deMULTIplex (version 1.0.2), cutadapt (version 4.1), STAR (version 2.7.9a), stringtie (version 2.0.6), BSMAP (version 2.90), MOABS (version 1.3.2), GATK (version 4.3.0.0), BD FACS Diva (version 8.0.1), Zeiss ZEN Blue (version 3.5), ZEN 2014 |
| Data analysis | R (version 4.1.0), Seurat (version 4.1.0), DESeq2 (version 1.32.0), WebGestaltR (version 0.4.4), ComplexHeatmap (version 2.7.11), bedtools (version 2.30.0), metilene (version 0.2-8), IGV (version 2.15.2), Zeiss ZEN Black (version 2.3 SP1 FP3), ImageJ2 (version 2.3.0/1.53q), FlowJo (version 10.8.1), samtools (version 1.18), deeptools (version 3.5.2)<br><br>Custom code is available at https://doi.org/10.5281/zenodo.10926934. |

For manuscripts utilizing custom algorithms or software that are central to the research but not yet described in published literature, software must be made available to editors and reviewers. We strongly encourage code deposition in a community repository (e.g. GitHub). See the Nature Portfolio guidelines for submitting code & software for further information.

# Data

Policy information about availability of data

All manuscripts must include a data availability statement. This statement should provide the following information, where applicable:

- Accession codes, unique identifiers, or web links for publicly available datasets
- A description of any restrictions on data availability
- For clinical datasets or third party data, please ensure that the statement adheres to our policy

Sequencing data sets generated within the scope of this study have been deposited in the Gene Expression Omnibus under accession no. GSE250084. scRNA-seq data sets of E8.75 gut endoderm and E9.5-E15.5 gastrointestinal tract were obtained from GSE123046 and GSE186525, respectively. WGBS data sets of wild type E6.5 epiblast were obtained from GSE137337. The mouse reference genome mm10 was obtained from UCSC (https://hgdownload.soe.ucsc.edu/goldenPath/mm10/bigZips/). Annotations of CpG islands for mm10 were downloaded from UCSC (https://genome.ucsc.edu/cgi-bin/hgTables). The mm10 gene annotation was downloaded from GENCODE (VM23, https://www.gencodegenes.org/mouse/release_M23.html). Source data are provided at https://doi.org/10.5281/zenodo.10926934. All other data supporting the findings of this study are available from the corresponding author on reasonable request

# Human research participants

Policy information about studies involving human research participants and Sex and Gender in Research.

| | |
|---|---|
| Reporting on sex and gender | N/A |
| Population characteristics | N/A |
| Recruitment | N/A |
| Ethics oversight | N/A |

Note that full information on the approval of the study protocol must also be provided in the manuscript.

# Field-specific reporting

Please select the one below that is the best fit for your research. If you are not sure, read the appropriate sections before making your selection.

☒ Life sciences  ☐ Behavioural & social sciences  ☐ Ecological, evolutionary & environmental sciences

For a reference copy of the document with all sections, see nature.com/documents/nr-reporting-summary-flat.pdf

# Life sciences study design

All studies must disclose on these points even when the disclosure is negative.

| | |
|---|---|
| Sample size | No statistical methods were used to predetermine sample sizes but our sample sizes are similar to those reported in previous publications (Auclair et al. Genome Biology 2014, Nowotschin et al. Nature 2019, Grosswendt et al. Nature 2020, Scheibner et al. Nature Cell Biology 2021, Rothová et al. Nature Cell Biology 2022). Sample sizes are indicated in the figure panels or legends. |
| Data exclusions | Prior to downstream analysis and experiments, resorping embryos were excluded. For the downstream experiments with the two-color lineage-tracing, only embryos with gut-specific mCherry+ signal were used, mCherry+ only embryos were excluded. No other data was excluded. |
| Replication | For RNAseq and RRBS experiments three to four replicates were generated. For E9.5, embryos were pooled while for E6.5 and E13.5, single embryo replicates were generated. For WGBS experiments, two replicates were generated for each E6.5 tissue (exEndo 1 and 2) and one replicate was generated for each E9.5 tissue. For the E9.5 scRNA-seq analysis, one experiment was performed, which contained cells of different sort groups (dual+ low, intermediate and high populations, mCherry+ population) from 15 pooled embryos. For the E13.5 scRNA-seq analysis, four WT embryos and four Trp53 knockout embryos were included in the experimental set-up labeled by MULTI-seq barcodes, which allowed comparison of cell state distributions across single embryo replicates. For imaging experiments and FACS analysis, 3-10 embryos were analyzed, the exact number is indicated in the respective figure or legend. All attempts at replication were successful. |
| Randomization | For assessing the outcome of the complementation assays, embryos were collected without a preconceived selection strategy or prioritization by morphology. Our genomic analyses are independent of human intervention and analyze each sample equally and in an unbiased fashion. |
| Blinding | Data collection and analysis were not performed blind to the conditions of the experiments. Blinding was not relevant for this study since this is not an intervention study. However, our analytical pipeline followed uniform criteria applied to all samples, allowing us to analyse our data in an unbiased manner. |

# Reporting for specific materials, systems and methods

We require information from authors about some types of materials, experimental systems and methods used in many studies. Here, indicate whether each material, system or method listed is relevant to your study. If you are not sure if a list item applies to your research, read the appropriate section before selecting a response.

## Materials & experimental systems

| n/a | Involved in the study |
|---|---|
| ☐ | ☒ Antibodies |
| ☐ | ☒ Eukaryotic cell lines |
| ☒ | ☐ Palaeontology and archaeology |
| ☐ | ☒ Animals and other organisms |
| ☒ | ☐ Clinical data |
| ☒ | ☐ Dual use research of concern |

## Methods

| n/a | Involved in the study |
|---|---|
| ☒ | ☐ ChIP-seq |
| ☐ | ☒ Flow cytometry |
| ☒ | ☐ MRI-based neuroimaging |

## Antibodies

| | |
|---|---|
| Antibodies used | Primary antibodies: Foxa2 (HNF-3β) antibody (Santa Cruz, sc-6554, 1:250 dilution), E-cadherin antibody (Cell Signaling Technology, 3195, 1:250 dilution), cleaved-Caspase3 antibody (Cell Signaling Technology, 9661, 1:250 dilution), mCherry antibody (abcam, ab167453, 1:200 dilution), mCherry antibody (antibodies-online, ABIN1440058, 1:500 dilution), Epcam (CD326) antibody Alexa Fluor 647 (BioLegend, 118211, 1:400 dilution)<br>Secondary antibodies: Alexa Fluor 647 Donkey anti-Goat (Invitrogen, A21447, 1:250 dilution), Alexa Fluor 647 Donkey anti-Rabbit (Invitrogen, A31573, 1:250 dilution), Alexa Fluor 546 Donkey anti-Goat (Invitrogen, A11056, 1:250 dilution), Alexa Fluor 568 Donkey anti-Rabbit (Invitrogen, A10042, 1:250 dilution) |
| Validation | All antibodies were validated by their manufacturers:<br>Foxa2 antibody: https://www.scbt.com/p/hnf-3beta-antibody-m-20?productCanUrl=hnf-3beta-antibody-m-20&_requestid=2453355<br>E-cadherin antibody: https://www.cellsignal.de/products/primary-antibodies/e-cadherin-24e10-rabbit-mab/3195<br>cleaved-Caspase3 antibody: https://www.cellsignal.de/products/primary-antibodies/cleaved-caspase-3-asp175-antibody/9661<br>mCherry antibody: https://www.abcam.com/mcherry-antibody-ab167453.html<br>mCherry antibody: https://www.antibodies-online.com/antibody/1440058/anti-mCherry+Fluorescent+Protein+antibody/<br>Epcam (CD326) antibody: https://www.biolegend.com/en-us/search-results/alexa-fluor-647-anti-mouse-cd326-ep-cam-antibody-4973?GroupID=BLG5748<br>lexa Fluor 647 Donkey anti-Goat: https://www.thermofisher.com/antibody/product/Donkey-anti-Goat-IgG-H-L-Cross-Adsorbed-Secondary-Antibody-Polyclonal/A-21447<br>Alexa Fluor 647 Donkey anti-Rabbit: https://www.thermofisher.com/antibody/product/Donkey-anti-Rabbit-IgG-H-L-Highly-Cross-Adsorbed-Secondary-Antibody-Polyclonal/A-31573<br>Alexa Fluor 546 Donkey anti-Goat: https://www.thermofisher.com/antibody/product/Donkey-anti-Goat-IgG-H-L-Cross-Adsorbed-Secondary-Antibody-Polyclonal/A-11056<br>Alexa Fluor 568 Donkey anti-Rabbit: https://www.thermofisher.com/antibody/product/Donkey-anti-Rabbit-IgG-H-L-Highly-Cross-Adsorbed-Secondary-Antibody-Polyclonal/A10042 |

## Eukaryotic cell lines

Policy information about cell lines and Sex and Gender in Research

| | |
|---|---|
| Cell line source(s) | The cell lines used in this study (mCherry+ mESCs and GFP+ mESCs) were generated in house, are male and derived from an F1G4 genetic background (George et. al., 2007), which was obtained from the laboratory of A. Nagy. |
| Authentication | None of the transgenic cell lines generated in this study have been authenticated. |
| Mycoplasma contamination | All cell lines used in this study tested negative for mycoplasma contamination. |
| Commonly misidentified lines (See ICLAC register) | No commonly misidentified cell lines were used in this study. |

## Animals and other research organisms

Policy information about studies involving animals; ARRIVE guidelines recommended for reporting animal research, and Sex and Gender in Research

| | |
|---|---|
| Laboratory animals | All animal work performed in this study was approved by the local authorities (LAGeSo Berlin, license number G0243/18, G0098/23). Mice were kept in individually ventilated cages (IVC) under specified pathogen free (SPF) conditions in animal rooms with a light cycle of 12h/12h, a temperature of 20-24°C and a humidity of 45-65%. The mice received autoclaved water and a standard rodent diet ad libidum. For embryo generation, Hsd:ICR (CD-1) females (age 6-20 weeks) were mated with the indicated males (CD-1 or mCherry+) (age ≥8weeks). Blastocysts resulting from complementation assays were transfered into CD-1 foster females (age 6-20 weeks) that |

| | |
|---|---|
| | were previously mated to vasectomized CD-1 males (age ≥12 weeks) to induce pseudopregnancy. Hsd:ICR (CD-1) and C57BL/6J animals were obtained from Envigo/Inotiv. |
| Wild animals | The study did not involve wild animals. |
| Reporting on sex | GFP+ embryos originating from the lineage tracing assay are male because the mESC line used in the complementation assay is male. mCherry+ pre-implantation embryos were generated via natural mating, resulting in both male and female cells. |
| Field-collected samples | No field-collected samples were used in this study. |
| Ethics oversight | All research described here complies with the relevant ethical regulations and was approved by the LAGeSo Berlin, license number G0243/18 and G0098/23. |

Note that full information on the approval of the study protocol must also be provided in the manuscript.

# Flow Cytometry

## Plots

Confirm that:

☒ The axis labels state the marker and fluorochrome used (e.g. CD4-FITC).

☒ The axis scales are clearly visible. Include numbers along axes only for bottom left plot of group (a 'group' is an analysis of identical markers).

☒ All plots are contour plots with outliers or pseudocolor plots.

☒ A numerical value for number of cells or percentage (with statistics) is provided.

## Methodology

| | |
|---|---|
| Sample preparation | Deciduae were collected into ice-cold HBSS (Gibco #14175095), E9.5 embryos (somite number 18-28) were dissected in ice-cold M2 medium (Merck #MR-015-D), the extraembryonic tissues were completely removed, and the yolk sac was kept. For the single-cell RNA-seq analysis to determine the cell type identities of mCherry+ and dual+ cells, whole lineage-traced embryos were used. For assessing extraembryonic cell content in lineage-traced embryos (comparing wild type and p53 extraembryonic-specific knockout), the embryos were cut into two halves with a micro knife along the anterior-posterior axis, and the posterior half was used further. For RNA-seq, RRBS, and WGBS experiments, wild type lineage-traced E9.5 embryos were cut into two halves with a micro knife along the anterior-posterior axis. From the posterior half, the midgut was manually isolated using tungsten needles (Fine Science Tools #10130-10), and the most posterior part was also kept containing the hindgut. For each midgut and hindgut replicate, corresponding tissues from four embryos were pooled. The embryos, the isolated tissues, and the yolk sac were washed in ice-cold HBSS, dissociated with 0,25 % Trypsin-EDTA (Gibco #25200056) for 10 minutes at 37°C to obtain single cells. This was quenched with KnockOut DMEM (Thermo Fisher Scientific #10829018) with 10% FBS (PAN-Biotech #P30-2602) and 0,05 mg/ml DNase I (Merck #11284932001) to dissociate the cells via pipetting, and the cells were also washed once with this buffer. After blocking with Normal Mouse Serum (Invitrogen #31881) for 5 minutes on ice, cells were stained for EPCAM (Alexa Fluor® 647 anti-EPCAM, BioLegend #118212) in FACS buffer (HBSS with 2% FBS and 0,5 mM EDTA (Thermo Fischer Scientific #15575020)) for 10 minutes on ice. Specifically for the pooled midgut and hindgut samples, enrichment of EPCAM+ cells was performed by magnetic separation (MACS) using Anti-Cy5/Anti-Alexa Fluor 647 MicroBeads (Miltenyi Biotec #130-091-395), following the manufacturer's instructions with the MS columns (Miltenyi Biotec #130-042-201) and the OctoMACS™ Separator (Miltenyi Biotec #130-042-109). Last, cells were stained with DAPI (0.02 %, Roche Diagnostics #102362760019) in FACS buffer for 8 minutes on ice, then were washed once and resuspended in FACS buffer, and kept on ice until flow cytometry analyses or sorting. E13.5 embryos were dissected in DMEM/F-12 (Thermo Fischer Scientific #21041025) with 10 % FBS (PAN-Biotech #P30-2602). For scRNA-seq analysis of the wild type and p53 knockout embryos (see below), the gastrointestinal tract was isolated, then dissociation, staining for EPCAM, enrichment by MACS, and preparation for FACS was performed as described above for E9.5 midgut and hindgut samples. For RNA-seq and RRBS, E13.5 lineage-traced embryos were collected, and the intestine was isolated, and split into the small intestine and colon parts with a micro knife. Then, dissociation, staining for EPCAM, and sample preparation for FACS were performed as described above for E9.5 embryos. |
| Instrument | BD FACS ARIA FUSION (Becton Dickinson) |
| Software | DIVA, FlowJo (v10.8.1). |
| Cell population abundance | Given the low input for our sorting experiments, sort check was not performed on the sorted material, instead separate samples of the same type were used to sort the desired populations and perform post-sort checks which confirmed the purity of the sort-test sample. |
| Gating strategy | For analysis of extraembryonic gut cell content in embryos and organs, the following gating strategy was set up. First, an FSC-A vs SSC-A gating was used to identify the cell population. Next, two doublet removal steps were performed (FSC-W vs FSC-H and SSC-W vs SSC-H). Alive cells were gated based on DAPI, then epithelial/endoderm cells were gated based on EPCAM. For sorting gut endoderm and yolk sac endoderm cells, the following strategy was used. First, an FSC-A vs SSC-A gating was used to identify the cell population. Alive cells were gated based on DAPI, then epithelial/endoderm cells were gated based on EPCAM. Next, two doublet removal steps were performed (FSC-W vs FSC-H and SSC-W vs SSC-H). Finally, cells were sorted based on GFP and mCherry intensities. For sorting endoderm cells from the gastrointestinal tracts, the following strategy was used. First, an FSC-A vs SSC-A gating was used to identify the cell population. Alive cells were gated based on DAPI, then epithelial/endoderm cells were gated based on EPCAM. Next, two doublet removal steps were performed (FSC-W vs FSC-H |

and SSC-W vs SSC-H) and single cells were sorted. Representative gating strategies are provided in Supplementary Information 1, and example plots are also provided in the Figures and Extended Data Figures.

☒ Tick this box to confirm that a figure exemplifying the gating strategy is provided in the Supplementary Information.

