## [Peer Review File · Nature Cell Biology]

Peer Review Information

Journal: Nature Cell Biology

Manuscript Title: Extraembryonic gut endoderm cells undergo programmed cell death during development

Corresponding author name(s): Dr Alexander Meissner

Editorial Notes:

Reviewer Comments & Decisions:

Decision Letter, initial version:

Dear Alex,

Your manuscript "Selective elimination of extraembryonic cells from the developing gut", has now been seen by 3 referees, who are experts in cell death clearance (referee 1); development, endoderm, DNA methylation (referee 2); and endoderm, lineage tracing, single cell technologies, epigenetics (referee 3), and whose comments are pasted below. In light of their advice, we regret that we cannot offer to publish the study in Nature Cell Biology.

As you will see, although the reviewers find this work potentially interesting, they raise some serious concerns that question the strength of the data and their physiological relevance.

We would be open to the possibility of considering an appeal to this decision with a revised manuscript that would fully address the referee concerns. However, any decision to re-review such a revised study would depend on the strength of the revisions and the published literature at the time of resubmission. I would also like to clarify that we would expect this revision to address all issues raised by the referees, except for the issue about providing additional mechanistic insights, which we do consider insightful, but also beyond the scope of the current study.

We are very sorry that we could not be more positive on this occasion, but we thank you for the opportunity to consider this work.

With kind regards,
Stelios

Stylianos Lefkopoulos, PhD
He/him/his
Associate Editor
Nature Cell Biology
Springer Nature
Heidelberger Platz 3, 14197 Berlin, Germany

E-mail: stylianos.lefkopoulos@springernature.com
Twitter: @s_lefkopoulos

Reviewers' comments:

Reviewer #1 (Remarks to the Author):

This is an interesting and exciting new manuscript by Batki et al investigating the nature and persistence of extraembryonic cells to the developing mouse embryonic gut. Using a novel lineage tracing method, the authors demonstrate the presence and localization of extraembryonic cells. Because of their permanent label, the authors are then able to isolate them and using 10x RNAseq and Smart-Seq2 show that they have transcriptional signatures consistent with their origin. Excitingly, they also demonstrate that these extraembryonic cells are no longer present in the mouse gut by E13.5, and hypothesize that they are cleared by neighboring embryonic epithelial cells. Overall, the paper is highly significant and introduces new tools to answer an old and intriguing question of how extraembryonic and embryonic tissues mingle during development. And while the cell clearance hypothesis is exciting and there is evidence of similar phenomena happening in others systems, there is no direct evidence of mCherry+ cells being engulfed by GFP+ cells. The movies provided clearly show cell death of extraembryonic cells. However, whether they are actively being engulfed/phagocytosed by neighboring embryonic cells is unclear.

To make this point, more evidence needs to be shown. This could include incorporation of an acidic dye that labels acidic compartments in these studies (e.g., lysotracker or neutral red), higher resolution imaging to show engulfment, interactions between these cells in culture, or EM to show cell debris within cells. I appreciate none of these methods are easy or straightforward, but as the data stands now with regards to this point, it is unclear if clearance is actively happening. Alternatively, blocking cell death with caspase inhibitors and then seeing a perdurance of the extraembryonic signature would add more weight to this claim.

Reviewer #2 (Remarks to the Author):

This manuscript assesses the fate of visceral endoderm cells which integrate into the definitive endoderm of developing embryos and assume a very similar transcriptional signature. Using diploid complementation, the authors confirm that visceral endoderm cells are integrated in definitive endoderm during gastrulation and that they can be detected in the gut tube until around E12.5. The authors confirm the overall similar transcriptome of VE vs DE cells, but detect a few genes which are differentially expressed. VE upregulated genes are germline genes and relate to activation of p53 mediated apoptosis. The latter is in line with their observation that VE cells stain positive for Caspase and are fragmented and taken up by neighboring cells. The authors further characterize DNA methylation differences and observe that VE cells are generally hypermethylated compared to DE cells. This might also explain the upregulation of germline specific genes which are known to be sensitive to loss of DNA methylation.

The novel findings of this manuscript are that VE cells are lost during later embryonic development and that these cells largely maintain their early embryonic DNA methylation signature, although they assume a largely DE-like transcriptome.

A potential caveat of this study is the use of the diploid complementation system for all assays. It would be important to validate if similar transcriptional changes (or DNA methylation changes, or loss of VE cells) are seen in normally developing embryos (eventually combining lineage tracing and/or use of their newly identified VE specific genes). The authors could show that the mCherry pos. cells are able to form normally looking embryos, however, there are also transcriptional signatures in DE cells (Ex Fig. 6E, axonal development, etc.) which do not seem to be very physiologic.

In addition, resolving the mechanistic basis for either VE cell death or for the DNA methylation independent changes in the VE transcriptome would considerably strengthen the manuscript.

Minor comments:

(1) The expression changes between VE and DE (Fig 3B) could better be shown as mean expression vs. fold change plot with significant genes colored. Supplementary tables with gene expression data (TPM) need to be added.

(2) The heatmaps in Fig. 3D,E, Ex. Fig. 6F could be reordered to better be able to compare VE vs DE at the two developmental stages (E6.5 Epi, VE, E9.5 DE, VE).

(3) Is the overall reduced DNA methylation in VE related to expression or protein abundance changes in the DNA methylation machinery?

(4) line 244: "which also closely mirrors the methylation pattern in most human cancer types" If there is any relation to cancer development it should be explained in the discussion section.

Reviewer #3 (Remarks to the Author):

Synopsis:

In this study, Batki et al. developed a two-color lineage tracing strategy to track and isolate the extraembryonic cell-derived gut endodermal cells. Through this strategy, they found that gut cells with extraembryonic origin are subjected to programmed cell death and uptake by embryonic gut cells, subsequently being eliminated at E13.5. Moreover, they identified that these extraembryonic gut cells acquire a similar gene expression profile to embryonic gut cells, but retain the extraembryonic DNA methylation landscape.

This study claims the developmental fate of some extraembryonic cells in the gut endoderm, but the data are far from convincing. Some key experimental designs and data analyses are confusing.

Furthermore, the biological significance of these extraembryonic-derived gut endodermal cells remains unclear, making this study less meaningful.

Major points:

- 1) Traditional morula aggregation using tetraploid host embryos and diploid ESCs can generate an almost entirely ESC-derived embryo. That is because tetraploid cells fail to contribute to the embryo proper and are only limited to the extraembryonic yolk sac and trophoblasts. In this study, if diploid host embryos never subjected to any treatment are used, how to rule out the contributions of mCherry+ diploid morula cells to the embryo proper? The authors claim the mCherry+ cells as extraembryonic cells, which is unstringent at all. This invalid claim shakes the whole study.
- 2) The authors conclude that the mCherry+/GFP+ dual-labeled cells are resulted from the mCherry+ cell death and subsequent GFP+ cell uptake. This should be a phenomenon termed efferocytosis. Although immunofluorescence for Caspase3 and live imaging showed the programmed death of mCherry+ cells, there is no direct evidence showing the ingestion of mCherry+ cell remnants by the GFP+ cells. Efferocytosis assays such FACS efferocytosis assay and DNase II signals should be performed. Moreover, do the majority of or all the dual-labeled cells come from efferocytosis? How about other reasons such as cell fusion?
- 3) If the mCherry+/GFP+ dual-labeled cells mainly come from efferocytosis, they are embryonic cells, not from the extraembryonic origin. The use of these dual-labeled cells as extraembryonic gut cells to analyze differential gene expression between emVE and DE cells is illogical. Based on the prerequisite that all the mCherry+ cells are derived from the extraembryonic origin (which is not true as pointed out in the Point 1)), only mCherry single-positive cells should be used to compare with the GFP-labeled cells.
- 4) This study shows that extraembryonic emVE exhibits global hypomethylation and local hypermethylation compared to the embryonic DE. That is a common phenomenon in epigenetics, because any two cell populations of different origins may exhibit differences in global and local methylation levels. The authors should provide more about DNA methylation changes and corresponding sets of modified genes during emVE and DE development, rather than this simple comparison. Such as, is there a trend in the global or local methylation level of emVE at E9.5, E10, E11, and E12? Whether the elimination of mCherry+ cells at E13.5 directly related to methylation? The functions of the DNA methylation changes should be experimentally addressed.
- 5) What is the biological roles of these so-called extraembryonic-derived gut endodermal cells? If these cells are specifically ablated by genetic or other tools, does it affect the development of gut endoderm or particular tissues? The authors should clarify the biological significance of these cells.
- 6) The transcriptional identities of mCherry+/GFP+ dual-labeled cells should be compared with the GFP+ embryonic cells in the gut endoderm and mCherry+ extraembryonic cells in the yolk sac, not just with the mCherry+ cells in the gut endoderm.

Minor points:

- 1) The basis for sorting of E6.5-Epi, E9.5-DE, E6.5-emVE, and E9.5-emVE needs to be clearly described one by one.
- 2) In lines 234-244 and Fig. 4f, the authors mentioned that "the retention of the extraembryonic DNA methylation landscape in emVE cells closely mirrors the methylation pattern in most human cancer types", but there is no explanation and discussion about this statement in the manuscript.

**Although we cannot publish your paper, it may be appropriate for another journal in the Nature Portfolio. If you wish to explore the journals and transfer your manuscript please use our manuscript transfer portal. You will not have to re-supply manuscript metadata and files, but please note that this link can only be used once and remains active until used. For more information, please see our manuscript transfer FAQ page.

Note that any decision to opt in to In Review at the original journal is not sent to the receiving journal on transfer. You can opt in to In Review at receiving journals that support this service by choosing to modify your manuscript on transfer. In Review is available for primary research manuscript types only.

**For Nature Portfolio general information and news for authors, see <http://npg.nature.com/authors>.

Author Rebuttal to Initial comments

We would like to thank the reviewers for their helpful and constructive feedback. We now provide multiple additional experiments and analyses to address all points raised. A detailed point-by-point response is provided below. In particular, we have emphasized the following overall areas of initial concern:

1) Independent validation of labeling/tracing strategy

We used published and newly generated scRNA-seq data, DNA methylation signatures as well as *in vitro/in vivo* experiments with p53 mutated cells to independently demonstrate the precise labeling and tracking of extraembryonic cells in the embryonic gut.

2) Confirm efferocytosis/phagocytosis

We used Lysotracker to label acidic compartments, such as phagolysosomes, and performed live imaging of ex utero cultured embryos. This experiment revealed mCherry foci, representing extraembryonic cell remnants, become positive for Lysotracker signal, which supports the clearance of mCherry fragments via phagocytosis.

3) Mechanistic insights into the cell death

In order to provide independent validation regarding our observations of extraembryonic cell clearance, we generated wild type scRNA-seq data of the gastrointestinal tract at E13.5, which confirmed that virtually no cells with extraembryonic origin (based on *Rhox5* and *Trap1a* expression) could be detected in the wild type. In contrast, the gastrointestinal tract of fully p53-mutant embryos contained surviving gut cells of extraembryonic origin, which showed that the elimination of these cells in the wild type context relies on a p53-dependent mechanism. Additionally, we made use of a publicly available scRNA-seq data set spanning E9.5 to E15.5 and found that while cells expressing the extraembryonic marker genes *Rhox5* and *Trap1a* are present at E9.5 in the intestine, they could no longer be detected by E11.5.

4) Effects of the distinct DNA methylation on gene expression

To strengthen the comparison of transcriptome and DNA methylation between the embryonic and extraembryonic gut lineage, we replaced our previous data with an improved cohort that contains additional control tissues as well as E13.5 (small intestine and colon) as a new time point and furthermore includes p53-mutant extraembryonic gut cells. For each tissue at E6.5 and E9.5, RNA-seq as well as whole genome bisulfite sequencing (WGBS) data for improved DNA methylation resolution was generated. Specifically, we added the E9.5 yolk sac endoderm as well as the part of the E6.5 extraembryonic endoderm that gives rise to it. Additionally, we sorted gut cells with embryonic and extraembryonic origin at E9.5 from two different regions of the gut endoderm (midgut and hindgut) and thus generated more replicates from distinct localizations within the embryo. To account for the newly generated data sets, our nomenclature changed throughout the paper as can be seen in Fig. 3a:

The previously used term E6.5 emVE is now exEndo1 (while exEndo2 refers to the progenitor giving rise to the yolk sac endoderm). Additionally, the E9.5 emVE is now exGut while the

E9.5 DE is now emGut (samples generated for midgut and hindgut). We believe that the new nomenclature together with our new cohort leads to an improved understanding of the differentiation and transcriptional dynamics.

Lastly, as part of the revision, we newly generated 44 RNA-seq, 32 RRBS, seven WGBS and two scRNA-seq data sets together with several new analyses as outlined below.

Reviewers' comments:

Reviewer #1 (Remarks to the Author):

This is an interesting and exciting new manuscript by Batki et al investigating the nature and persistence of extraembryonic cells to the developing mouse embryonic gut. Using a novel lineage tracing method, the authors demonstrate the presence and localization of extraembryonic cells. Because of their permanent label, the authors are then able to isolate them and using 10x RNAseq and Smart-Seq2 show that they have transcriptional signatures consistent with their origin. Excitingly, they also demonstrate that these extraembryonic cells are no longer present in the mouse gut by E13.5, and hypothesize that they are cleared by neighboring embryonic epithelial cells. Overall, the paper is highly significant and introduces new tools to answer an old and intriguing question of how extraembryonic and embryonic tissues mingle during development. And while the cell clearance hypothesis is exciting and there is evidence of similar phenomena happening in others systems, there is no direct evidence of mCherry+ cells being engulfed by GFP+ cells. The movies provided clearly show cell death of extraembryonic cells. However, whether they are actively being engulfed/phagocytosed by neighboring embryonic cells is unclear.

To make this point, more evidence needs to be shown. This could include incorporation of an acidic dye that labels acidic compartments in these studies (e.g., lysotracker or neutral red), higher resolution imaging to show engulfment, interactions between these cells in culture, or EM to show cell debris within cells. I appreciate none of these methods are easy or straightforward, but as the data stands now with regards to this point, it is unclear if clearance is actively happening. Alternatively, blocking cell death with caspase inhibitors and then seeing a perdurance of the extraembryonic signature would add more weight to this claim.

Response: We appreciate the helpful and constructive comments. While the cell death of extraembryonic cells has been demonstrated by our data, the reviewer is correct that clearing and phagocytosis performed by embryonic cells was so far only indirectly shown through flow cytometry and imaging. Specifically, we reported embryonic cells with remnants of the eliminated extraembryonic cells (dual+ cells: cells with high GFP intensity and varying level of mCherry signal).

Action taken: To explore the clearance further, we used Lysotracker—as suggested by the reviewer—to label acidic compartments, such as phagolysosomes. We performed live imaging of ex utero cultured embryos and found that mCherry foci, representing extraembryonic cell remnants, become positive for Lysotracker signal. This result confirmed that extraembryonic remnants are indeed taken up and then localize in acidic compartments, which happens during phagocytosis (**new Ext. Data Fig. 5b,c, Supplementary video 3**).

We also provide high resolution images acquired by light sheet microscopy, where we used the membrane marker, E-CADHERIN, to show that extraembryonic remnants (represented by mCherry foci) localize inside embryonic cells (GFP+) (**new Ext. Data Fig. 5a**).

We very much agree with the reviewer that blocking cell death and exploring the outcome is of major interest. To address this, we zygotically knocked out p53 and showed that this enables the survival of extraembryonic cells and establishes a p53-dependent elimination mechanism (**new Fig. 5c,d, Ext. Data Fig. 11,12**). We then performed transcriptome and DNA methylome analysis of the surviving p53-mutant extraembryonic cells and found that the lineage origin-specific signatures, including the global DNA methylation landscape, are still preserved (**new Fig. 5e,f, Ext. Data Fig. 14**). The latter is rather striking and the analysis highlights that these cells not only persist but continue to differentiate. Lastly, we also show that the survival of p53-mutant extraembryonic cells extends also to the *in vitro* culture of gut assemblies generated from sorted cells (**new Ext. Data Fig. 13**).

Reviewer #2 (Remarks to the Author):

This manuscript assesses the fate of visceral endoderm cells which integrate into the definitive endoderm of developing embryos and assume a very similar transcriptional signature. Using diploid complementation, the authors confirm that visceral endoderm cells are integrated in definitive endoderm during gastrulation and that they can be detected in the gut tube until around E12.5. The authors confirm the overall similar transcriptome of VE vs DE cells, but detect a few genes which are differentially expressed. VE upregulated genes are germline genes and relate to activation of p53 mediated apoptosis. The latter is in line with their observation that VE cells stain positive for Caspase and are fragmented and taken up by neighboring cells. The authors further characterize DNA methylation differences and observe that VE cells are generally hypermethylated compared to DE cells. This might also explain the upregulation of germline specific genes which are known to be sensitive to loss of DNA methylation. The novel findings of this manuscript are that VE cells are not lost during later embryonic development and that these cells largely maintain their early embryonic DNA methylation signature, although they assume a largely DE-like transcriptome.

We would like to thank the Reviewer for the detailed summary and helpful comments to which we provide a detailed point-by-point response below.

A potential caveat of this study is the use of the diploid complementation system for all assays. It would be important to validate if similar transcriptional changes (or DNA methylation changes, or loss of VE cells) are seen in normally developing embryos (eventually combining lineage tracing and/or use of their newly identified VE specific genes).

Response: Given the central role in our study we have of course extensively validated the labeling strategy and further extended this in the revised manuscript. Nonetheless, we agree that it is valuable to provide validation of the findings, independent of our dual-color lineage tracing strategy. Below we outline the four different ways in which we confirmed our core findings.

Action taken: First, we obtained and analyzed data from a previously published scRNA-seq atlas of the mouse gastrointestinal tract between E9.5 and E15.5 (Zhao et al. 2022, *Cell Reports*). Using extraembryonic marker genes established at early organogenesis (*Rhox5* and

Trap1a), we found extraembryonic gut cells at E9.5 as expected, while such cells were not detectable at the later stages (**new Fig. 2f, Ext. Data Fig. 6a**). This is in agreement with our findings using the lineage tracing strategy showing that extraembryonic cells are eliminated by midgestation. Second, we also performed our own scRNA-seq analysis of organs from wild type E13.5 embryos, and found virtually no extraembryonic cells (**new Fig. 5a-c, Ext. Data Fig. 11**). Third, when we analyzed organs from our p53-mutant embryos (without the use of the dual-color lineage tracer) we also found persisting extraembryonic cells (**new Fig. 5a-c, Ext. Data Fig. 11**). Fourth, both *in vitro* and *in vivo* use of the original dual-color lineage tracing strategy combined with extraembryonic-specific p53 knockout further demonstrates that our diploid complementation strategy is a suitable tool to track and study extraembryonic cells within the developing gut (**new Fig. 5d, Ext. Data Fig. 12,13**).

Lastly, to further validate the transcriptional signatures identified by our dual-color lineage tracing, we analyzed available scRNA-seq datasets of the developing gut from two independent studies at two embryonic stages (E8.75 and E9.5, Nowotschin et al. 2019, *Nature* and Zhao et al. 2022, *Cell Reports*, respectively). Both origin-related signatures identified by our lineage tracing—germline program-related in exGut high category and axonogenesis-related in exGut low category—were recapitulated in the two published datasets (**new Ext. Data Fig. 9e**). Furthermore, we also found the cell death-related transcriptional signature (differentially expressed p53-target genes) in extraembryonic cells in the published datasets (**new Ext. Data Fig. 11b**).

The authors could show that the mCherry pos. cells are able to form normally looking embryos, however, there are also transcriptional signatures in DE cells (Ex Fig. 6E, axonal development, etc.) which do not seem to be very physiologic.

Response: We appreciate the reviewers careful assessment of our results. First, to further expand on the mCherry positive cells, we did an additional control experiment to show that the mCherry+ reporter is not silenced in the colon and small intestine at E13.5. To this end we generated and analyzed fully mCherry embryos, which we found to develop normally and maintain mCherry expression throughout (**new Ext. Data Fig. 7f-h**). Moreover, mCherry+ mice have been breeding, and we have obtained viable and fertile progeny, further proving that mCherry+ cells are physiologically normal.

As correctly pointed out by the reviewer, we identified transcriptional signatures in embryonic cells that are related to axonogenesis and synaptic membrane. While this may not seem physiologic at first, we would like to point out that genes related to these terms are generally involved in cell-cell communication and adhesion, and have known function in both neural and non-neural contexts, including various epithelial tissues (Beamish et al. 2018, *Cold Spring Harb. Perspect. Biol.*). We therefore postulate that such differences might be relevant for cell-cell interaction and ultimately contributing to the distinct fate of embryonic and extraembryonic cells. Moreover, we note that the axonogenesis-related signatures were also identified in published scRNA-seq datasets (Nowotschin et al. 2019, *Nature* and Zhao et al. 2022, *Cell Reports*) (**new Ext. Data Fig. 9e**).

In addition, resolving the mechanistic basis for either VE cell death or for the DNA methylation independent changes in the VE transcriptome would considerably strengthen the manuscript.

Response: We fully agree with the reviewer and therefore the bulk of our energy during the past 8 month was aimed at thoroughly addressing this point. We are thrilled to share some really exciting new data as summarized below and presented in **new Figure 5**.

Action taken: Based on the literature and our transcriptional analysis, p53 was an obvious candidate for the programmed cell death (**new Ext. Data Fig.11**). To investigate its role, we used zygotic Cas9-mediated disruption to generate p53 mutant embryos (independent of the dual lineage tracing strategy) and extraembryonic lineage-specific p53 knockout embryos (using the lineage tracing). We then performed a number of FACS, imaging and molecular characterizations. Briefly, we now demonstrated that p53 is required for the elimination of extraembryonic cells and strikingly the p53-mutant extraembryonic cells can survive (**new Fig. 5a-d, Ext. Data Fig. 11-12**). Moreover, we also recapitulated the p53 knockout-dependent survival and proliferation of extraembryonic gut cells *in vitro* (**new Ext. Data Fig. 13**). As would be expected, the DNA-methylation independent p53-related transcriptional signature is diminished in the p53-mutant extraembryonic gut cells and p53-target genes are no longer upregulated in p53-mutant extraembryonic gut cells. (**new Ext. Data Fig. 11i**).

We then performed transcriptome and DNA methylome analysis of the surviving p53-mutant extraembryonic cells and found that the lineage origin-specific signatures, including the global DNA methylation landscape, are still preserved (**new Fig. 5e,f, Ext. Data Fig. 14**). The latter is rather striking and the analysis highlights that these cells not only persist but continue to differentiate.

Minor comments:

(1) The expression changes between VE and DE (Fig 3B) could better be shown as mean expression vs. fold change plot with significant genes colored. Supplementary tables with gene expression data (TPM) need to be added.

Action taken: We now provide the plots showing the mean expression versus fold change, where the significant genes are colored (**new Ext. Data Fig. 9a**). We also added **new Supplementary Table 3** that summarizes the gene expression data (TPM).

(2) The heatmaps in Fig. 3D,E, Ex. Fig. 6F could be reordered to better be able to compare VE vs DE at the two developmental stages (E6.5 Epi, VE, E9.5 DE, VE).

Action taken: We agree with the reviewer that the suggested order makes it easier to compare gene expression between cells of distinct origins at each developmental stage. However, our aim was to contrast the expression profiles of the two lineages (embryonic vs extraembryonic) by emphasizing the similarity of samples associated with them, specifically for the exGut high genes. Nevertheless, we provide, here, for the Reviewer, the new Fig. 3e (top) and a Reviewer Figure with the changed order for comparison (bottom).

New Figure 3e

Reviewer Figure - order suggested by the reviewer

(3) Is the overall reduced DNA methylation in VE related to expression or protein abundance changes in the DNA methylation machinery?

Response: The short answer is that there is no obvious association based on expression level.

Action taken: Here, we provide a **Reviewer figure** that shows the gene expression level of the DNA methyltransferases, TET enzymes and PRC2 components. At E6.5, Dnmt3a and Dnmt3b are higher expressed in the epiblast compared to the extraembryonic endoderm. The levels drop, however, in the gut cells and become extremely similar between exGut and emGut cells. The maintenance methyltransferase Dnmt1 shows similar expression levels across all tissues at both stages. Overall, there seems to be no strong difference of epigenetic regulator expression between gut cells with embryonic and extraembryonic origin. We have added this to the revised manuscript (new **Ext. Data Fig. 10h**), and provide it here for the Reviewer's convenience.

(4) line 244: "which also closely mirrors the methylation pattern in most human cancer types" If there is any relation to cancer development it should be explained in the discussion section.

Response: This statement was supported by some of our prior work (Smith et al. 2017, *Nature*, and Weigert, Hetzel et al. 2023, *Nature Cell Biology*). However, given that this is not the primary focus of this study, we have removed the sentence.

We have however added a brief new discussion point on *Trap1a* and cancer/testis antigens on page 9: *"Intriguingly, Trap1a (tumor rejection antigen P1A) was identified in mice as an endogenous gene that shows little to no expression in normal cells³⁹. More generally, cancer/testis antigens are typically restricted to the germline and extraembryonic tissues, and are aberrantly expressed in various cancers where they are often immunogenic⁴⁰. A large fraction of these genes is located in clusters on the X chromosome and sensitive to loss of DNA methylation, which are characteristics shared with many of the extraembryonic gut marker genes."*

Reviewer #3 (Remarks to the Author):

Synopsis:

In this study, Batki et al. developed a two-color lineage tracing strategy to track and isolate the extraembryonic cell-derived gut endodermal cells. Through this strategy, they found that gut cells with extraembryonic origin are subjected to programmed cell death and uptake by embryonic gut cells, subsequently being eliminated at E13.5. Moreover, they identified that these extraembryonic gut cells acquire a similar gene expression profile to embryonic gut cells, but retain the extraembryonic DNA methylation landscape. This study claims the developmental fate of some extraembryonic cells in the gut endoderm, but the data are far from convincing. Some key experimental designs and data analyses are confusing. Furthermore, the biological significance of these extraembryonic-derived gut endodermal cells remains unclear, making this study less meaningful.

Response: We appreciate the feedback and have used it to revise and fully rewrite essential experimental parts. We have added several new experiments and additional analysis as well as independent validations. We hope that these clarification and additional controls for the assay combined with our exciting new p53 knockout data address all the concerns raised by the Reviewer.

Major points:

1) Traditional morula aggregation using tetraploid host embryos and diploid ESCs can generate an almost entirely ESC-derived embryo. That is because tetraploid cells fail to contribute to the embryo proper and are only limited to the extraembryonic yolk sac and trophoblasts. In this study, if diploid host embryos never subjected to any treatment are used, how to rule out the contributions of mCherry⁺ diploid morula cells to the embryo proper? The authors claim the mCherry⁺ cells as extraembryonic cells, which is unstringent at all. This invalid claim shakes the whole study.

Response: We appreciate the points raised here and can assure the reviewer that we have thoroughly validated this point. Our lab has used diploid and tetraploid injection as well as aggregation for over 15 years now and our PI acquired the expertise already during his training with Rudolf Jaenisch. We have now undertaken a number of independent experimental approaches to carefully address all of the reviewers' concerns/questions and as such improve the presentation in the revised manuscript. We have also fully rewritten the first section describing our approach.

Action taken: First, by taking advantage of the fluorescent reporters, we performed a systematic comparison of complementation assays, including diploid and tetraploid morula aggregation as well as diploid blastocyst injection (**new Ext. Data Fig. 2**). This confirmed that our diploid complementation assay with early-stage blastocysts results in comparable outcome to the conventional tetraploid morula aggregation (**Ext. Data Fig. 2a-c**). In contrast, the diploid blastocysts injection, as expected, produced true chimeras (**Ext. Data Fig. 2d-f**). The injection-derived chimeras contain mCherry⁺ cells throughout the embryos, which is in striking contrast to the gut-specific mCherry signal observed in embryos generated by the diploid aggregation (**Fig. 1e, Ext. Data Fig. 3c**).

Second, our extraembryonic cells preserve a unique DNA methylation signature. Had this been reset/reprogrammed, it would have been more difficult to assess the origin, but given the striking epigenetic memory, we have a clear molecular signature that confirms the extraembryonic origin of the cells. As such we can use the DNA methylation sequencing data generated within this study as additional support that mCherry⁺ cells are of extraembryonic origin (**new Fig. 4b,c, Ext. Data Fig. 10d,e, 14**).

Third, we analyzed data from a previously published scRNA-seq atlas of the mouse gastrointestinal tract between E9.5 and E15.5 (Zhao et al. 2022, *Cell Reports*). Using extraembryonic marker genes established at early organogenesis (*Rhox5* and *Trap1a*), we found extraembryonic gut cells at E9.5 as expected, while such cells were not detectable at the later stages (**new Fig. 2f, Ext. Data Fig. 6a**). This is in agreement with our findings using the lineage tracing strategy showing that extraembryonic cells are present and subsequently eliminated by midgestation.

Fourth, we also generated our own wild type scRNA-seq data of the gastrointestinal tract at E13.5, which confirmed the absence of extraembryonic gut cells (based on *Rhox5* and *Trap1a* expression). When we analyzed organs from our p53-mutant embryos (without the use of the dual-color lineage tracer), we found persisting extraembryonic cells (**new Fig. 5a-c, Ext. Data Fig. 11**). This demonstrated that p53 is required for the elimination of extraembryonic cells.

Fifth, we were able to recapitulate the p53-dependent cell death both *in vitro* and *in vivo*, using the dual-color lineage tracing strategy combined with extraembryonic-specific p53 knockout (**new Fig. 5d, Ext. Data Fig. 12,13**). This further demonstrates that our diploid

complementation strategy is a suitable tool to track and study extraembryonic cells within the developing gut.

Lastly, independent validation for the transcriptional signatures were added (**new Ext. Data Fig. 9e, 11b**).

Taken together, we believe all these results support the validity of our findings and the use of the dual-color lineage-tracing strategy.

2) The authors conclude that the mCherry+/GFP+ dual-labeled cells are resulted from the mCherry+ cell death and subsequent GFP+ cell uptake. This should be a phenomenon termed efferocytosis.

Response: We appreciate this specific comment. In our reading of the literature, efferocytosis is the effective clearance of apoptotic cells by professional and non-professional phagocytes. While we kept the terminology broader, referring to the clearing of extraembryonic remnants as phagocytosis, we agree with the reviewer that efferocytosis is an appropriate term describing the observed phenomenon and have adjusted the text on page 4.

Although immunofluorescence for Caspase3 and live imaging showed the programmed death of mCherry+ cells, there is no direct evidence showing the ingestion of mCherry+ cell remnants by the GFP+ cells. Efferocytosis assays such FACS efferocytosis assay and DNase II signals should be performed.

Response: We thank the reviewer for this important feedback. Indeed, while cell death of extraembryonic cells has been clearly demonstrated by our data, clearing and phagocytosis performed by embryonic cells was only indirectly shown by flow cytometry, where we detected embryonic cells with remnants of the eliminated extraembryonic cells (dual+ cells: cells with high GFP intensity and varying level of mCherry signal).

Action taken: To more directly show ingestion of mCherry+ cells remnants, we used Lysotracker, as suggested by Reviewer 1, that labels acidic compartments, such as phagolysosomes. We then performed live imaging of *ex utero* cultured embryos and found that mCherry foci—representing extraembryonic cell remnants—become positive for the Lysotracker signal (**new Ext. Data Fig. 5b,c, new Supplementary video 3**). In our opinion this result further strengthens the uptake of the extraembryonic remnants by GFP+ cells and shows they localize in acidic compartments, which is known to occur during phagocytosis.

Moreover, do the majority of or all the dual-labeled cells come from efferocytosis? How about other reasons such as cell fusion?

Response: That is a good point to double check and in line with the reviewers question, we reevaluated our data and concluded that cell fusion is not fitting our observations. In case of cell fusion, both mCherry and GFP would be encoded in the same cell. This would result in an intact cell which contains GFP and mCherry distributed in the cell without the foci-like localization that we observe. Combined with all the other molecular data, the p53 knockout and the newly added lysotracker results, we are confident that we are describing programmed cell death followed by efferocytosis.

3) If the mCherry+/GFP+ dual-labeled cells mainly come from efferocytosis, they are

embryonic cells, not from the extraembryonic origin. The use of these dual-labeled cells as extraembryonic gut cells to analyze differential gene expression between emVE and DE cells is illogical. Based on the prerequisite that all the mCherry+ cells are derived from the extraembryonic origin (which is not true as pointed out in the Point 1)), only mCherry single-positive cells should be used to compare with the GFP-labeled cells.

Response: We apologize if our prior version was not entirely clear. The reviewer is correct that the mCherry+/GFP+ dual-labeled cells are embryonic cells (as indicated by the high GFP intensity and verified by the lack of extraembryonic marker genes in **Fig. 1h**, in contrast to the mCherry+ extraembryonic gut cells).

In the differential gene expression analysis, we use mCherry single-positive cells as extraembryonic cells. As a stage and position matched embryonic counterpart, the use GFP single-positive gut cells was unattainable, because no unique surface marker is available that is specific to the epithelial gut endoderm. EPCAM—an epithelial surface marker—also labels surface ectoderm and primordial germ cells (PGCs), which are also both positive for GFP.

As a reasonable compromise, we decided to utilize the dual+ cells as the majority of these cells are embryonic gut cells, which are positive for EPCAM (**Fig. 1**). Thus, in addition to EPCAM+ single mCherry+ cells, which are the extraembryonic gut cells (as discussed above under Point 1), we sorted EPCAM+ dual+ cells as the embryonic gut cells, with low mCherry intensities (in order to minimize any extraembryonic remnant, also note, if anything that would reduce our true differential signature rather than giving false signals). Even though these dual+ cells might contain remnants of the extraembryonic cells that they ingested, this signal is much lower than the transcriptional signal of intact mCherry cells. Given the strong cell type-specific transcriptional effect that would bias our analysis if we would sort EPCAM+ single GFP+ cells that are heavily contaminated by surface ectoderm and PGCs, the dual+ cells represent the best stage and tissue-matched embryonic control for our mCherry+ cells.

Action taken: To better explain the rationale, we adjusted the text on **page 6** as follows:

*“We isolated E6.5 epiblast as well as distal and proximal extraembryonic endoderm (exEndo 1 and 2) to analyze the three E6.5 progenitor populations that differentiate into embryonic gut cells, extraembryonic gut cells and extraembryonic yolk sac, respectively (**Fig. 3a**). At E9.5, we sorted yolk sac endoderm cells (YsEndo) and gut cells from the posterior half of embryos (midgut and hindgut), where the majority of the remaining mCherry+ extraembryonic gut cells are found at this developmental stage (**Ext. Data Fig. 8a,b, Supplementary Table 2**). This gut cell isolation approach should minimize transcriptional differences due to spatial localization along the anterior-posterior axis. For our differential gene expression analysis, we selected dual+ embryonic gut cells (EPCAM-positive) as a closely matched (stage and position) embryonic control (**Ext. Data Fig. 8b**, see Methods for details on the differential gene expression analysis and selection of reference cell types).”*

We then provide more details in the Methods section on **page 29 and 34**:

Page 29: *“For the low-input bulk RNA-seq, RRBS and WGBS analysis of gut cells from E9.5 wild type lineage-traced embryos (see below), EPCAM-positive dual+ cells were sorted with low mCherry+ signal intensity as embryonic gut control for the mCherry+ extraembryonic gut. (Note: at E9.5, the EPCAM-positive GFP-only cell population contains surface ectoderm and primordial germ cells in addition to embryonic gut endoderm cells, thus is not suitable as an embryonic gut endoderm control. As established by our scRNA-seq, dual+ cells are of embryonic origin, and EPCAM+ dual+ cells correspond to gut endoderm.)”*

Page 34: “Differentially expressed genes between E9.5 exGut (test) and emGut (control) were determined using DESeq2 (version 1.32.0, parameters: minReplicatesForReplace = 10)⁵². As explained in the FACS section, dual+ cells were used as emGut control due to the contamination of E9.5 GFP+ EPCAM+ cells with surface ectoderm and primordial germ cells, which would have a strong effect on gene expression analysis. Dual+ cells have been shown by the scRNA analysis to be of embryonic, mostly endoderm origin with extraembryonic remnants. In order to reduce the contamination of extraembryonic fragments, only dual+ cells with low mCherry signal were sorted (see FACS section). Even though the dual+ cells can contain some extraembryonic transcript remnants, this will only weaken the true differential signal we detect between exGut and emGut cells and not introduce wrong, cell type-specific signatures such as present within the single GFP+ EPCAM+ population.”

4) This study shows that extraembryonic emVE exhibits global hypomethylation and local hypermethylation compared to the embryonic DE. That is a common phenomenon in epigenetics, because any two cell populations of different origins may exhibit differences in global and local methylation levels. The authors should provide more about DNA methylation changes and corresponding sets of modified genes during emVE and DE development, rather than this simple comparison. Such as, is there a trend in the global or local methylation level of emVE at E9.5, E10, E11, and E12? Whether the elimination of mCherry+ cells at E13.5 directly related to methylation? The functions of the DNA methylation changes should be experimentally addressed.

Response: Here we need to briefly correct a misunderstanding. Of course, the reviewer is correct that any two cell types will have a distinct DNA methylation signature. However, the differences that we discuss here between embryonic and extraembryonic epigenome are fundamentally different. More than 80% of the genome is differentially methylated including unusual (non-canonical) hypermethylation of CpG islands that is not found in embryonic cells (Smith et al. 2017, *Nature* and Zhang et al. 2018, *Nature Genetics*).

As extraembryonic gut cells are rapidly eliminated between E9.5 and E13.5 (**new Fig. 2f**), we could not assess DNA methylation dynamics in this time window. However, our newly generated p53-mutant extraembryonic gut cell data created that opportunity and we now report these results in **new Figure 5**.

Action taken: Using our newly generated WGBS and RNA-seq/RRBS data we added several new analyses on DNA methylation and its relation to gene expression. Specifically, we generated RNA-seq data of E6.5 epiblast and extraembryonic progenitors (exEndo 1 and exEndo 2), the E9.5 gut cells with distinct origins (midgut and hindgut) and the yolk sac endoderm (**new Fig. 3a**). We also generated corresponding whole-genome bisulfite sequencing (WGBS) datasets for all samples. To put the extraembryonic DNA methylation changes both globally and at hypermethylated CGIs into context, we provide a comparison of the DNA methylation level of E6.5 progenitors, E9.5 gut and yolk sac endoderm as well as a selection of publicly available adult somatic tissues here. Even though there are slight differences between any two cell types, the strong global depletion and simultaneous gain of methylation at CGIs hypermethylated in the exEndo 1 is only present in cell types with extraembryonic origin. The global loss is most pronounced in the exEndo 2 and YsEndo but also clearly present in the exEndo 1 and exGut cells.

Additionally, we generated RRBS data for E13.5 embryonic gut cells and extraembryonic gut cells with p53 mutation matching the newly generated E13.5 RNA-seq data. Surviving p53-mutant extraembryonic gut cells at E13.5 show the same lineage origin-specific global and local DNA methylation signatures as found at E9.5 (new Fig. 5f, Ext. Data Fig. 14).

5) What is the biological roles of these so-called extraembryonic-derived gut endodermal cells? If these cells are specifically ablated by genetic or other tools, does it affect the development of gut endoderm or particular tissues? The authors should clarify the biological significance of these cells.

Response: The reviewer raises of course a fundamental and very interesting question that has been unanswered ever since the initial study of Anna-Katerina Hadjantonakis (Kwon et al. 2008, *Developmental Cell*). Unfortunately, that is a very complex question and beyond the scope of our current study. However, our work strongly supports the merit of investigating the transient function of these cells in the future.

In this study, we answered three equally important fundamental questions about the extraembryonic gut cells in the mouse embryo. What is the fate of these cells (how long are they present), what is the molecular state (transcriptional and epigenetic) and after having answered the first, can we make them survive longer? All of these have been answered and we find the results convincing and extremely interesting.

6) The transcriptional identities of mCherry+/GFP+ dual-labeled cells should be compared with the GFP+ embryonic cells in the gut endoderm and mCherry+ extraembryonic cells in the yolk sac, not just with the mCherry+ cells in the gut endoderm.

Response: We agree with the reviewer that these comparisons are important. As summarized above, the isolation of GFP single-positive gut cells at E9.5 was not feasible due to shared surface markers of ectoderm and gut cells (Sarrach et al. 2018, *Scientific Reports*). Additionally, EPCAM+ primordial germ cells migrate along the gut tube, which hinders sorting cells from manually isolated gut using EPCAM+ as endoderm marker (Anderson et al. 2000, *Mechanisms of development*). However, at E13.5, EPCAM can be used as a surface marker to sort endoderm cells from the dissected colon and small intestine as no contaminating cell type with the same surface marker exists within these tissues.

Action taken: To address the reviewer points, we generated RNA-seq, RRBS and WGBS for E9.5 yolk sac endoderm as a comparison to the extraembryonic gut cells. We also generated RNA-seq and WGBS for the E6.5 exEndo 2 progenitor that gives rise to the E9.5 yolk sac endoderm to complement our E6.5 and E9.5 cohort. At E13.5, we generated RNA-seq and RRBS from colon and small intestine endoderm cells with embryonic and extraembryonic origin (extraembryonic cells are p53-mutated). Here, the embryonic control data sets consist of GFP single-positive cells.

We found that cells with embryonic and extraembryonic origin from the same tissue are overall transcriptionally very similar to each other but distinct compared to the same lineage across tissues (**new Fig. 3b, Fig. 5e**). Specifically, the yolk sac endoderm differs drastically from the gut-related samples based on the transcriptome, even though the respective progenitor cell types E6.5 exEndo 1 and 2 are extremely similar to each other (**new Fig. 3b**). In contrast to the overall levels, DNA methylation-dependent genes expressed in the extraembryonic compared to the embryonic gut (including germline genes) are expressed across all extraembryonic lineages including the yolk sac (**new Fig. 3e, Ext. Data. Fig. 10f and 14d**). This is in line with the DNA methylation memory that represents a shared characteristic of cells with extraembryonic origin (**new Fig. 4b-d and 5f, Ext. Data. Fig. 10d and e, 14**).

Minor points:

1) The basis for sorting of E6.5-Epi, E9.5-DE, E6.5-emVE, and E9.5-emVE needs to be clearly described one by one.

Response: We agree that the justification for selecting and sorting the four cell types was insufficiently explained before. The E6.5 tissues were manually isolated, and no sorting was involved, while samples from E9.5 and E13.5 stages were sorted.

Action taken: We now provided a better justification in the text (page 5) and added a schematic that clarifies the developmental trajectories that lead to the selection of the more extensive new cohort (**new Fig. 3a**). Additionally, we provided individual sections for each post-implantation stage and more details for the sorting strategy underlying our sequenced samples in the Methods.

2) In lines 234-244 and Fig. 4f, the authors mentioned that "the retention of the extraembryonic DNA methylation landscape in emVE cells closely mirrors the methylation pattern in most human cancer types", but there is no explanation and discussion about this statement in the manuscript.

Response: This statement was supported by some of our prior work (Smith et al. 2017, *Nature*, and Weigert, Hetzel et al. 2023, *Nature Cell Biology*). However, given that this is not the primary focus of this study, we have removed this point and focused on the exciting developmental aspects of the study.

Decision Letter, first revision:

Our ref: NCB-LE50736A-Z

8th March 2024

Dear Alex,

Thank you for submitting your revised manuscript "p53-dependent elimination of extraembryonic cells from the gut" (NCB-LE50736A-Z) and also for your patience as we navigated the peer review process. As you know, while reviewers #1 and #2 signed off, reviewer #3 continued to raise concerns in this second round of review, which is why we asked for your response to their comments. We shared your responses with them, but they were still not convinced that their concerns were addressed (please see "ADDITIONAL COMMENTS TO THE AUTHORS' RESPONSE TO THE POINTS ABOVE" in their report below). As it is our common policy to try and provide as much of a balanced and fair review process as possible and to ensure we had an additional view on the issues raised, we sought to obtain advice from a fourth reviewer. This fourth reviewer was provided with all the information regarding your manuscript history and was asked to avoid raising any new points, but rather focus on the persisting issues raised by reviewer #3 and your responses to them.

Therefore, your manuscript has now been seen by the original referees, plus an additional referee (reviewer #4) who is an expert in imaging and development. All the reviewer comments can be found below. The overall reviewer view, following the advice obtained by reviewer #4 as well, is that the paper has improved in revision, and therefore we'll be happy in principle to publish it in Nature Cell Biology, pending minor revisions to satisfy the referees' final requests and to comply with our editorial and formatting guidelines.

If the current version of your manuscript is in a PDF format, please email us a copy of the file in an editable format (Microsoft Word or LaTeX)-- we cannot proceed with PDFs at this stage.

We are now performing detailed checks on your paper and will send you a checklist detailing our editorial and formatting requirements in about 1-2 weeks. Please note that, when the time comes for you to revise your manuscript per our list, we will also expect you to textually address the remaining points #1 and #3 offered by reviewer #3, as suggested by reviewer #4. Please do not upload the final materials and make any revisions until you receive this additional information from us.

Thank you again for your interest in Nature Cell Biology. Please do not hesitate to contact me if you have any questions.

Best regards,
Stelios

Stylianos Lefkopoulos, PhD

He/him/his
Senior Editor, Nature Cell Biology
Springer Nature
Heidelberger Platz 3, 14197 Berlin, Germany

E-mail: stylianos.lefkopoulos@springernature.com
Twitter: @s_lefkopoulos
LinkedIn: [linkedin.com/in/stylianos-lefkopoulos-81b007a0](https://www.linkedin.com/in/stylianos-lefkopoulos-81b007a0)

Reviewer #1 (Remarks to the Author):

This is an appealed manuscript by Meissner et al. investigating the nature and persistence of extraembryonic cells in the mouse. In the previous round of review the data was clear and the hypothesis interesting. However, there were no studies to directly demonstrate that neighboring cells phagocytosed these dying cells. In this revised manuscripts, the authors have added significantly more data to support this claim, including some in vivo imaging experiments using lysotracker as well as p53 null animals. These studies have satisfied my concerns and allow the authors to claim that phagocytosis is the mechanism at play.

Reviewer #2 (Remarks to the Author):

The authors have incorporated a multitude of extra datasets and analyses, effectively addressing all of the issues I raised.

Reviewer #3 (Remarks to the Author):

In the revised manuscript, the authors have taken active steps to incorporate new data to improve their work. However, some pivotal concerns still remain to be effectively addressed, which harm the overall quality, stringency, and impact of the study for Nature Cell Biology.

Major concerns:

- 1) Using the diploid morula aggregation to track and isolate the extraembryonic gut cells lacks the necessary stringency, which remains a major concern. The authors have not directly responded to the point whether the mCherry+ diploid morula cells contribute to the embryo proper yet.
- 1a) In the fluorescent images of the E9.5 embryos (Fig.1b, Ext. Data Fig. 2b and 8a), apart from mCherry+ cells in the gut endoderm, weak but visible mCherry signals could also be detected in other embryonic regions. These signals indicate the presence of chimeric mCherry+ cells in non-endodermal tissue. How could the authors exclude the possibilities that these cells are of embryonic origin?
- 1b) In the differential gene expression analyses (Fig. 3c, Ext. Data Fig. 4b, 8c and 9a), the well-known extraembryonic marker genes were detected in the sorted mCherry+ cells. However, these analyses were based on pooled cell populations, which could mask cellular heterogeneity. For instance, if 20% of mCherry+ cells were embryonic and 80% were extraembryonic, the results should still show similar

expression patterns. How could the authors ensure that all the mCherry+ cells are of extraembryonic origin?

2) Since the presence and basic characteristics of extraembryonic gut cells have been reported in previous studies, the novelty of this study is limited without characterizing the biological functions of these cells. My original review had proposed the ablation of these cells with genetic tools to analyze the impact on the embryonic development. These preliminary functional experiments would be necessary to enhance the novelty of this study for Nature Cell Biology.

3) While agree the conclusion that the mCherry+ cells could be cleared by the efferocytosis of GFP+ cells, the possibility of cell fusion could not be entirely ruled out yet. In the live cell imaging (Fig. 2a-b, Ext. Data Fig. 5a-c), some of the intact mCherry+ cells could be observed overlapping with the GFP+ cells, and these cells did not show any mCherry remnant or lysotracker signal. It is essential to rigorously consider all plausible mechanisms that could result in the dual-labeled cells.

ADDITIONAL COMMENTS TO THE AUTHORS' RESPONSE TO THE POINTS ABOVE

1) The authors claim that first, all the mCherry+ cells in the chimeric embryos obtained by diploid morula aggregation are of extraembryonic origin, and second, these cells exclusively contribute to the endoderm but not other germ layers. However, based on current data, these claims remain technically unconvincing according to the following two points.

1a) In the fluorescent images referred to in my previous comments (Fig.1b, Ext. Data Fig. 2b and 8a), I do not believe these weak mCherry signals present in non-endodermal tissues are entirely attributed to autofluorescence. The authors should not distinguish autofluorescence and authentic signal as they wish. Although the authors have provided confocal images with higher resolutions (Fig. 1c, Ext. Data Fig. 1f), from my experience, these images have been subjected to background subtraction.

1b) In the scRNA-seq analyses of the mCherry+ cells (Fig. 1g, Ext. Data Fig. 4b and 4d), 10 out of the 471 mCherry+ cells (approximately 2.1%) are situated within mesodermal cell clusters. These analyses were even based on isolated gut tissues rather than whole embryos. If these experiments had been performed using whole embryos, higher proportions of mCherry+ cells should have been detected in non-endodermal tissues. The authors should perform scRNA-seq using whole chimeric embryos rather than isolated gut tissues, if they want to draw a conclusion that these mCherry+ cells are entirely in the endoderm.

2) It is true that this study answers the question of permanent or transient presence. However, on the way from the initial discovery of these extraembryonic gut cells to the understanding of their biological functions, the contribution of this study is limited and not significant enough for Nature Cell Biology. Without basic functional analyses, this study lacks sufficient impact to the field.

3) Similar to the question previously raised in the live cell imaging (Fig. 2a-b, Ext. Data Fig. 5b-c), some of the intact mCherry+ cells could still be observed overlapping with the GFP+ cells in the single optical sections (Ext. Data Fig. 3c, the upper most cells, and Ext. Data Fig. 5a). Moreover, in FACS analyses (Ext. Data Fig. 6c-d), low mCherry and high GFP intensities of the dual-labelled cells at later stages do not conclusively indicate that GFP+ cells take up mCherry remnants. The authors should quantify the number of cells exhibiting clear efferocytosis signals and those not involved in the process, while stating the possibility of other mechanisms.

Reviewer #4 (Remarks to the Author):

I have been invited to provide my opinion regarding some of the remaining points raised about this manuscript. Overall, I find that the authors have satisfactorily addressed these points in their last submission and response letter.

Briefly, they used a two-color lineage tracing strategy to track extraembryonic cells during early mouse development. They combined diploid morula aggregation, traditional tetraploid complementation, and classic blastocyst injection with high-resolution microscopy, to characterize a population of gut endoderm cells (marked by mCherry) of extraembryonic origin. They characterize the genetic and epigenetic signature of these cells and show that they represent a transient population, which is eliminated via cell death (in p53-dependent manner).

Main points:

1) Lineage contribution: There were concerns about the specificity of the diploid complementation approach. The authors provide sufficient evidence that the mCherry-positive cells only contribute to extraembryonic, and not embryonic tissues. This is supported by the gene expression analysis but also by laser scanning microscopy, where they distinguish single cells in 2D planes and show no overlap.

Based on the quality of the images, I do not have further concerns to support that all (or at least most) mCherry-positive cells contribute to extraembryonic tissues. It is true that some overlap can be seen in some examples, but this is likely a result of using simple stereoscopes vs the more careful comparison performed in their 2D planes (as outlined in the authors' responses). I suggest that the authors carefully address this point in their Results section too, as it may help further readers.

2) Function of this cell population: The authors demonstrate that these extraembryonic cells contribute to the gut but are then eliminated. I feel that further exploring the function of these cells is beyond the scope of the current study. Indeed, they may have no major long-term function and only represent a transient population. I do not consider that this diminishes the potential interest of the current findings.

3) Cell elimination mechanism: They provide sufficient mechanistic insight to support the idea that the cells are eliminated in a p53-dependent manner. I do not find sufficient evidence for cell fusion, yet I agree with some of the reviewer comments and suggest that the authors dedicate part of the text to discussing this interesting point without additional experiments.

Decision Letter, final checks:

Our ref: NCB-LE50736A-Z

21st March 2024

Dear Dr. Meissner,

Thank you for your patience as we've prepared the guidelines for final submission of your Nature Cell Biology manuscript, "p53-dependent elimination of extraembryonic cells from the gut" (NCB-LE50736A-Z). Please carefully follow the step-by-step instructions provided in the attached file, and add a response in each row of the table to indicate the changes that you have made. Ensuring that each point is addressed will help to ensure that your revised manuscript can be swiftly handed over to our production team.

In recognition of the time and expertise our reviewers provide to Nature Cell Biology's editorial process, we would like to formally acknowledge their contribution to the external peer review of your manuscript entitled "p53-dependent elimination of extraembryonic cells from the gut". For those reviewers who give their assent, we will be publishing their names alongside the published article.

Nature Cell Biology offers a Transparent Peer Review option for new original research manuscripts submitted after December 1st, 2019. As part of this initiative, we encourage our authors to support increased transparency into the peer review process by agreeing to have the reviewer comments, author rebuttal letters, and editorial decision letters published as a Supplementary item. When you submit your final files please clearly state in your cover letter whether or not you would like to participate in this initiative. Please note that failure to state your preference will result in delays in accepting your manuscript for publication.

Cover suggestions

COVER ARTWORK: We welcome submissions of artwork for consideration for our cover. For more information, please see our guide for cover artwork.

Nature Cell Biology has now transitioned to a unified Rights Collection system which will allow our Author Services team to quickly and easily collect the rights and permissions required to publish your work. Approximately 10 days after your paper is formally accepted, you will receive an email in providing you with a link to complete the grant of rights. If your paper is eligible for Open Access, our Author Services team will also be in touch regarding any additional information that may be required to arrange payment for your article.

Please note that *Nature Cell Biology* is a Transformative Journal (TJ). Authors may publish their research with us through the traditional subscription access route or make their paper immediately

open access through payment of an article-processing charge (APC). Authors will not be required to make a final decision about access to their article until it has been accepted. Find out more about Transformative Journals

Please use the following link for uploading these materials:
<https://mts-ncb.nature.com/cgi-bin/main.plex?el=A1C1CFT2A1BTru7J5A9ftdKWL0pORh4L0qYZ7Vet8gZ>

Best regards,

Kendra Donahue
Staff
Nature Cell Biology

On behalf of

Stylios Lefkopoulos, PhD
He/him/his
Senior Editor, Nature Cell Biology
Springer Nature
Heidelberger Platz 3, 14197 Berlin, Germany

E-mail: stylios.lefkopoulos@springernature.com
Twitter: [@s_lefkopoulos](https://twitter.com/s_lefkopoulos)
LinkedIn: [linkedin.com/in/stylios-lefkopoulos-81b007a0](https://www.linkedin.com/in/stylios-lefkopoulos-81b007a0)

Reviewer #1:

Remarks to the Author:

This is an appealed manuscript by Meissner et al. investigating the nature and persistence of extraembryonic cells in the mouse. In the previous round of review the data was clear and the hypothesis interesting. However, there were no studies to directly demonstrate that neighboring cells phagocytosed these dying cells. In this revised manuscripts, the authors have added significantly more data to support this claim, including some in vivo imaging experiments using lysotracker as well as p53 null animals. These studies have satisfied my concerns and allow the authors to claim that phagocytosis is the mechanism at play.

Reviewer #2:

Remarks to the Author:

The authors have incorporated a multitude of extra datasets and analyses, effectively addressing all of the issues I raised.

Reviewer #3:

Remarks to the Author:

In the revised manuscript, the authors have taken active steps to incorporate new data to improve their work. However, some pivotal concerns still remain to be effectively addressed, which harm the overall quality, stringency, and impact of the study for Nature Cell Biology.

Major concerns:

1) Using the diploid morula aggregation to track and isolate the extraembryonic gut cells lacks the necessary stringency, which remains a major concern. The authors have not directly responded to the point whether the mCherry+ diploid morula cells contribute to the embryo proper yet.

1a) In the fluorescent images of the E9.5 embryos (Fig.1b, Ext. Data Fig. 2b and 8a), apart from mCherry+ cells in the gut endoderm, weak but visible mCherry signals could also be detected in other embryonic regions. These signals indicate the presence of chimeric mCherry+ cells in non-endodermal tissue. How could the authors exclude the possibilities that these cells are of embryonic origin?

1b) In the differential gene expression analyses (Fig. 3c, Ext. Data Fig. 4b, 8c and 9a), the well-known extraembryonic marker genes were detected in the sorted mCherry+ cells. However, these analyses were based on pooled cell populations, which could mask cellular heterogeneity. For instance, if 20% of mCherry+ cells were embryonic and 80% were extraembryonic, the results should still show similar expression patterns. How could the authors ensure that all the mCherry+ cells are of extraembryonic origin?

2) Since the presence and basic characteristics of extraembryonic gut cells have been reported in previous studies, the novelty of this study is limited without characterizing the biological functions of these cells. My original review had proposed the ablation of these cells with genetic tools to analyze the impact on the embryonic development. These preliminary functional experiments would be necessary to enhance the novelty of this study for Nature Cell Biology.

3) While agree the conclusion that the mCherry+ cells could be cleared by the efferocytosis of GFP+ cells, the possibility of cell fusion could not be entirely ruled out yet. In the live cell imaging (Fig. 2a-b,

Ext. Data Fig. 5a-c), some of the intact mCherry+ cells could be observed overlapping with the GFP+ cells, and these cells did not show any mCherry remnant or lysotracker signal. It is essential to rigorously consider all plausible mechanisms that could result in the dual-labeled cells.

ADDITIONAL COMMENTS TO THE AUTHORS' RESPONSE TO THE POINTS ABOVE

1) The authors claim that first, all the mCherry+ cells in the chimeric embryos obtained by diploid morula aggregation are of extraembryonic origin, and second, these cells exclusively contribute to the endoderm but not other germ layers. However, based on current data, these claims remain technically unconvincing according to the following two points.

1a) In the fluorescent images referred to in my previous comments (Fig.1b, Ext. Data Fig. 2b and 8a), I do not believe these weak mCherry signals present in non-endodermal tissues are entirely attributed to autofluorescence. The authors should not distinguish autofluorescence and authentic signal as they wish. Although the authors have provided confocal images with higher resolutions (Fig. 1c, Ext. Data Fig. 1f), from my experience, these images have been subjected to background subtraction.

1b) In the scRNA-seq analyses of the mCherry+ cells (Fig. 1g, Ext. Data Fig. 4b and 4d), 10 out of the 471 mCherry+ cells (approximately 2.1%) are situated within mesodermal cell clusters. These analyses were even based on isolated gut tissues rather than whole embryos. If these experiments had been performed using whole embryos, higher proportions of mCherry+ cells should have been detected in non-endodermal tissues. The authors should perform scRNA-seq using whole chimeric embryos rather than isolated gut tissues, if they want to draw a conclusion that these mCherry+ cells are entirely in the endoderm.

2) It is true that this study answers the question of permanent or transient presence. However, on the way from the initial discovery of these extraembryonic gut cells to the understanding of their biological functions, the contribution of this study is limited and not significant enough for Nature Cell Biology. Without basic functional analyses, this study lacks sufficient impact to the field.

3) Similar to the question previously raised in the live cell imaging (Fig. 2a-b, Ext. Data Fig. 5b-c), some of the intact mCherry+ cells could still be observed overlapping with the GFP+ cells in the single optical sections (Ext. Data Fig. 3c, the upper most cells, and Ext. Data Fig. 5a). Moreover, in FACS analyses (Ext. Data Fig. 6c-d), low mCherry and high GFP intensities of the dual-labelled cells at later stages do not conclusively indicate that GFP+ cells take up mCherry remnants. The authors should quantify the number of cells exhibiting clear efferocytosis signals and those not involved in the process, while stating the possibility of other mechanisms.

Reviewer #4:

Remarks to the Author:

I have been invited to provide my opinion regarding some of the remaining points raised about this manuscript. Overall, I find that the authors have satisfactorily addressed these points in their last submission and response letter.

Briefly, they used a two-color lineage tracing strategy to track extraembryonic cells during early mouse development. They combined diploid morula aggregation, traditional tetraploid

complementation, and classic blastocyst injection with high-resolution microscopy, to characterize a population of gut endoderm cells (marked by mCherry) of extraembryonic origin. They characterize the genetic and epigenetic signature of these cells and show that they represent a transient population, which is eliminated via cell death (in p53-dependent manner).

Main points:

1) Lineage contribution: There were concerns about the specificity of the diploid complementation approach. The authors provide sufficient evidence that the mCherry-positive cells only contribute to extraembryonic, and not embryonic tissues. This is supported by the gene expression analysis but also by laser scanning microscopy, where they distinguish single cells in 2D planes and show no overlap.

Based on the quality of the images, I do not have further concerns to support that all (or at least most) mCherry-positive cells contribute to extraembryonic tissues. It is true that some overlap can be seen in some examples, but this is likely a result of using simple stereoscopes vs the more careful comparison performed in their 2D planes (as outlined in the authors' responses). I suggest that the authors carefully address this point in their Results section too, as it may help further readers.

2) Function of this cell population: The authors demonstrate that these extraembryonic cells contribute to the gut but are then eliminated. I feel that further exploring the function of these cells is beyond the scope of the current study. Indeed, they may have no major long-term function and only represent a transient population. I do not consider that this diminishes the potential interest of the current findings.

3) Cell elimination mechanism: They provide sufficient mechanistic insight to support the idea that the cells are eliminated in a p53-dependent manner. I do not find sufficient evidence for cell fusion, yet I agree with some of the reviewer comments and suggest that the authors dedicate part of the text to discussing this interesting point without additional experiments.

Author Rebuttal, first revision:

We were glad to see that Reviewer 1 and 2 were satisfied with our extensive revisions. Despite additional explanations via email that were provided to Reviewer 3, they raised further concerns, which we briefly address below together with a response to the newly assigned Reviewer 4 (who agreed with Reviewer 1 and 2).

Reviewer #3:

ADDITIONAL COMMENTS TO THE AUTHORS' RESPONSE TO THE POINTS ABOVE

1) The authors claim that first, all the mCherry+ cells in the chimeric embryos obtained by diploid morula aggregation are of extraembryonic origin, and second, these cells exclusively contribute to the endoderm but not other germ layers. However, based on current data, these claims remain technically unconvincing

according to the following two points.

1a) In the fluorescent images referred to in my previous comments (Fig.1b, Ext. Data Fig. 2b and 8a), I do not believe these weak mCherry signals present in non-endodermal tissues are entirely attributed to autofluorescence. The authors should not distinguish autofluorescence and authentic signal as they wish. Although the authors have provided confocal images with higher resolutions (Fig. 1c, Ext. Data Fig. 1f), from my experience, these images have been subjected to background subtraction.

We would like to respectfully disagree with the reviewer. We have many years of experience in imaging fluorescent embryos and are fortunate to also have a high-end imaging core facility. As such, we are certain that the weak signal is autofluorescence in the images where the whole embryo overview was acquired with a fluorescent stereo microscope. Every single embryo used in this study was carefully assessed using the highest magnification at the fluorescent stereo microscope as well, which also made it absolutely clear that no mCherry+ cell was present outside of the gut tube. We do provide details of these images and most investigators that have used this type of microscope would presumably agree with our interpretation. That said, the data are provided and every scientist can of course apply their own interpretation of the images. Importantly, the core results are not affected by either interpretation.

In the confocal images with higher resolution, our settings allowed us to detect even small mCherry+ foci (remnants of extraembryonic cells), which makes us confident that mCherry+ intact cells outside of the gut tube would have been readily detected if present, however, that was not the case.

But just to be extra sure and as an additional, independent verification, we analyzed the non-endodermal limb tissue of E13.5 embryos generated via our standard lineage tracing (no mCherry+ extraembryonic gut cells survive) and via the lineage tracing combined with extraembryonic-specific p53 knockout (mCherry+ extraembryonic gut cells are still present in endodermal organs). We found in both cases that there are no mCherry+ cells in the limb, providing yet another confirmation that mCherry+ cells of extraembryonic origin do not contribute to the embryo proper (new **Extended Data Fig. 4f and 9f**).

1b) In the scRNA-seq analyses of the mCherry+ cells (Fig. 1g, Ext. Data Fig. 4b and 4d), 10 out of the 471 mCherry+ cells (approximately 2.1%) are situated within mesodermal cell clusters. These analyses were even based on isolated gut tissues rather than whole embryos. If these experiments had been performed using whole embryos, higher proportions of mCherry+ cells should have been detected in non-endodermal tissues. The authors should perform scRNA-seq using whole chimeric embryos rather than isolated gut tissues, if they want to draw a conclusion that these mCherry+ cells are entirely in the endoderm.

Unfortunately, the reviewer must have misunderstood our description. The reviewer states that the scRNA-seq experiment to determine the cell type identity of mCherry+ cells inside the embryo was performed on cells isolated from gut tissues.

That is not the case, as in the experiment on page 3 (Fig. 2), whole embryos were dissociated and used for the cell sorting and then for further analysis, just like the reviewer suggested. The isolated gut tissues were used for bulk RNA-seq and DNA methylome analysis. We, therefore, improved the description of the cell isolation experiment for the scRNA-seq in the Methods of the final revision in order to avoid similar misunderstandings.

E9.5 post-implantation embryo collection and preparation for downstream experiments: “For the single-cell RNA-seq analysis (see below) to determine the cell type identities of mCherry+ and dual+ cells, whole lineage-traced embryos were used.”

Fluorescence-activated Cell Sorting (FACS): “For the single-cell RNA-seq analysis (see below) to determine the cell type identities of mCherry+ and dual+ cells, wild type lineage-traced whole E9.5 embryos were used, and the dual+ cells were sorted into three populations based on the mCherry signal intensities.”

Additionally, only four mCherry+ cells were assigned to non-endodermal cell types (0.8%). We now indicated this in **Extended Data Fig. 3g**. For the cell type assignment the algorithm identifies first ‘anchors’ between the dual+ reference and the mCherry+ cells that should be assigned to it. These represent pairwise correspondences between individual cells (one in each dataset) that are hypothesized to originate from the same biological state. Thus, each cell state assignment is associated with a probability that a cell comes from a particular state and does only represent a prediction (and its certainty). The very limited number of non-endoderm assigned cells display overall lower assignment probabilities than the majority of cells in the assigned cell states with high cell numbers (hindgut, colon, small intestine, foregut). This reflects a higher than normal uncertainty of the assignment, which in our opinion in combination with the very small cell number can be interpreted as most likely misassignments.

2) It is true that this study answers the question of permanent or transient presence. However, on the way from the initial discovery of these extraembryonic gut cells to the understanding of their biological functions, the contribution of this study is limited and not significant enough for Nature Cell Biology. Without basic functional analyses, this study lacks sufficient impact to the field.

In our previous response, we explained in detail why we believe that our discoveries provide significant advances and impact to the field.

3) Similar to the question previously raised in the live cell imaging (Fig. 2a-b, Ext. Data Fig. 5b-c), some of the intact mCherry+ cells could still be observed overlapping with the GFP+ cells in the single optical sections (Ext. Data Fig. 3c, the upper most cells, and Ext. Data Fig. 5a). Moreover, in FACS analyses (Ext. Data Fig. 6c-d), low mCherry and high GFP intensities of the dual-labelled cells at later stages do not conclusively indicate that GFP+ cells take up mCherry remnants. The authors should quantify the number of cells exhibiting clear efferocytosis signals and those not involved in the process, while stating the possibility of other mechanisms.

Based on our assessment, the intact mCherry+ cells do not overlap with GFP+ cells, and cell fusion would not explain the low mCherry and high GFP intensities of the dual-labelled cells.

We have updated the results on page 3 to make our point, while acknowledging the possibility of rare fusion events: “Both the imaging and expression analysis argue against cell fusion as the source of the dual+ cells, though this may occur in rare instances.”

Reviewer #4:

Remarks to the Author:

I have been invited to provide my opinion regarding some of the remaining points raised about this manuscript. Overall, I find that the authors have satisfactorily addressed these points in their last submission and response letter.

Briefly, they used a two-color lineage tracing strategy to track extraembryonic cells during early mouse development. They combined diploid morula aggregation, traditional tetraploid complementation, and classic blastocyst injection with high-resolution microscopy, to characterize a population of gut endoderm cells (marked by mCherry) of extraembryonic origin. They characterize the genetic and epigenetic signature of these cells and show that they represent a transient population, which is eliminated via cell death (in p53-dependent manner).

We would like to thank the reviewer for helping to resolve the situation and providing expert opinion. We appreciate the careful assessment of our data upon short notice and the feedback on Reviewer 3's comments. Below, we provide a short response.

Main points:

1) Lineage contribution: There were concerns about the specificity of the diploid complementation approach. The authors provide sufficient evidence that the mCherry-positive cells only contribute to extraembryonic, and not embryonic tissues. This is supported by the gene expression analysis but also by laser scanning microscopy, where they distinguish single cells in 2D planes and show no overlap.

Based on the quality of the images, I do not have further concerns to support that all (or at least most) mCherry-positive cells contribute to extraembryonic tissues. It is true that some overlap can be seen in some examples, but this is likely a result of using simple stereoscopes vs the more careful comparison performed in their 2D planes (as outlined in the authors' responses). I suggest that the authors carefully address this point in their Results section too, as it may help further readers.

We thank the reviewer and have added more details. Given that we had to shorten the main text substantially, we decided to add details regarding the use of the fluorescent stereo microscope to the Methods section.

Microscopy: "Embryos were imaged using a Zeiss Axio Zoom V16 stereo microscope for brightfield and fluorescence microscopy images to acquire whole embryo overviews. A weak background signal, presumably autofluorescence can be seen, which is often noted and due to the limitations of this technique to eliminate out-of-focus light when thick biological specimens are imaged. Embryos were imaged using a

Zeiss LSM880 laser scanning microscope with Airyscan detector or Zeiss Light sheet LS Z1 microscope to acquire high-resolution images and optical sections.”

2) Function of this cell population: The authors demonstrate that these extraembryonic cells contribute to the gut but are then eliminated. I feel that further exploring the function of these cells is beyond the scope of the current study. Indeed, they may have no major long-term function and only represent a transient population. I do not consider that this diminishes the potential interest of the current findings.

We thank the reviewer for assessing the scope of the current study and highlighting the potential interest of our work.

3) Cell elimination mechanism: They provide sufficient mechanistic insight to support the idea that the cells are eliminated in a p53-dependent manner. I do not find sufficient evidence for cell fusion, yet I agree with some of the reviewer comments and suggest that the authors dedicate part of the text to discussing this interesting point without additional experiments.

As noted above for Reviewer 3, we have updated the results on page 3 to make our point, while acknowledging the possibility of rare fusion events: “Both the imaging and expression analysis argue against cell fusion as the source of the dual+ cells, though this may occur in rare instances.”

Final Decision Letter:

Dear Alex,

I am pleased to inform you that your manuscript, "Extraembryonic gut endoderm cells undergo programmed cell death during development", has now been accepted for publication in Nature Cell Biology. Congratulations!

Over the next few weeks, your paper will be copyedited to ensure that it conforms to Nature Cell

Biology style. Once your paper is typeset, you will receive an email with a link to choose the appropriate publishing options for your paper and our Author Services team will be in touch regarding any additional information that may be required.

Publication is conditional on the manuscript not being published elsewhere and on there being no announcement of this work to any media outlet until the online publication date in *Nature Cell Biology*.

Please note that *Nature Cell Biology* is a Transformative Journal (TJ). Authors may publish their research with us through the traditional subscription access route or make their paper immediately open access through payment of an article-processing charge (APC). Authors will not be required to make a final decision about access to their article until it has been accepted. Find out more about Transformative Journals

If you have not already done so, we strongly recommend that you upload the step-by-step protocols used in this manuscript to the Protocol Exchange (www.nature.com/protocolexchange), an open online resource established by Nature Protocols that allows researchers to share their detailed experimental know-how. All uploaded protocols are made freely available, assigned DOIs for ease of citation and are fully searchable through nature.com. Protocols and Nature Portfolio journal papers in which they are used can be linked to one another, and this link is clearly and prominently visible in the online versions of both papers. Authors who performed the specific experiments can act as primary authors for the Protocol as they will be best placed to share the methodology details, but the Corresponding Author of the present research paper should be included as one of the authors. By uploading your Protocols to Protocol Exchange, you are enabling researchers to more readily reproduce or adapt the methodology you use, as well as increasing the visibility of your protocols and papers. You can also establish a dedicated page to collect your lab Protocols. Further information can be found at www.nature.com/protocolexchange/about

With kind regards,
Stelios

Stylianos Lefkopoulos, PhD
He/him/his
Senior Editor, Nature Cell Biology
Springer Nature
Heidelberger Platz 3, 14197 Berlin, Germany

E-mail: stylianos.lefkopoulos@springernature.com
Twitter: @s_lefkopoulos
LinkedIn: [linkedin.com/in/stylianos-lefkopoulos-81b007a0](https://www.linkedin.com/in/stylianos-lefkopoulos-81b007a0)

** Visit the Springer Nature Editorial and Publishing website at www.springernature.com/editorial-and-publishing-jobs for more information about our career opportunities. If you have any questions please click here.**